# Cerebrospinal fluid-contacting neurons are sensory neurons with uniform morphological and region-specific electrophysiological properties in the mouse spinal cord
Elysa Crozat [1,2], Edith Blasco [1,2], Jorge Ramirez-Franco [1,2], Priscille Riondel[1], Nina Jurčić [1], Riad Seddik [1], Caroline Michelle [1], Jérôme Trousland [1] & Nicolas Wanaverbecq [1] ✉

Cerebrospinal Fluid-contacting neurons (CSF-cNs) are GABAergic bipolar neurons found, in contact with the cerebrospinal fluid, along the vertebrate medullo-spinal central canal. They express Polycystin Kidney Disease 2-Like 1 channels (PKD2L1), members of the Transient Receptor Potential superfamily, and were shown to modulate motor activity and therefore suggested to act as a novel sensory system. However, in mice, they remain largely uncharacterized and it is crucial to comprehensively characterize their morphological and electrophysiological properties to determine whether they form a homogenous neuronal population and understand their role in the CNS. We show that CSF-cNs are distributed throughout the spinal cord with a uniform morphology and a primarily ventral localization. They exhibit region-specific properties, expression of voltage-dependent and ligand-gated conductances and detect variation in extracellular pH through activation of PKD2L1 and Acid-sensing Ion Channels. They possess GABA$_B$ and muscarinic receptors, not glutamatergic metabotropic ones, to modulate Ca$^{2+}$ channels. CSF-cNs represent unique sensory neurons with a uniform morphology and electrophysiological properties that appear specific to the spinal cord segment inserted in. The future challenges in the field, will be to elucidate the physiological stimuli activating CSF-cNs and the neuronal network they are integrated in to modulate body function through specific local spinal network.

Neurons in contact with cerebrospinal fluid (CSF-cNs) are GABAergic bipolar neurons present around the central canal (cc) in vertebrates[1–10]. They have a unique morphology with a soma located beneath or within the ependymal cell monolayer lining the cc[1–9] and a single dendrite extending to the cc lumen ending in a ciliated protrusion[7,9,11,12]. Their axon extends into the ventral spinal cord (SC), and form long, bilateral fiber bundles at the median fissure[5,11,13–15] running along several SC segments and extending collaterals within the spinal tissue[14–17] with recurrent synapses onto CSF-cNs[14,15].

In rodents, CSF-cNs exhibit spontaneous action potential (AP) firing, mediated by sodium and potassium voltage-dependent channels[9], and they also express voltage-dependent calcium channels[18–20]. Further, CSF-cNs have sensory properties with the expression of Polycystin Kidney Disease

2-Like 1 (PKD2L1) channels that detect changes in extracellular pH and osmolarity (chemosensitivity)[12,21–24] and respond, in zebrafish larvae, to CSF flow, and SC bending (mechanosensitivity)[25–27]. Additionally, they express acid-sensing ion channels (ASICs)[12,21,23] and both channels are capable of modulating CSF-cN excitability[21,23]. Medullary CSF-cNs were shown to express classical synaptic receptors and to receive mainly inhibitory inputs (GABA$_A$ and glycine receptors)[12,18] but also excitatory ones (AMPA/kainate receptors)[6,18]. These synaptic inputs are modulated by homo- and hetero-synaptic activation of metabotropic GABAergic receptors (GABA$_B$-Rs)[18] suggesting CSF-cNs are inserted in local neuronal networks.

In zebrafish larvae and juvenile rodents, CSF-cNs present in the cc ventral region exhibit immature phenotypes (expression of doublecortine,

[1]Institut de Neurosciences de la Timone, Aix Marseille Université (AMU) & CNRS, Marseille, France. [2]These authors contributed equally: Elysa Crozat, Edith Blasco, Jorge Ramirez-Franco. ✉e-mail: nicolas.wanaverbecq@univ-amu.fr

PSA-NCAM and homeoboxes Nkx2.2 and 6.1)[5,8,17,28,29] and fire a single AP[8,28], a feature often associated to immaturity[30]. In contrast, dorso-lateral CSF-cNs have a more mature phenotype (expression of the neuronal nuclear protein, NeuN) and tonic AP discharge[8,17,28,29]. Therefore CSF-cNs are thought to be grouped in two subpopulations. In older mice, although not localized in specific clusters, two subpopulations of CSF-cNs exist. They show expression of these immaturity markers, exhibit two distinct AP firing patterns and, in lumbar segments, can be distinguished based on their responses to GABAergic signaling, either inducing depolarization or hyperpolarization[31].

At the behavioral level, in zebrafish larvae, CSF-cNs selectively activate motor neurons and interneurons to influence swimming activity[25,32,33], while, in mice, they were shown to modulate posture, balance[15], and adaptive locomotion[14]. Further, in zebrafish larvae, they participate in the body's immune defense by detecting bacterial toxins in the CSF[34]. Finally, recent reports indicate that, in mice, spinal CSF-cN constitutive activation through κ-opioid signaling is halted, leading to disinhibition of ependymal cell proliferation to promote scar formation following SC injury[35]. CSF-cNs appear to play a key role in integrating sensory and chemical signals to modulate a large set of body functions.

Along the SC axis, specific local networks control specific physiological functions, and one can wonder whether CSF-cNs inserted within a given SC segment would exhibit specific morpho-functional properties. Although this information is crucial to better characterize CSF-cN physiology and demonstrate their role along the SC, such an analysis has not been carried out. We therefore conducted a comprehensive and systematic study to examine spinal CSF-cN anatomical and electrophysiological properties from the cervical to the lumbar segments and assess whether regional differences along the mouse SC levels can be observed.

We found that spinal CSF-cNs, primarily located in the cc ventral region, form a dense morphologically uniform neuronal population that shares similar sensory properties to integrate signals circulating in the CSF along the SC. However, they exhibit region-specific electrophysiological features that might serve the specific role they play in a given spinal network. Anatomical and functional evidence suggests that CSF-cNs act as a novel sensory system intrinsic to the CNS. Our study provides novel cues on the physiology of spinal CSF-cNs and is crucial for the deeper understanding of mammalian CSF-cNs within the CNS. It sets ground for the future challenges in the field to determine their integration and modulation within local spinal and supraspinal networks and to ultimately demonstrate their role in the modulation of body functions both in physiological and pathological conditions.

## Results

### CSF-cNs exhibit a uniform morphology and distribution along the spinal cord

CSF-contacting neurons (CSF-cNs) are found along the entire cc in lamprey[6], zebrafish larvae[5,17], turtle[36], rat[9], mouse[7,12–15,31,35] and macaques[4,10,37] where they exhibit a consistent morphology. In this study, we assessed the distribution and density of CSF-cNs in the mouse SC using the Pkd2l1-Cre::tdTomato (see Methods section for details) mouse model, to selectively label CSF-cNs with fluorescent tdTomato. CNS tissue was cleared using the vDISCO[38,39] method and imaged *via* light sheet microscopy, allowing 3D visualization of the entire SC. Figure 1A shows the anatomy of the CNS (tissue autofluorescence) and the localization of CSF-cNs along the cc (red line). Imaging of cervical, thoracic, and lumbar regions at higher magnification (Boxes 1-3 in Fig. 1A, *Bottom*) reveals CSF-cN cell bodies distributed along the cc in all segments (Fig. 1B–D). Their axons projected into the ventral SC, forming bilateral fiber bundles (Fig. 1B–D, *Right* caudo-rostral view). These neurons were also found in distal ventral regions, consistent with previous studies[13]. 3D reconstruction images were segmented to quantify CSF-cN cell bodies (yellow objects) as well as their axonal projection (labeled in blue; Fig. 1B–D, *Bottom*). To better resolve CSF-cN distribution and quantification along the cc axis, we acquired images using higher magnification objectives (4x and 12x; see Supplementary Fig. 1).

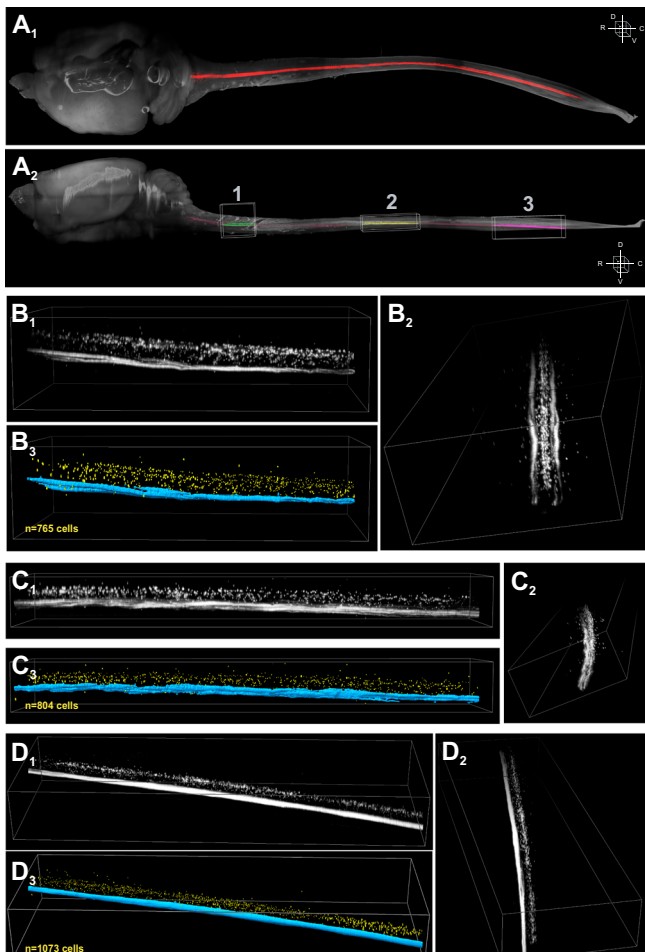

**Fig. 1 | Localization of spinal CSF-contacting neurons in the mouse central nervous system. A** Dorsal view (**A₁**) of the full rostro-caudal distribution of the CSF-cN system (red) within the mouse CNS (grey). **A₂** Lateral view of the image shown in the *Top* Panel. Three boxes are indicated to depict representative cervical (1, red), thoracic (2, blue), and lumbar (3, green) regions acquired at higher magnifications. Boxes sizes are in µm along the Rostro-Caudal (RC) x Latero-Median (LM) x Dorso-Ventral (DV): (1) 1764.75 × 1825.50 × 1726.00; (2) 3076.13 × 1784.25 × 818.00; (3) 3734.25 × 1228.50 × 1192.00. **B–D** Higher magnification views and segmentation for the cervical (**B₁₋₃**), thoracic (**C₁₋₃**) and lumbar (**D₁₋₃**) SC segments for the boxes labelled in Fig. 1A₂, as 1, 2 or 3 respectively (see color code in Fig. 1A). For each Panel: **B₁**, **C₁** and **D₁**. Higher magnification views of the boxes in Fig. 1A with the same orientation. **B₂**, **C₂** and **D₂**. 3D representation of the segmentation and cell counting of CSF-cNs for each SC segments (see Methods for details; CSF-cNs in yellow and axon bundles in blue). **B₃**, **C₃** and **D₃**. 90° rotation view of the image in the **B₁**, **C₁** and **D₁** Panels. Dimensions of the box are for the RC x LM x DV axes (in µm): cervical segment (**B**), 1764.75 × 913.25 × 463.00; thoracic segment (**C**), 3076.13 × 658.15 × 322.00 and lumbar segment (**D**), 3734.25 × 1228.50 × 492.00.

However, due to the small size of CSF-cNs, clustering tendencies, and tissue shrinkage from the clearing method, we faced technical limitations and reached microscopy optical limits to resolve and identify single CSF-cN somata (see Supplementary Fig. 2). To validate our clearing technique and light-sheet microscopy acquisitions, we conducted two additional experiments. First, we performed viral delivery into the ventricular system[14] of ChAT-Cre::tdTomato mice to infect CSF-cNs, allowing us to compare the size of cholinergic neurons and CSF-cNs within the same preparation (Supplementary Fig. 2A, B). Second, we cleared the spinal cords of ChAT-Cre::tdTomato mice and imaged them using the same settings previously applied to CSF-cNs. Under these conditions, we observed that the technical limitations of light-sheet imaging affect CSF-cNs but no other neuronal populations, likely due to their size differences (Supplementary Fig. 2).

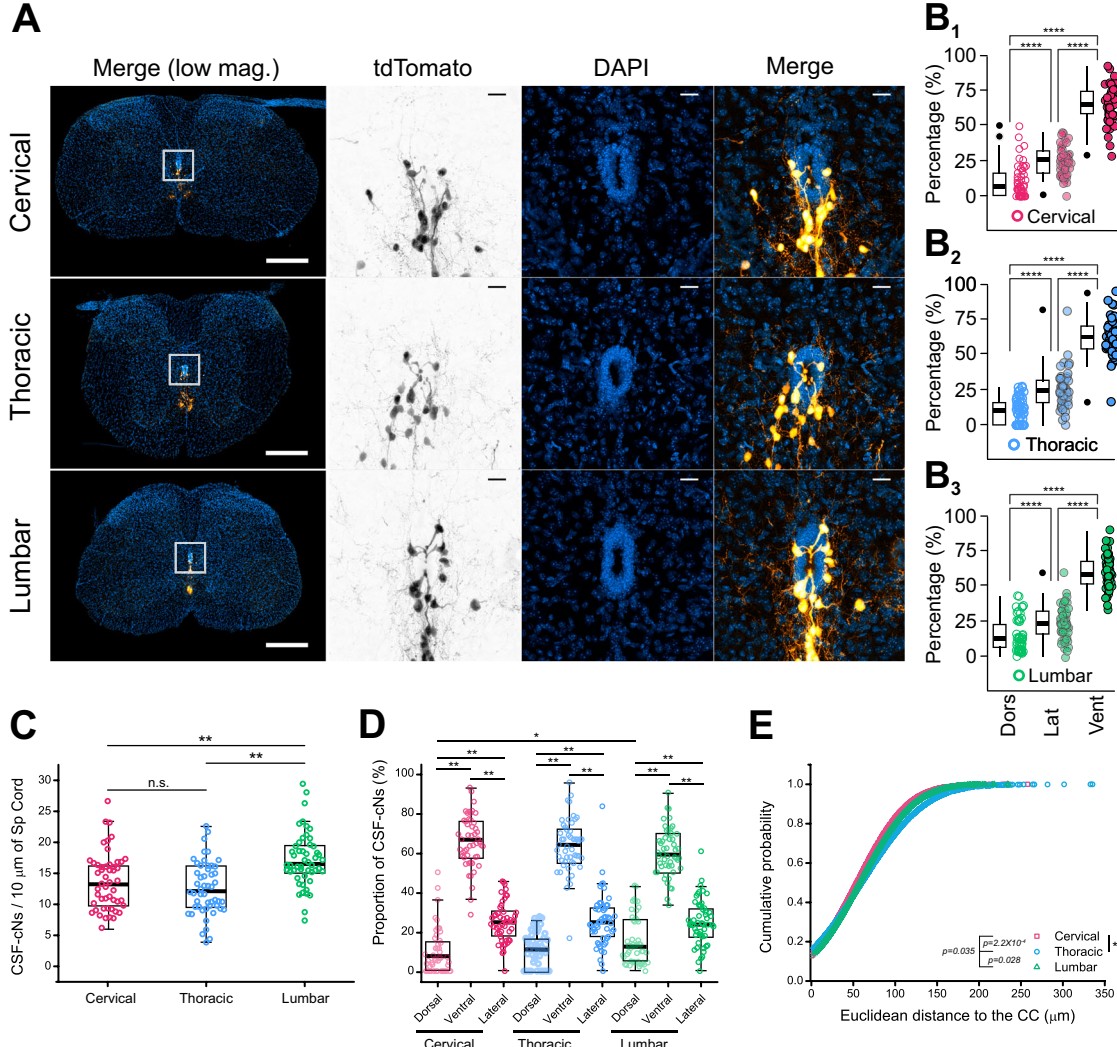

**Fig. 2 | Confocal analysis of CSF-cN distribution in the mouse spinal cord.**
**A** Confocal images of tdTomato fluorescence in 3 weeks-old Pkd2l1-Cre::tdTomato mouse at cervical, thoracic, and lumbar levels of the SC. tdTomato fluorescence alone (inverted greyscale image), merge images (orange hot) and DAPI (blue). Scale bar = 200 μm in *Left* panels; 20 μm in *Middle* and *Right* panels. **B)** Summary boxplots with whiskers of CSF-cN distribution in dorsal (Dors), lateral (Lat), and ventral (Vent) quadrants at cervical ($B_1$), thoracic ($B_2$) and lumbar ($B_3$) levels of the SC (see Methods section for details; color code as for Fig. A). (C: $N = 3$, $n = 50$; T: $N = 6$, $n = 11$ and L: $N = 4$, $n = 9$). *ANOVA.lme*: F = 229.4, df=8 and 261, p(F) < 2.2 × 10$^{-16}$ and *Tukey (EMM) post hoc* test to compare Dorsal *vs.* Lateral and Ventral and Lateral vs. ventral within the different regions: ****$p < 0.0001$ in C, T and L segments. There

is no difference in the distribution within quadrant between C, T and L regions. **C** CSF-cN density per 10 μm of SC tissue at cervical, thoracic and lumbar levels. *Kruskal-Wallis rank sum test*: $\chi^2 = 18.745$, df=2, p($\chi^2$) = 8.505 × 10$^{-5}$ and *post hoc pairwise comparisons using Wilcoxon rank sum test* with continuity correction: C *vs.* T, $p = 0.64140$; C *vs.* L, $p = 6.9 × 10^{-4}$ and T *vs.* L, $p = 2.7 × 10^{-4}$. **D** CSF-cN distribution at the different SC levels. *Kruskal-Wallis ANOVA* followed by *Dunn's post hoc test*, *$p < 0.05$; **$p < 0.0$; ns: not significative. **E** Cumulative distribution function plots of the euclidean distances of CSF-cNs to the cc lumen at the SC levels of interest. *Kolmogorov-Smirnov test*. *$p < 0.05$; ***$p < 0.001$. Dots represent each slice in Figs. 2B, C and each individual cell in Fig. 2D.

---

Further analysis was carried out using thin sections and confocal microscopy (Fig. 2A) prepared from 3-week-old mice. Since our electrophysiological analyses were carried out in mice aged from 3-6 weeks, 6-week-old mice were also tested to assess potential differences in the anatomical properties, but it did not highlight such differences. A larger proportion of CSF-cNs is located in the ventral cc (Fig. 2B; Percentage of cells for Dorsal: 6 ± 6%, 9 ± 7% and 11 ± 8%; Lateral: 24 ± 11%, 24 ± 11% and 24 ± 8% and Ventral: 71 ± 11%, 66 ± 11% and 65 ± 10% Data in the order C, T and L and $N = 3$, $n = 30$ from all segments. *ANOVA.lme*: F = 229.4, df = 8 and 261, p(F) < 2.2 × 10$^{-16}$ and *Tukey (EMM) post hoc* test to compare Dorsal *vs.* Lateral and Ventral and Lateral *vs.* Ventral within the different regions: ****$p < 0.0001$ in C, T and L segments. There is no difference in the distribution within quadrants between C, T and L regions.; and see Fig. 2D). Our results show that CSF-cNs are present in all SC segments but at a higher density in the lumbar compared to thoracic and cervical regions (Fig. 2C;

Density:13 ± 4, 12 ± 4 and 18 ± 5 cells per 10 μm tissue depth, Data in the order C, T and L and $N = 3$, $n = 30$ sections from all segments. *Kruskal-Wallis rank sum test*: $\chi^2 = 18.745$, df=2, p($\chi^2$) = 8.505 × 10$^{-5}$ and *post hoc pairwise comparisons using Wilcoxon rank sum test* with continuity correction: C *vs.* T, $p = 0.64140$; C *vs.* L, $p = 6.9 × 10^{-4}$ and T *vs.* L, $p = 2.7 × 10^{-4}$) and neurons are closer to the cc in cervical and lumbar segments but further away in the thoracic one (Fig. 2E).

These findings suggest that, along the SC, CSF-cNs form a uniform population with a conserved morphology, a predominantly ventral localization and a larger density in the lumbar region.

## Spinal CSF-cN intrinsic and sensory properties along the central canal axis

**CSF-cNs have region-specific passive properties and firing patterns.** CSF-cNs are found along the entire cc axis. However, only

## Table 1 | Intrinsic properties of CSF-cNs along the rostro-caudal central canal axis

| Spinal Segment | | Cervical (C) | Thoracic (T) | Lumbar (T) |
|---|---|---|---|---|
| **Input resistance** | $R_m$ | 3.6 ± 2.1 GΩ | 3.6 ± 1.9 GΩ | 4.1 ± 12.1 GΩ |
| Kruskal-Wallis's test: : $\chi^2$ = 5.5711; degree of freedom (df)=2; $p$ = 0.0617<br>Post hoc pairwise Wilcoxon rank sum test: C *vs*. T, $p$ = 0.826, ns; C *vs*. L, $p$ = 0.069, ns & T *vs*. L, $p$ = 0.069, ns | | | | |
| **Membrane capacitance** | $C_m$ | 5.6 ± 2.9 pF | 5.0 ± 2.5 pF | 6.2 ± 2.9 pF |
| Kruskal-Wallis's test: $\chi^2$ = 25.722; df=2; $p$ = 2.597.10-6<br>Post hoc pairwise Wilcoxon rank sum test: C *vs*. T, $p$ = 0.0064, **; C *vs*. L, $p$ = 0.0242, * & T *vs*. L, $p$ = 1.3.10-6, *** | | | | |
| **Membrane time constant** | $\tau_m$ | 92 ± 58 µs | 86 ± 49 µs | 90 ± 49 µs |
| Kruskal-Wallis's test: $\chi^2$ = 3.8282; df = 2; $p$ = 0.1475<br>Post hoc pairwise Wilcoxon rank sum test: C *vs*. T, $p$ = 0.40, ns; C *vs*. L, $p$ = 0.57, ns & T *vs*. L, $p$ = 0.12, ns | | | | |
| **N (animals), *n* (cells)** | | 50, 227 | 65, 239 | 70, 193 |
| **Resting Membrane Potential** | RMP | −46 ± 4 mV | −45 ± 6 mV | −61 ± 19 mV |
| Kruskal-Wallis's test: $\chi^2$ = 18.368; df = 2; $p$ = 1.027.$10^{-5}$<br>Post hoc pairwise Wilcoxon rank sum test: C *vs*. T, $p$ = 0.73535, ns; C *vs*. L, $p$ = 0.00779, * & T *vs*. L, $p$ = 0.00025, *** | | | | |
| ***N* (animals), *n* (cells)** | | 6, 20 | 10, 41 | 9, 64 |

medullary CSF-cNs in mice have been extensively studied[12,21]. Here, we characterized the intrinsic properties of cervical (C-), thoracic (T-), and lumbar (L-) CSF-cNs to determine if they form a homogenous population. Consistent with previous studies in the brainstem[12,21], spinal CSF-cNs have a high input resistance ($R_m$, 3.7 ± 0.3 GΩ; N = 191, n = 659; N and n for the number of animals used and the sample size, respectively) and small membrane capacitance ($C_m$, 5.6 ± 0.6 pF; N = 191, n = 659), with a fast membrane time constant ($\tau_m$, 89 ± 3 µs; N = 191, n = 659) in agreement with their small soma and dendritic arborization. When comparing across spinal segments, our data indicate that CSF-cNs exhibit similar $R_m$ and membrane time constant in all segments, while $C_m$ was the highest in L-CSF-cNs (lowest in T-CSF-cNs) compared to C- and T-CSF-cNs (Table 1). Here and in the following experiments, recordings have been conducted on CSF-cNs localized in the different quadrants (dorsal, lateral and ventral) around the cc to test whether we could reveal localization-specific properties. However, our data do not indicate that CSF-cNs would group in functional clusters. We therefore pool together our data according to their presence in a given segment, not a specific localization around the cc.

In current-clamp mode (I = 0), resting membrane potentials (RMP) is on average −51 ± 9 mV (N = 25, n = 125), with L-CSF-cNs being hyperpolarized compared to more depolarized CSF-cNs in rostral regions (Table 1; aCSF and intracellular solution A, Supplementary Tables 1 and 2). AP discharge patterns were also assessed (Fig. 3). In line with previous findings in juvenile rats[9] and mice[8], spinal CSF-cNs in older mice exhibit either tonic or single spike discharges. Following a positive current injection step ( + 10 pA, 200–500 ms duration) from RMP, 58% of the neurons showed tonic AP firing (38 cells out of 65 recorded), while 42% fired a single AP (27 cells out of 65 recorded; Fig. 3). Comparative analysis between regions revealed that most L-CSF-cNs and C-CSF-cNs had tonic AP discharges while T-CSF-cNs exhibited a primarily single-spike pattern (Fig. 3B). Nevertheless, in contrast with the situation observed in juvenile rodents, we could not associate a specific discharge pattern with CSF-cN localization around cc.

**CSF-cN modulation by extracellular pH through the activity of PKD2L1 and ASICs.** In agreement with their expression of PKD2L1 channels[5,12,21,22,27], we observed spontaneous unitary PKD2L1 activity in CSF-cNs recorded at a holding potential ($V_h$) of −80 mV (aCSF with all synaptic receptors blocked, see *Material and Methods*), with an average current amplitude of −13 ± 2 pA and an open probability (NPo) of 0.03 ± 0.04 (N = 23, n = 60; Fig. 4A, B). No differences were found in amplitude or NPo across spinal segments. PKD2L1, a sensory chemoreceptor, responds to pH changes[12,21,22] and exposure to alkaline pH (pH9) causes no change in current amplitude (−13 ± 2, −13 ± 2 and −13 ± 2 pA in control, pH9 and Wash, respectively; N = 13, n = 32; *Kruskal-Wallis*

*rank sum test*: $\chi^2$ = 2.7856, df = 2, p($\chi^2$) = 0.2484 and *post hoc pairwise comparisons using Wilcoxon rank sum test* with continuity correction: CTR *vs*. pH9, $p$ = 0.36; CTR *vs*. Wash, $p$ = 0.36 and Wash *vs*. pH9, $p$ = 0.57) but increased NPo by 20% (16 ± 30%) from 0.04 ± 0.05 to 0.16 ± 0.12 before returning to baseline value upon Wash (0.08 ± 0.1; *Kruskal-Wallis rank sum test*: $\chi^2$ = 27.219, df = 2, p($\chi^2$) = 1.229 × $10^{-6}$ and *post hoc pairwise comparisons using Wilcoxon rank sum test* with continuity correction: CTR *vs*. pH9, $p$ = 1.3 × $10^{-7}$; CTR *vs*. Wash, $p$ = 0.16162 and Wash *vs*. pH9, $p$ = 0.00045; N = 13, n = 32). Again, all spinal CSF-cNs show a similar response to alkaline pH exposure (Fig. 4A, B).

PKD2L1 acts as an AP generator at the single channel level and exposure to alkaline pH increases PKD2L1 activity as well as CSF-cN excitability in the *medulla*[21]. In recordings conducted in current-clamp mode at RMP (aCSF with all synaptic receptors blocked, see *Material and Methods*), all spinal CSF-cNs exhibit an increase in the AP firing frequency from 1.6 ± 2.1 Hz in control to 2.4 ± 2.4 Hz under alkaline pH, which also reversed upon wash (frequency in Wash: 1.4 ± 2.0 Hz; Fig. 4C, D; *Kruskal-Wallis rank sum test*: $\chi^2$ = 9.4037, df = 2, p($\chi^2$) = 0.009079 and *post hoc pairwise comparisons using Wilcoxon rank sum test* with continuity correction: CTR *vs*. pH9, $p$ = 0.012; CTR *vs*. Wash, $p$ = 0.902 and Wash *vs*. pH9, $p$ = 0.012; N = 12, n = 32).

In contrast, exposure to acidic pH (pH5; same extracellular and intracellular solutions as above) inhibited PKD2L1 activity, reducing the current amplitude from −13 ± 2 pA in control to 0.9 ± 2.1 pA in pH5 (Fig. 4E), followed by a recovery to −14 ± 2 pA upon Wash (N = 10, n = 28; *Kruskal-Wallis rank sum test*: $\chi^2$ = 55.896, df=2, p($\chi^2$) = 7.282 × $10^{-13}$ and *post hoc pairwise comparisons using Wilcoxon rank sum test* with continuity correction: CTR *vs*. pH5, $p$ = 3.9 × $10^{-16}$; CTR *vs*. Wash, $p$ = 0.27 and Wash *vs*. pH5, $p$ = 3.9 × $10^{-16}$). NPo also decreased reversibly under acidic conditions from 0.024 ± 0.018 in control to 0.001 ± 0.001 in the presence of acidic solution (Fig. 4E,F; Wash: 0.048 ± 0.076; N = 10 n = 28; *Kruskal-Wallis rank sum test*: $\chi^2$ = 52.84, df = 2, p($\chi^2$) = 3.357 × $10^{-12}$ and *post hoc-pairwise comparisons using Wilcoxon rank sum test* with continuity correction: CTR *vs*. pH5, $p$ = 7.1 × $10^{-10}$; CTR *vs*. Wash, $p$ = 0.95 and Wash *vs*. pH5, $p$ = 8.2 × $10^{-10}$).

Additionally, acidic pH triggered in all spinal CSF-cNs a large transient inward current (on average −801 ± 411 pA; N = 10, n = 24) followed by a persistent phase (Fig. 4G), typical of ASICs activation and previously reported in medullary CSF-cNs[12,21] (see also Jalavand and colleagues[23]). The ASICs currents had similar amplitude in all spinal CSF-cNs recorded (C: -712 ± 289 pA; T: −685 ± 189 pA and L: −1005 ± 599 pA; n = 8 for each segment; *Kruskal-Wallis rank sum test*: $\chi^2$ = 0.74, df = 2, p($\chi^2$) = 0.6907 and *post hoc pairwise comparisons using Wilcoxon rank sum test* with continuity correction: C *vs*. T, $p$ = 0.76; C *vs*. L, $p$ = 0.96 and T *vs*. L, $p$ = 0.76). During

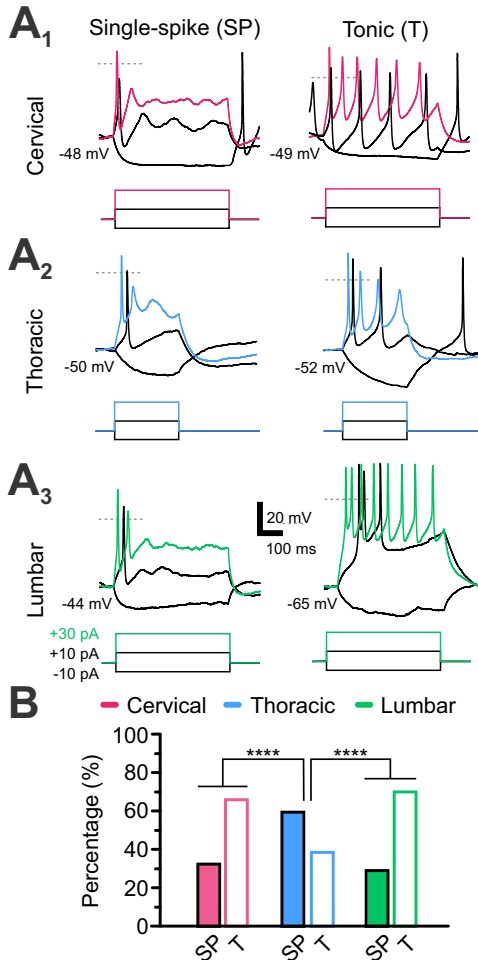

**Fig. 3 | Cervical and lumbar CSF-cNs mainly exhibit tonic AP firing while thoracic ones show single-spiking. A** Representative voltage traces for CSF-cNs recorded in the three regions of interest ($A_1$, cervical (C); $A_2$, thoracic (T) and $A_3$, lumbar (L) from RMP (indicated below the traces). Variation in the membrane potential was elicited with DC current injection steps (−10, +10 and +30 pA for 500 and 250 ms in $A_1$, $A_3$ and $A_2$, respectively) and upon positive current injections, CSF-cNs exhibit either single-spike (SP) or tonic (T) firing, Gray dashed lines indicate the 0 mV level. **B** Summary histogram for the proportion (in percent) of CSF-cNs showing either SP or T firing patterns at C, T and L levels (C: $N = 10$ for the number of animals, $n = 21$ for the sample size; T: $N = 12$, $n = 23$ and L: $N = 9$, $n = 21$). *Fisher's Exact test*, $p = 3.772 \times 10^{-6}$ and *post hoc* pairwise comparisons using *Wilcoxon rank sum test* with continuity correction: C *vs.* T, $p = 1.19 \times 10^{-4}$; C *vs.* L, $p = 0.6466$ and T *vs.* L, $p = 8.88 \times 10^{-6}$).

the persistent phase of ASICs current, PKD2L1 channels were inhibited (colored boxes in Fig. 4G and see 4E). In current-clamp recordings at RMP, ASICs activation led on average to a +37 ± 9 mV depolarization (Fig. 4H, $I_1$; $N = 11$, $n = 38$) that was large enough to trigger APs in all spinal CSF-cNs (Fig. 4H, $I_2$). The current properties observed in our study are in line with those reported in previous studies[21–23] and suggest that they are mediated by PKD2L1 channels and ASIC. An observation that is further supported by the transcriptomic analysis recently carried out by Yue and collaborators[35] where gene for PKD2L1 and ASICs 1 and 2 can be found (see Supplementary Fig. 3A and Discussion).

Overall, our findings suggest that CSF-cNs are highly resistive neurons showing passive properties as well as firing patterns that differ along the cc. They further exhibit shared chemosensory functions by responding to variations in extracellular pH. These differences were observed between CSF-cNs recorded in different segments but not associated to a specific localization around the cc.

## Spinal CSF-cNs express functional sodium, potassium and calcium voltage-dependent channels

Marichal and colleagues[9] showed functional sodium ($Na_V$) and potassium ($K_V$) voltage-dependent conductances in juvenile rat CSF-cNs. We analyzed these neurons in mouse SC to identify the presence and types of voltage-dependent conductances.

We first analyzed $Na^+$ voltage-dependent currents. Recordings of spinal CSF-cNs in voltage-clamp mode at $V_h$ -80 mV with specific solutions to isolate $I_{Na}$ currents ($Na_V$ solution and intracellular solution B, Supplementary Tables 1 and 2) reveal fast, inactivating inward currents in all cells during incremental voltage steps ($V_{Step}$ from −60 to +60 mV, $\Delta V = +10$ mV, 100 ms; Fig. 5A *Left*). The current activates at $V_{Step}$ more depolarized than −40 mV, peaks at 0 mV with an average current amplitude of −712 ± 450 pA ($N = 10$, $n = 41$) and a current density of −124 ± 67 pA.pF$^{-1}$ ($N = 10$, $n = 41$; Fig. 5B, C). Subsequently the current decreases for more depolarized $V_{Step}$ and reversed above +50 mV in agreement with the calculated sodium equilibrium potential ($E_{Na} = +46$ mV, Supplementary Table 1). We did not observe difference in the $Na^+$ current amplitude between C-, T- and L-CSF-cNs (Fig. 5C). Tetrodotoxin (TTX, 0.5 μM; Fig. 5A *Right*, violet traces) completely blocked the current confirming expression of functional TTX-sensitive $Na_V$ channels in spinal CSF-cNs (Fig. 5D; inhibition by 106 ± 9% from -141 ± 71 in control to 8 ± 11 pA.pF$^{-1}$ in TTX ($N = 10$, $n = 28$; cervical: −139 ± 65 pA.pF$^{-1}$ and 8 ± 11, pA.pF$^{-1}$ ($N = 4$, $n = 12$); thoracic: −178 ± 92 pA.pF$^{-1}$ and 10 ± 14 pA.pF$^{-1}$ ($N = 4$, $n = 6$) and lumbar: −122 ± 61 pA.pF$^{-1}$ and 7 ± 9 pA.pF$^{-1}$ ($N = 2$, $n = 10$) in control and TTX, respectively; *ANOVA.lme*: F = 25.87, df=5 and 50, p(F) = 8.582 × 10$^{-13}$ and *Tukey (EMM) post hoc test* to compare CTR *vs.* TTX within the different regions: ***$p < 0.0001$ in cervical, thoracic and lumbar segments. There is no difference between CTR and TTX between regions). The analysis of the gene expression in spinal CSF-cNs confirms the presence of TTX-sensitive $Na_V$ in spinal CSF-cNs (Supplementary Fig. 3B).

In juvenile rats[9], delayed rectifier ($I_{KD}$) and A-type K$^+$ currents ($I_A$) were identified. To determine the functional expression of voltage-dependent K$^+$ channels ($K_V$) in mouse spinal CSF-cNs, we conducted recordings at $V_h$ -80 mV using specific solutions to isolate $K_V$ currents ($K_V$ solution and intracellular solution A, Supplementary Tables 1 and 2) and applied a 50 ms prestep to -100 mV, followed by $V_{Step}$ from −40 to +60 mV ($\Delta V = +10$ mV, 100 ms; Fig. 6A). The elicited current had an average current amplitude at the peak of 5478 ± 2012 pA and a current density of 900 ± 402 pA.pF$^{-1}$ at +60 mV ($N = 14$, $n = 60$). The I–V curve (Fig. 6B), in agreement with the potassium reversal potential ($E_K$ = −94 mV, Supplementary Table 1), shows outward currents activated for $V_{Step}$ larger than -30 mV and the current increased proportionally to the $V_{Step}$ amplitude. Typically, the activated currents presented an initial transient phase (Peak (E, $): 900 ± 402 pA.pF$^{-1}$) followed by a persistent one (Persistent (P, #): 778 ± 368 pA.pF$^{-1}$; $N = 14$, $n = 60$) lasting for the whole duration of $V_{Step}$.

Application of TEA (10 mM) reduced the current by 56 ± 18% (current density of 344 ± 131 pA.pF$^{-1}$; $N = 14$, $n = 39$; Fig. 6A, current traces +TEA and Fig. 6D) and the TEA-sensitive (digital subtraction of the current recorded in control minus that in TEA; blue traces in Fig. 6A) exhibited characteristic kinetics of $I_{KD}$ (Fig. 6A). Adding 4-AP (4 mM; +4-AP) further decreased the current by 44 ± 29% (amplitude of 183 ± 69 pA.pF$^{-1}$; $N = 14$, $n = 34$; Fig. 6A, D). The 4-AP sensitive current (digital subtraction of the current recorded in TEA only minus that in TEA and 4-AP; dark blue traces in Fig. 6A) exhibits fast-rising phase, and rapid inactivation compatible with $I_A$ (Fig. 6A). The remaining current was likely non-selective or K$^+$ leak currents. The functional expression of delayed and A-type/4AP sensitive potassium channels is supported by the presence of their genes in spinal CSF-cNs (Supplementary Fig. 3C).

In a previous report, we indicated that mouse medullary CSF-cNs express functional voltage-dependent Ca$^{2+}$ channel (Ca$_V$) mainly of the N-type (Ca$_V$2.2)[18] while in juvenile rats[9] they appear to express both High (HVA) and Low Voltage-Activated (LVA) Ca$^{2+}$ conductances. To investigate this in spinal CSF-cNs, we isolated Ca$^{2+}$ currents (Ca$_V$

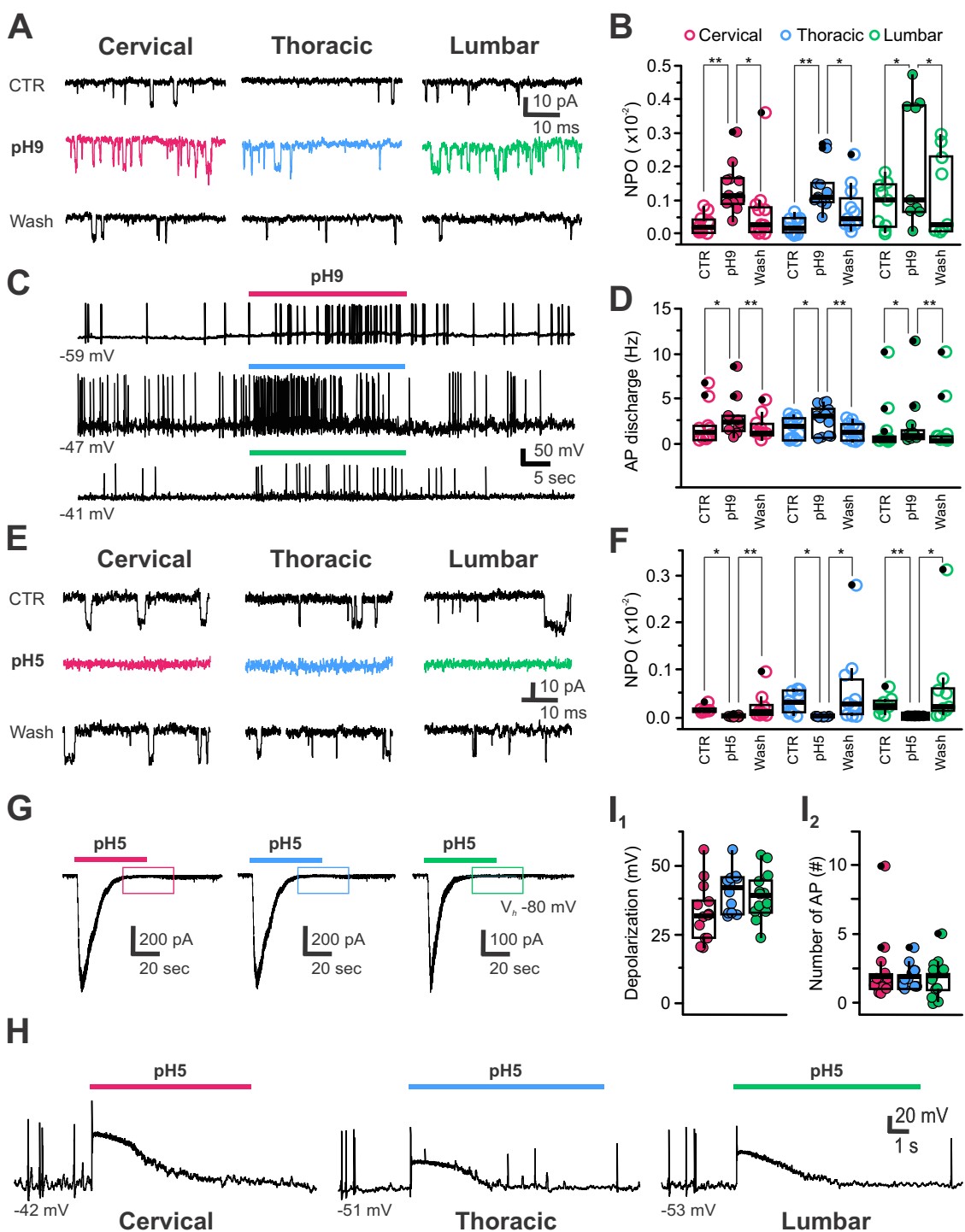

solution and intracellular solution B, Supplementary Tables 1 and 2) and recorded neurons at $V_h$ -80 mV with $V_{Step}$ from –40 to +30 mV ($\Delta V = +10$ mV, 100 ms). The elicited inward currents show a fast rise, an initial transient decrease followed by a persistent steady-state current. They also exhibit fast tail currents upon repolarization of the membrane potential (Fig. 7A). At $V_h$ -60 mV, the I-V curve revealed that currents activates at $V_{Step}$ larger than –40 mV, peaks at 0 mV (Fig. 7A, B), with an average current amplitude of –278 ± 168 pA and current density of –48 ± 34 pA.pF$^{-1}$ ($N = 17$, $n = 58$; Fig. 7C). T-CSF-cNs showed the largest $Ca^{2+}$ current peak amplitude while it was the smallest in C-CSF-cNs (Fig. 7C). Altogether the properties of the currents recorded agree with those mediated by $Ca_V$ and this observation was further confirmed by a

current block of 112 ± 10% from –46 ± 39 pA.pF$^{-1}$ in control to 4 ± 4 pA.pF$^{-1}$ in the presence of $Cd^{2+}$ (200 μM, pressure application; Fig. 7A, *Right*, violet traces and *7D*; $N = 17$, $n = 29$), a $Ca_V$ selective blocker. In juvenile rat[9] and mouse thoracic segment[19], CSF-cNs were suggested to also express LVA $Ca_V$ of the T-type that are inactivated for membrane potential around -60 mV. Using ramp protocols from $V_h$ -80 mV, revealed a characteristic 'shoulder' in all spinal CSF-cNs (average amplitude: -51 ± 32 pA, membrane potential: –29 ± 7 mV ($N = 19$, $n = 172$); Fig. 7E). This shoulder was not observed in recordings carried out from $V_h$ -60 mV, a membrane potential where T-type $Ca^{2+}$ ($Ca_V$ 3) are known to be inactivated (Fig. 7E). Note that the current traces illustrated in Fig. 7E have been normalized to the peak value for a better

**Fig. 4 | CSF-cNs share chemosensory properties through PKD2L1 channels and ASICs modulation. A** Representative recordings in voltage-clamp mode at $V_h$ -80 mV of PKD2L1 channel activity in CSF-cNs from the cervical, thoracic and lumbar regions in control, upon exposure to extracellular alkalinization (pH9; colored traces) and following washout (Wash). **B** Summary boxplots with whiskers for the average channel open probability (NPo) measured in control (CTR), in pH9 and after Wash for CSF-cNs for the three regions of interest (C: $N = 3$, $n = 12$; T: $N = 6$, $n = 11$ and L: $N = 4$, $n = 9$). *ANOVA.lme*: F = 4.52, df=8 and 87, p(F) = 0.0001272 and *Tukey (EMM) post hoc test* to compare CTR *vs.* ph9: $p = 0.0016$, 0.0020 and 0.0140 for C, T and L, respectively and pH9 *vs.* Wash: $p = 0.0140$ at L). **C** Representative recordings in current-clamp mode at RMP (values under the traces) of CSF-cN AP firing activity in control, during pH9 exposure (colored bars) from C, T and L segments (see color code). **D** Summary boxplots for the average instantaneous AP frequency in control (CTR), in pH9 and after Wash for CSF-cNs from the C, T and L segments (C: $N = 4$, $n = 14$; T: $N = 5$, $n = 13$ and L: $N = 3$, $n = 12$. *ANOVA.lme*: F = 18.52, df=8 and 96, p(F) = $2.2 \times 10^{16}$ and *Tukey (EMM) post hoc test* to compare CTR *vs.* pH9: $p = 0.0219$, 0.0098 and 0.036 and pH9 *vs.* Wash: $p = 0.0024$, 0.0001 and 0.0066 at C, T and L, respectively). **E** Representative recordings in voltage-clamp at $V_h$ -80 mV of PKD2L1 channel activity in CSF-cNs from the C, T and L regions in CTR, upon exposure to extracellular acidification (pH5; colored traces) and following washout (Wash). **F** Summary boxplots for the

average channel open probability (NPo) in control (CTR), in pH5 and after Wash for CSF-cNs from the C, T and L segments (C: $N = 3$, $n = 9$; T: $N = 4$, $n = 10$ and L: $N = 3$, $n = 9$). *ANOVA.lme*: F = 2.499, df=8 and 75, p(F) = 0.01835 and *Tukey (EMM) post hoc test* to compare CTR *vs.* pH5: $p = 0.0140$, 0.0417 and 0.0029 and pH5 *vs.* Wash: $p = 0.0012$, 0.0110 and 0.0065 at C, T and L, respectively). **G** Representative current traces elicited in CSF-cNs recorded in voltage-clamp mode at $V_h$ -80 mV in the C, T and L segments (see color code) upon pressure application of an acidic solution (pH5, colored bars). The colored boxes during the persistent phases of the current illustrates the absence of PKD2L1 activity (see also Panel 4**E**). **H** Representative voltage traces recorded in current-clamp mode at RMP (value under the traces) in C-, T- and L-CSF-cNs upon pressure application of the acidic solution (pH5, colored bars). **I** Summary boxplots for the average amplitude of the depolarization induced ($I_1$) and the number of APs triggered ($I_2$) following exposure to acidic pH in C-, T- and L-CSF-cNs (C: $N = 5$, $n = 13$; T: $N = 3$, $n = 12$ and L: $N = 3$, $n = 13$. $I_1$, *Kruskal-Wallis rank sum test*: $\chi^2 = 4.9908$, df=2, p($\chi^2$) = 0.08246 and *post hoc pairwise comparisons using Wilcoxon rank sum test* with continuity correction: C *vs.* T and C *vs.* L, $p = 0.12$ and T *vs.* L, $p = 0.81$). $I_2$, *Kruskal-Wallis rank sum test*: $\chi^2 = 0.074495$, df=2, p($\chi^2$) = 0.9634 and *post hoc pairwise comparisons using Wilcoxon rank sum test* with continuity correction: C *vs.* T; C *vs.* L and T *vs.* L, $p = 0.98$). In **B**, **D** and **F**, Data are given in the CTR, pH and Wash order and in B and F, data are multiplied by 100 on the graph for a better visualization.

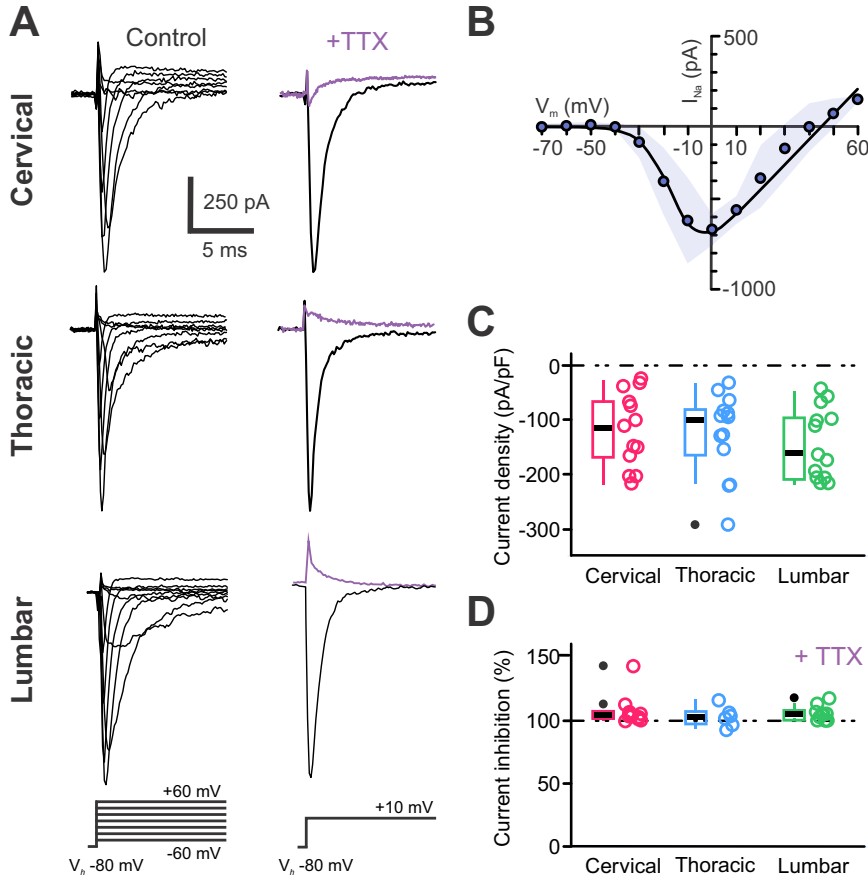

**Fig. 5 | CSF-cNs express functional TTX-sensitive sodium voltage-dependent channels. A** *Left*. Representative whole-cell current traces recorded in response to $V_{Step}$ from –60 mV to +60 mV (+10 mV increments for 50 ms) from $V_h$ -80 mV to elicit $Na^+$ current in a CSF-cN at the C, T and L levels. *Right*. $Na^+$ whole-cell currents elicited with a voltage step ($V_{Step}$) to +10 mV from $V_h$ -80 mV in control (black traces) and in the presence of 0.5 μM TTX (+ TTX, violet traces) recorded in the CSF-cNs illustrated Left. **B** Average current-voltage relationship (IV-curve) for the $Na^+$ currents recorded in CSF-cNs at all levels and generated from the peak current amplitude measured for each $V_{Step}$ and presented against the respective $V_{Step}$ (Data as mean ± SD, with SD represented as the blue shaded area). **C** Summary boxplots of the average amplitude of the $Na^+$ peak current density recorded in control at a $V_{Step}$ of +10 mV from $V_h$ -80 mV for CSF-cNs at each segment of interest (C: $N = 8$, $n = 23$; T: $N = 8$, $n = 13$ and L: $N = 4$, $n = 20$. *Kruskal-Wallis rank sum test*: $\chi^2$ = 2.2544, df=2, $p = 0.3239$; *post hoc pairwise comparisons using Wilcoxon rank sum test* with continuity correction: C *vs.* L, $p = 0.88$; T *vs.* C and T *vs.* L $p = 0.29$). **D** Summary boxplots of the averaged TTX inhibition (in percent) of the $Na^+$ current (C: $N = 4$, $n = 12$; T: $N = 4$, $n = 8$ and L: $N = 2$, $n = 6$). *Kruskal-Wallis rank sum test*: $\chi^2$ = 0.65025, df = 2, p($\chi^2$) = 0.7224 and *post hoc pairwise comparisons using Wilcoxon rank sum test* with continuity correction: C *vs.* T, $p = 0.83$; C *vs.* L, $p = 0.87$ and T *vs.* L, $p = 0.83$). In **C** and **D**, Single data points for all the recorded cells at each level are presented with colored opened circle (same color code as above).

visualization. These results demonstrate that spinal CSF-cNs express both HVA and LVA $Ca_V$, with T-CSF-cNs having higher $Ca^{2+}$ current density (and see also Supplementary Fig. 3D).

Spinal CSF-cNs express functional $Na_V$, along with $K_V$ of the delayed rectifier- ($I_{KD}$) and A-type ($I_A$), which present similarly expression densities along the cc axis and would be responsible for the AP generation. Additionally, all spinal CSF-cNs express both HVA and LVA $Ca_V$, with higher current densities in T-CSF-cNs. These findings are supported by a recent transcriptomic analysis of gene expression in spinal CSF-cNs[35] (see Supplementary Fig. 3B–D).

## CSF-cNs express ionotropic synaptic receptors with a density that differs along the cc

Mice medullary CSF-cNs express synaptic ionotropic receptors, including GABAergic, glycinergic, and AMPA/kainate glutamatergic receptors[12], mediating spontaneous and electrically evoked synaptic responses[18]. To assess the expression of these receptors in spinal CSF-cNs, we performed voltage-clamp recordings at $V_h$ -80 mV and applied selective agonists by pressure.

GABA (1 mM, 30 ms; Fig. 8A) and glycine (1 mM, 100 ms; Fig. 8C) elicited fast and large inward currents (with $E_{Cl} = +5$ mV, aCSF and

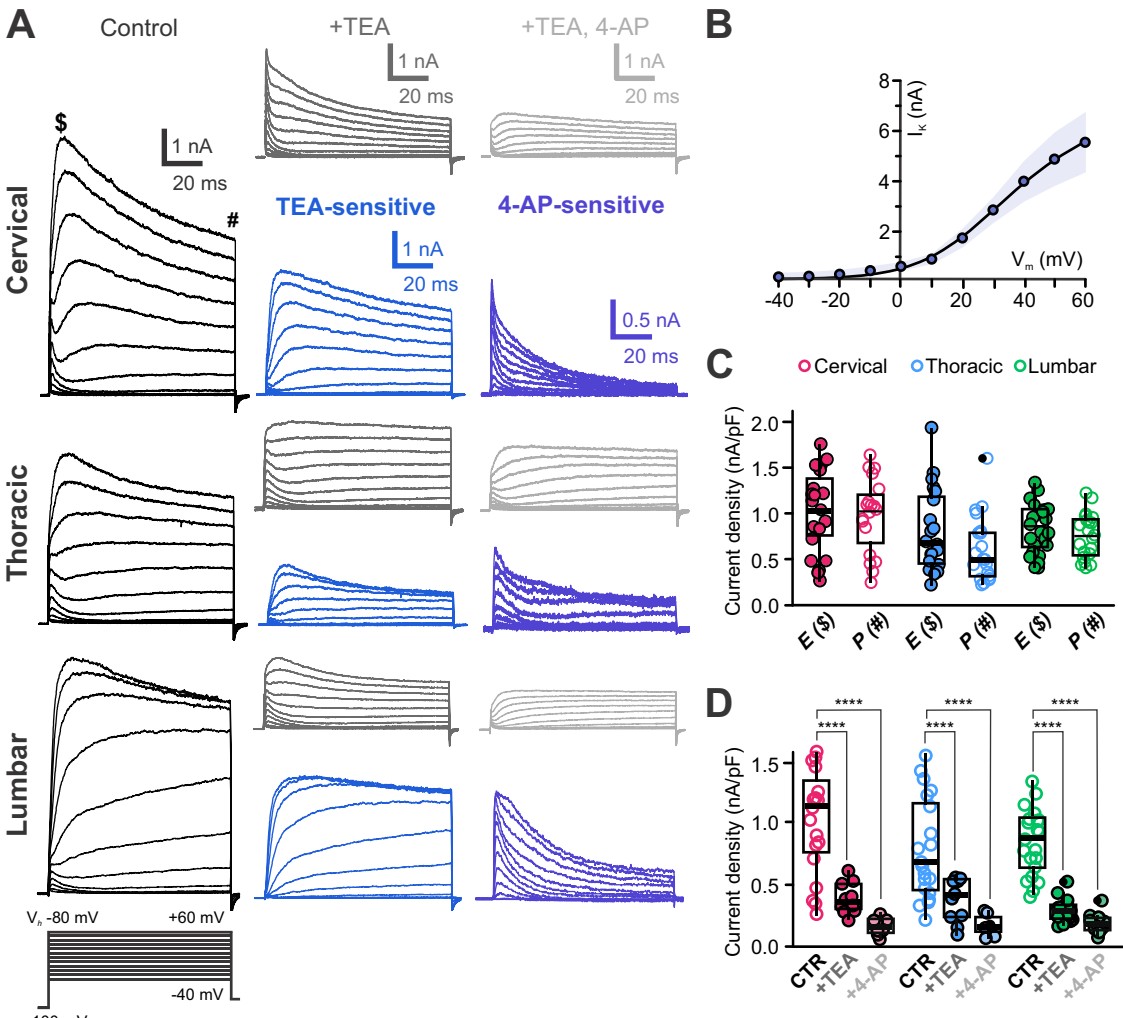

**Fig. 6 | Delayed rectifier and A-type voltage-dependent potassium channels are present in CSF-cNs. A** Representative whole-cell current traces recorded in response to $V_{Step}$ from –40 to +60 mV (increments of 10 mV, 100 ms) from $V_h$ -80 mV and with a pre-pulse at –100 mV in control (black traces), in the presence of TEA (grey traces) and in the presence of TEA and 4-AP (light gray traces) in C-, T- and L- CSF-cNs. For each level the TEA- (light blue traces) and 4-AP-senstive (dark blue traces) currents are presented (see Results for more details). Note that the early (E) and the persistent (P) current phases are indicated with $ and #, respectively. **B** Average IV-curve for the $K^+$ currents recorded in control in CSF-cNs from all levels (Data as mean ± SD, with SD represented as the blue shaded area). **C** Summary boxplots of the average amplitude of the peak $K^+$ current density recorded in control ($V_{Step}$ +60 mV from $V_h$ -80 mV) for the peak/early (E,$) and persistent (P, #) currents (C: $N = 4$, $n = 19$; T: $N = 4$, $n = 21$ and L: $N = 2$, $n = 20$); E and P in order;

*ANOVA.lme*: F = 3.35, df=5 and 114, p(F) = 0.007349 and *Tukey (EMM) post hoc test* to compare E *vs.* P within the different regions; ns, $p = 0.099$). **D** Summary boxplots of the average amplitude of the peak $K^+$ current density for CSF-cNs recorded ($V_{Step}$ +60 mV from $V_h$ -80 mV, prepulse at –100 mV) in the 3 regions of interest in control (CTR), in the presence of TEA (+ TEA) and of TEA + 4-AP (+ 4-AP) (C: $N = 4$, $n = 19$, 10, 10; T: $N = 5$, $n = 20$, 10, 7 and L: $N = 5$,$n = 21$, 19, 17; data given for CTR, +TEA and +4-AP, respectively. *ANOVA.lme*: F = 22.73, df = 8 and 124, p(F) < $2.2 \times 10^{16}$ and *Tukey (EMM) post hoc test* to compare amplitude within each region: C, $p < 0.0001$, <0.0001 and =0.1621; T, $p = 0.0003$, <0.0001 and =0.3023 and L, $p < 0.0001$, <0.0001 and =0.5291 for CTR *vs.* + TEA, CTR *vs.* +4-AP and +TEA *vs.* +4-AP, respectively. In **C** and **D**, Single data points for all the recorded cells at each level are presented with colored opened circle (same color code as above).

---

intracellular solution C, Supplementary Tables 1 and 2) in all CSF-cNs. The average current densities of –321 ± 36 pA.pF⁻¹ ($N = 6$, $n = 36$) and –529 ± 64 pA.pF⁻¹ ($N = 8$, $n = 32$) for GABA and glycine application, respectively (Fig. 8B, D). The currents were blocked in the presence of their selective antagonist, gabazine (Gbz, 10 µM; 97.6 ± 0.3%; $N = 6$, $n = 27$; Fig. 8A, violet traces) and strychnine (Stry, 1 µM; 95.5 ± 1.0%; $N = 9$, $n = 23$; Fig. 8C, violet traces), respectively and therefore mediated by $GABA_A$ and glycine receptor activation (Fig. 8B, C, violet traces) in agreement with the reports at the transcriptomic level (Supplementary Fig. 3E, F). When comparing the data along the cc axis, $GABA_A$-mediated currents were the highest in T-CSF-cNs compared to C- and L-CSF-cNs (Fig. 8A, B). In contrast, glycine-mediated currents were the smallest in C- compared to T- and L-CSF-cNs (Fig. 8C, D).

Next, medullary CSF-cNs express functional AMPA/kainate receptors[12,21], and our results show that pressure application of glutamate

(100 µM, 100–500 ms) induces inward currents in all spinal CSF-cNs with an average current amplitude and density of –22 ± 14 pA and –6 ± 5 pA.pF⁻¹, respectively and characterized by slow kinetics similar to those observed in medullary CSF-cNs (Fig. 9A₁). AMPA/kainate receptor activation was confirmed by the selective current inhibition in the presence of DNQX (400 µM; 91 ± 2% current inhibition; Fig. 9B). AMPA (100 µM, Fig. 9A₂) and kainate (100 µM, Fig. 9A₃) produced similar responses, while NMDA (100 µM) did not induce currents, suggesting the expression of AMPA/kainate but not NMDA receptors in spinal CSF-cNs. The analysis of the transcriptomic data supports these finding with genes for AMPA and kainate receptors subunits that are found in spinal CSF-cNs but not those for NMDA receptors (Supplementary Fig. 3G). These currents were present across all spinal segments but were significantly smaller in L-CSF-cNs (Fig. 9C) (aCSF and intracellular

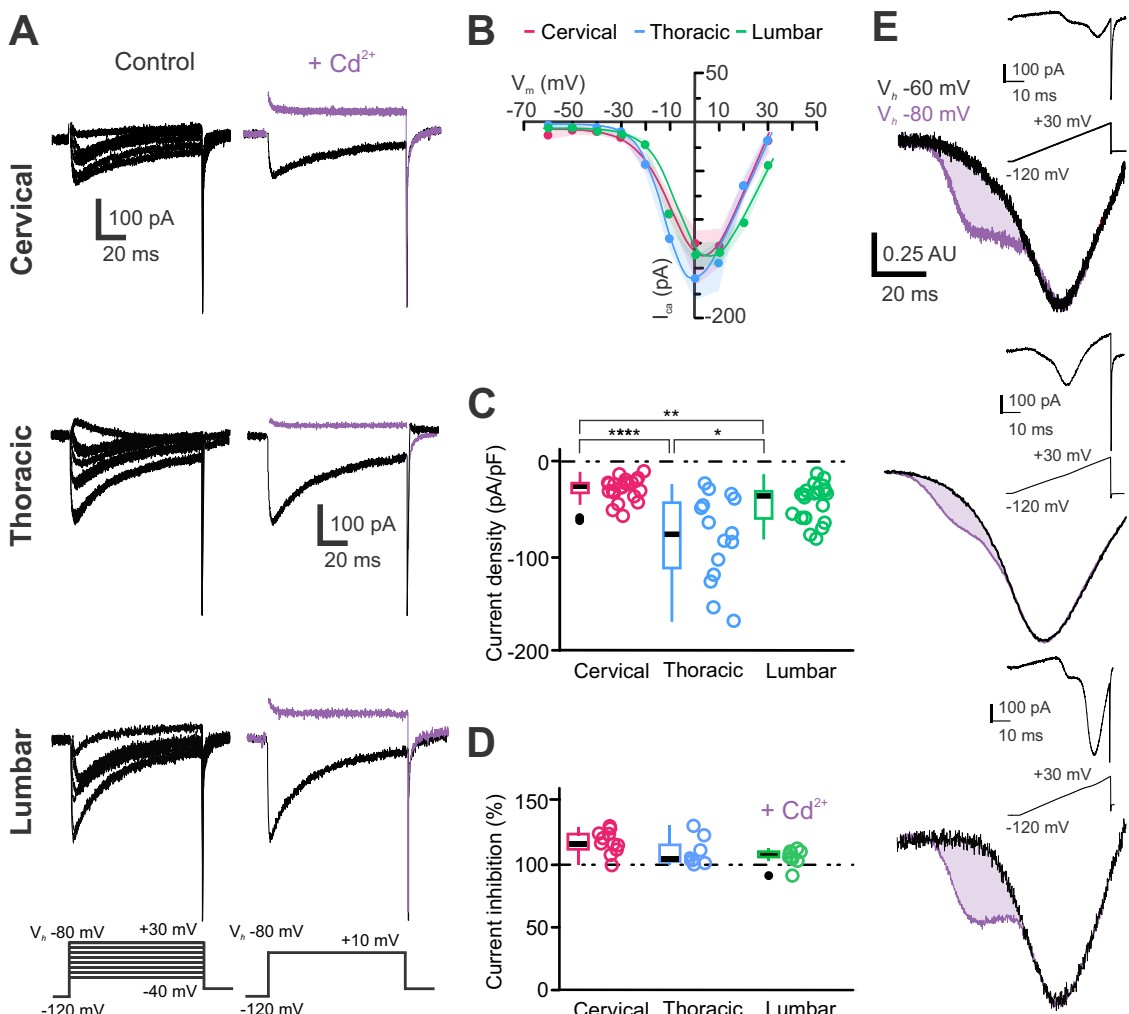

**Fig. 7 | CSF-cNs express high and low voltage-activated calcium channels with a higher density at the thoracic level. A** *Left*. Representative whole-cell current traces recorded in response to $V_{Step}$ from –40 mV to +30 mV ($V_{Step}$, 10 mV increments, 100 ms) from $V_h$ -80 mV with a prepulse to –120 mV to elicit $I_{Ca}$ in one CSF-cN at the C, T and L levels (*Top* to *Bottom*). *Right*. $Ca^{2+}$ whole-cell currents elicited at the peak with a $V_{Step}$ to +10 mV from $V_h$ - 80 mV in control (black traces) and in the presence of 200 μM $Cd^{2+}$ (+ $Cd^{2+}$, violet traces) recorded in the CSF-cNs illustrated *Left*. **B** Average IV-curves for the $I_{Ca}$ recorded from $V_h$ -60 mV in CSF-cNs from the C ($N = 10$, $n = 41$), T ($N = 10$, $n = 41$) and L ($N = 10$, $n = 41$) segments using a ramp protocol to elicit current against time and subsequently converted as current against voltage relationship (Data as mean ± SD, with SD represented with the colored shaded areas; color code as above). **C** Summary boxplots of the average amplitude of the $Ca^{2+}$ peak current density recorded in control at a $V_{Step}$ of +10 mV from $V_h$ -80 mV and measured at the peak in CSF-cNs from each segment of interest (C: $N = 2$, $n = 22$; T: $N = 5$, $n = 15$ and L: $N = 10$, $n = 21$). *Kruskal-Wallis rank sum test*: $χ^2 = 20.143$, df = 2, p($χ^2$) = $4.226 × 10^5$ and *post hoc pairwise comparisons using*

*Wilcoxon rank sum test* with continuity correction: C *vs.* T, $p = 5.6 × 10^{-5}$; C *vs.* L, $p = 0.0049$ and T *vs.* L, $p = 0.0110$). **D** Summary boxplots of the averaged $Cd^{2+}$ inhibition (in percent) of $I_{Ca}$ (C: $N = 2$, $n = 11$; T: $N = 5$, $n = 8$ and L-CSF-cNs: $N = 10$ $n = 10$). *Kruskal-Wallis rank sum test*: $χ^2 = 0.38571$, df = 2, p($χ^2$) = 0.8246 and *post hoc pairwise comparisons using Wilcoxon rank sum test* with continuity correction: C *vs.* T, C *vs.* L and T *vs.* L, $p = 0.97$, respectively). **E** Averaged $Ca^{2+}$ IV-Curves converted from the recordings obtained using a ramp protocol from $V_h$ -60 mV (black traces) and –80 mV (violet traces) for CSF-cNs at the C, T and L levels (*Top* to *Bottom*; see Methods for more details). The violet shaded area indicates the LVA (T-type channel) currents. Note that the data are normalized to the peak and presented in arbitrary units (AU) for a better comparison. Inset illustrates the activation protocol and an elicited representative recording for each region of interest obtained with a ramp protocol from $V_h$ -80 mV. In **C** and **D**, Single data points for all the recorded cells at each level are presented with colored opened circle (same color code as above).

solution C, Supplementary Tables 1 and 2). In current-clamp mode at –60 mV (DC current injection of –10 to –15 pA; (aCSF and intracellular solution A, Supplementary Tables 1 and 2), glutamate application caused significant membrane depolarization (+36 ± 6.5 mV, Fig. 9D, $E_1$; $N = 4$, $n = 21$). In agreement with the lower current density observed in L-CSF-cNs, glutamate application induced larger depolarizations in C- and T-CSF-cNs than L-CSF-cNs (Fig. 9$E_1$) and led to AP firing in C- and T-CSF-cNs but not in L-CSF-cNs (Fig. 9$E_2$). The glutamate-mediated effect was absent in the presence of DNQX (Fig. 9D, violet traces).

Finally, we tested for functional nicotinic cholinergic receptors (nACh-Rs; see also Corns and colleagues[40]) expression in spinal CSF-cNs. Acetylcholine application (ACh, 4 mM for 500 ms; aCSF and intracellular

solution D, Supplementary Tables 1 and 2) elicits currents in all CSF-cNs with an average amplitude of -9.5 ± 8.0 pA and current density of –1.2 ± 0.1 pA.pF⁻¹ ($N = 28$, $n = 65$; Fig. 10A, B *Left*). These responses were blocked by 94 ± 19% ($N = 28$, $n = 34$; Fig. 10A, violet traces and *10B Right*) in the presence of D-tubocurarine (D-Tubo, 100–200 μM). ACh-mediated currents were the smallest in T-CSF-cNs (Fig. 10A, B). This result indicates the functional expression of nACh-Rs in spinal CSF-cNs that is further supported at the transcriptomic level, where the corresponding genes of nACh-R subunits can be found (Supplementary Fig. 3H). In current-clamp recordings at -60 mV (DC current injection of -10 to -20 pA; aCSF and intracellular solution A, Supplementary Tables 1 and 2), ACh application depolarized the membrane potential by +23 ± 15 mV ($N = 10$, $n = 19$;

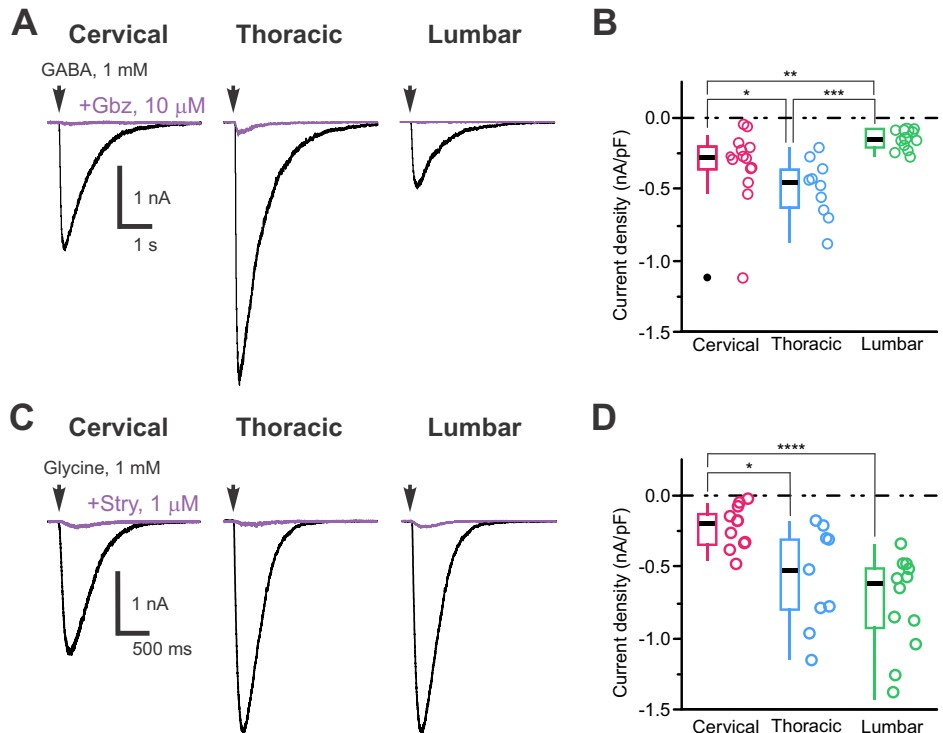

**Fig. 8 | CSF-cNs express differential GABA$_A$ and glycine receptors density along the central canal axis. A** Representative whole-cell current traces recorded at V$_h$ -80 mV in response to pressure application of GABA (1 mM, 30 ms, arrow) in control (black traces) and in the presence of gabazine (+ Gbz, 10 μM; violet traces), a selective GABA$_A$ receptor antagonist, in CSF-cNs from the C, T and L segments (*Left* to *Right*). **B** Summary boxplots with whiskers of the average amplitude of the GABA-mediated current densities recorded in control at V$_h$ -80 mV (C: $N$ = 2, $n$ = 14; T: $N$ = 2, $n$ = 10 and L: $N$ = 2, $n$ = 12). *Kruskal-Wallis rank sum test*: $\chi^2$ = 17.975, df = 2, p($\chi^2$) = 1.25 × 10$^4$ and *post hoc pairwise comparisons using Wilcoxon rank sum test with continuity correction*: C *vs.* T, $p$ = 0.02602; C *vs.* L, $p$ = 0.00306 and T *vs.* L, $p$ = 0.00011. **C** Representative whole-cell current traces recorded at V$_h$ -80 mV in

response to pressure application of glycine (1 mM, 30 ms, arrow) in control (black traces) and in the presence of strychnine ( + Stry,10 μM; violet traces), a selective glycine receptor antagonist, in CSF-cNs from the C, T and L segments (*Left* to *Right*). **D** Summary boxplots of the averaged amplitude of the glycine-mediated current densities recorded in control at V$_h$ -80 mV (C: $N$ = 2, $n$ = 11; T: $N$ = 9, $n$ = 10 and L: $N$ = 6, $n$ = 12). *Kruskal-Wallis rank sum test*: $\chi^2$ = 14.979, df = 2, p($\chi^2$) = 5.589 × 10$^4$ and *post hoc pairwise comparisons using Wilcoxon rank sum test with continuity correction*: C *vs.* T, $p$ = 0.03; C *vs.* L, $p$ = 5.3 × 10$^{-5}$ and T *vs.* L, $p$ = 0.25. In **B** and **D**, Single data points for all the recorded cells at each level are presented with colored opened circle (same color code as above).

Fig. 10C, D$_1$), triggering APs except in T-CSF-cNs, likely due to their lower nACh-R-mediated current (Fig. 10C, D$_2$). This depolarization was absent in the presence of D-Tubocurarine (Fig. 10C, *Bottom*).

In contrast to findings in juvenile rats[9], where CSF-cNs express functional P2X purinergic receptors and although the genes for these receptors were detected in CSF-cNs of mice (Supplementary Fig. 3I), we were unable to replicate these results in mouse spinal CSF-cNs, as no ATP-γ-S-mediated current was detected (Supplementary Fig. 3I and 4).

To summarize, our data show that all spinal CSF-cNs express functional GABA$_A$, glycine, AMPA/kainate glutamatergic, and nACh receptors but at different densities depending on the spinal cord segment considered. Activation of AMPA/kainate and nACh receptors modulates CSF-cN excitability and can trigger AP firing.

**Spinal CSF-cNs share metabotropic receptor expression modulating voltage-gated calcium channels**

Spinal CSF-cNs express functional ionotropic receptors for the major neurotransmitters (GABA, glutamate and ACh) and in the *medulla*, we indicated that they also express the GABA$_B$ metabotropic subtype capable of regulating Ca$_V$ activity[18]. Here, we investigated whether spinal CSF-cNs also express functional metabotropic receptors, and if these receptors modulate Ca$_V$ activity postsynaptically. To test this, we elicited I$_{Ca}$ at V$_h$ -80 mV by applying a V$_{Step}$ to +10 mV for 100 ms to record peak I$_{Ca}$ (Ca$_V$ modulation solution and intracellular solution B, Supplementary Tables 1 and 2). Before each step, a 50 ms hyperpolarizing step to −120 mV was used to fully de-inactivate all Ca$^{2+}$ channel

subtypes. This protocol was repeated every 20 seconds under control conditions before pressure-applying agonists for a given metabotropic receptor and subsequent wash.

We first assessed whether GABA$_B$ receptor (GABA$_B$-Rs) activation modulates Ca$_V$ in spinal CSF-cNs (Fig. 11). In control conditions, the peak I$_{Ca}$ had an average amplitude of −58 ± 29 pA.pF$^{-1}$ ($N$ = 8, $n$ = 29), which decreased to −38 ± 19 pA.pF$^{-1}$ ($N$ = 8, $n$ = 29) upon pressure application of baclofen (Bcl, 100 μM, for 40 s) to selectively activate GABA$_B$-Rs (Fig. 11A, C; *Kruskal-Wallis rank sum test*: $\chi^2$ = 8.2502, df = 2, p($\chi^2$) = 0.01616 and *post hoc pairwise comparisons using Wilcoxon rank sum test with continuity correction*: CTR *vs.* Bcl, $p$ = 0.014 and Bcl *vs.* Wash, $p$ = 0.093). The Bcl-mediated inhibition of I$_{Ca}$ over time is shown in Fig. 11B for CSF-cNs across the spinal regions. On average, Bcl inhibited I$_{Ca}$ by 34 ± 7% ($N$ = 8, $n$ = 29; Fig. 11D, *Left*; *Kruskal-Wallis rank sum test*: $\chi^2$ = 31.082, df=1, p($\chi^2$) = 2.473 × 10$^{-8}$ and *post hoc pairwise comparisons using Wilcoxon rank sum test with continuity correction*: CTR *vs.* Wash, $p$ = 5.7 × 10$^{-10}$). This inhibition was reversible, with currents recovering to −51 ± 27 pA.pF$^{-1}$ ($N$ = 8, $n$ = 29; CTR *vs.* Wash, $p$ = 0.345), reaching 88 ± 12% of control values. Although T-CSF-cNs had the largest I$_{Ca}$ amplitudes (Fig. 7C), GABA$_B$-R activation similarly inhibited currents across segments (Fig. 11B–D; *Kruskal-Wallis rank sum test*: $\chi^2$ = 1.0398, df=2, p($\chi^2$) = 0.5946 and *post hoc pairwise comparisons using Wilcoxon rank sum test with continuity correction*: C *vs.* T, C *vs.* L and T *vs.* L, $p$ = 0.62). In the presence of CGP (2 μM, a GABA$_B$-R antagonist; Fig. 11A, *Right* violet traces and B-D), I$_{Ca}$ amplitude was −55 ± 29 pA.pF$^{-1}$ in control ($N$ = 8, $n$ = 26), and Bcl application failed to inhibit I$_{Ca}$ (average amplitude: −55 ± 30 pA.pF$^{-1}$;

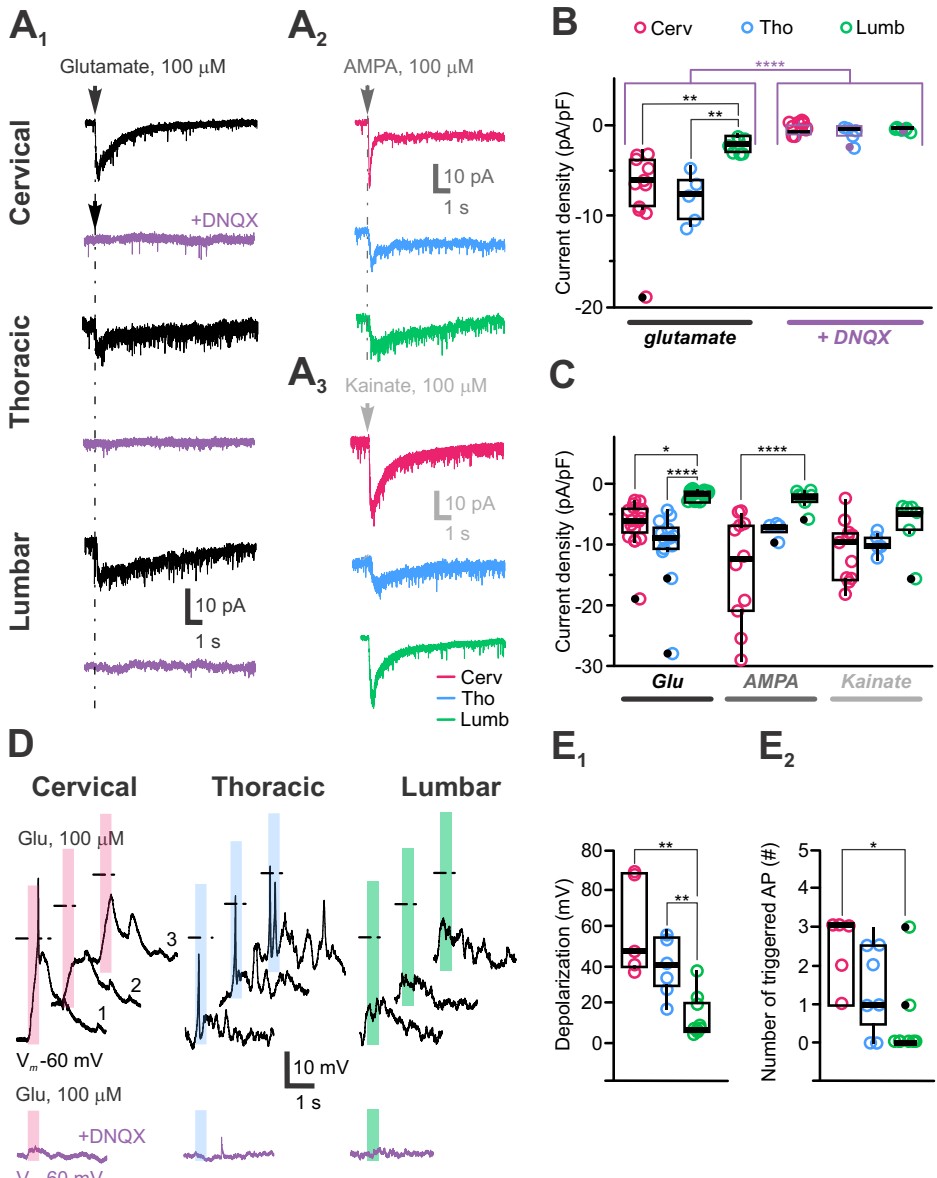

**Fig. 9 | CSF-cNs differentially express slow kinetics-low amplitude AMPA/kainate glutamatergic receptors that modulate their excitability. A** Representative whole-cell current traces recorded in CSF-cNs from the C, T and L segments (*Top* to *Bottom*) at $V_h$ -80 mV in response to pressure application of glutamate (100 μM, 100 ms and 500 ms for L; **A₁**), AMPA (100 μM, 30 ms; **A₂**) and kainate (100 μM, 30 ms; **A₃**). In **A₁**, the representative currents recorded in the presence of DNQX (100-400 μM; violet traces) upon glutamate applications are illustrated for each segment below the control traces. Arrows and dashed lines in A₁, A₂ and A₃ indicate time of agonist application. **B** Summary boxplots of the average amplitude of the glutamate-mediated currents recorded at $V_h$ -80 mV in control (*Left*) and in the presence of DNQX (*Right*) for the 3 segments of interest (C: $N = 2$, $n = 11$; T: $N = 2$, $n = 5$ and L: $N = 2$, $n = 7$). *ANOVA.lme*: F = 13.83, df=5 and 40, p(F) = $7.362 \times 10^{-6}$ and *Tukey (EMM) post hoc test* to compare glutamate-mediated currents densities between regions (C *vs.* T, $p = 0.8119$; C *vs.* L, $p = 0.005$ and T *vs.* L, $p = 0.0008$; black asterisk and bars) and the current densities recorded in control and in the presence of DNQX (CTR vs. DNQX, $p = 2.892 \times 10^{-8}$; violet bars and asterisks). **C** Summary boxplots of the averaged amplitude of the current densities recorded at $V_h$ -80 in CSF-cNs from the C, T and L segments in response to pressure application of glutamate (Glu), AMPA and kainate (C: $N = 3$, $n = 16$, 11 and 11; T: $N = 3$, $n = 13$, 4 and 5 and L: $N = 3$, $n = 17$, 7 and 6; Data and n are given for Glu-, AMPA- and -kainate-mediated current, respectively). *ANOVA.lme*: F = 7.907, df=8 and 81, p(F) = $8.36 \times 10^8$ and *Tukey (EMM) post hoc test* to compare Glu-, AMPA- and

kainate-mediated currents between the different regions; Glu: C *vs.* T, $p = 0.0938$; C *vs.* L, $p = 0.0170$ and T *vs.* L, $p < 0.0001$; AMPA: C *vs.* T, $p = 0.074$; C *vs.* L, $p < 0.0001$ and T *vs.* L, $p = 0.2032$; kainate: C *vs.* T, $p = 0.8941$; C *vs.* L, $p = 0.1516$ and T *vs.* L, $p = 0.4731$). **D** Representative traces of the membrane potential changes recorded at $V_m$ -60 mV in current-clamp mode in CSF-cNs from the C, T and L segments (*Left* to *Right*) in response to pressure application of glutamate (100 μM, 100 ms). Bottom traces in violet are membrane potential changes recorded upon glutamate application in the presence of DNQX. Shaded colored bars indicate the duration of the agonist application and dashed lines the 0 mV voltage. **E₁** Summary boxplots of the average depolarization levels of the membrane potential in from the C- ($N = 2$, $n = 5$), T- ($N = 3$, $n = 7$) and L-CSF-cNs ($N = 2$, $n = 9$) in response to pressure application of glutamate. *Kruskal-Wallis rank sum test*: $\chi^2 = 11.957$, d = 2, $p(\chi^2) = 2.533 \times 10^3$ and *post hoc pairwise comparisons using Wilcoxon rank sum test* with continuity correction: C *vs.* L, $p = 0.530$; C *vs.* T and T *vs.* L, $p = 0.005$). **E₂** Summary box-and-whiskers plots of the average number of AP triggered in CSF-cNs from C- ($N = 2$, $n = 5$), T- ($N = 3$, $n = 7$) and L-CSF-cNs ($N = 2$, $n = 9$) in response to pressure application of glutamate. *Kruskal-Wallis rank sum test*: $\chi^2 = 7.5111$, df=2, $p(\chi^2) = 0.02339$ and *post hoc pairwise comparisons using Wilcoxon rank sum test* with continuity correction: C *vs.* T, $p = 0.302$; C *vs.* L, $p = 0.034$ and T *vs.* L, $p = 0.117$). In **B, D** and **E**. Single data points for all the recorded cells at each level are presented with colored opened circle (same color code as above).

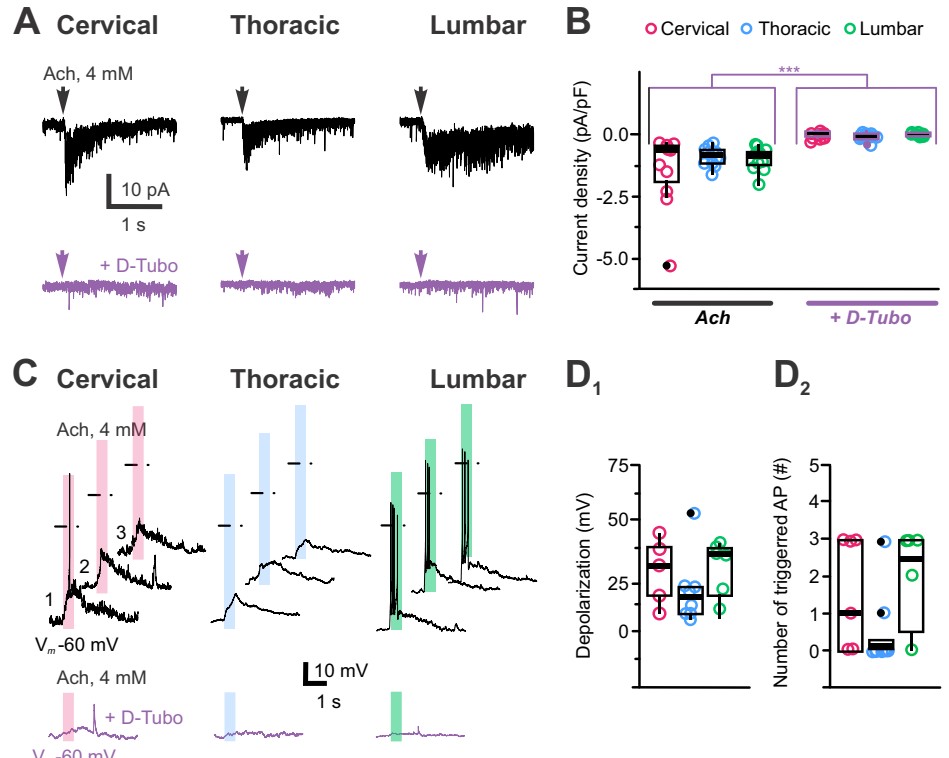

**Fig. 10 | Activation of nicotinic cholinergic receptors modulates CSF-cN excitability. A** Representative whole-cell current traces recorded in CSF-cNs from the C, T and L segments (*Left* to *Right*) at $V_h$ -80 mV in response to pressure application of acetylcholine (ACh, 4 mM for 500 ms). The representative currents recorded in the presence of D-tubocurarine (D-Tubo, 100–200 μM; violet traces) upon ACh applications are illustrated for each segment below the control traces. **B** Summary boxplots of the average amplitude of the ACh-mediated currents recorded at $V_h$ -80 mV in control (*Left*) and in the presence of D-Tubo (*Right*) for the 3 segments of interest (C: $N = 8$, $n = 11$; T: $N = 13$, $n = 12$ and L-CSF-cNs: $N = 7$, $n = 11$). *ANO-VA.lme*: F = 9.192, df=5 and 62, p(F) = $1.39 \times 10^{-6}$ and *Tukey (EMM) post hoc test* to compare ACh-mediated current densities between regions (C *vs.* T, p = 0.1289; C *vs.* L, p = 0.2404 and T *vs.* L, p = 0.9512; black asterisk and bars) and the current densities recorded in control and in the presence of D-Tubo (CTR vs. D-Tubo, p = $1.991 \times 10^{-8}$; violet bars and asterisks). **C** Representative traces of the membrane potential changes recorded at $V_m$ -60 mV in current-clamp mode in C-, T- and L-

CSF-cNs (*Left* to *Right*) in response to pressure application of ACh (4 mM, 500 ms). Bottom traces in violet are membrane potential changes recorded upon ACh application in the presence of D-Tubo. Shaded colored bars indicates the duration of the agonist application and dashed lines the 0 mV voltage. **D₁** Summary boxplots of the average depolarization levels of the membrane potential in C- ($N = 4$, $n = 5$), T- ($N = 3$, $n = 7$) and L-CSF-cNs ($N = 4$, $n = 9$) in response to pressure application of ACh. *Kruskal-Wallis rank sum test*: $\chi^2 = 1.3411$, d = 2, p($\chi^2$) = 0.5114 and *post hoc pairwise comparisons using Wilcoxon rank sum test* with continuity correction: C *vs.* T, p = 1.0; C *vs.* L, p = 0.62 and T *vs.* L, p = 0.62). **D₂** Summary boxplots of the average number of AP triggered in CSF-cNs from the C ($N = 4$, $n = 5$), T ($N = 3$, $n = 7$) and L ($N = 4$, $n = 9$) segments in response to pressure application of ACh. *Kruskal-Wallis rank sum test*: $\chi^2 = 3.1265$, df=2, p($\chi^2$) = 0.2095 and *post hoc pairwise comparisons using Wilcoxon rank sum test* with continuity correction: C *vs.* T, p = 0.36; C *vs.* L, p = 0.77 and T *vs.* L, p = 0.33). In **B** and **D**, Single data points for all the recorded cells at each level are presented with colored opened circle (same color code as above).

---

inhibition by 0.1 ± 7.9%, $N = 8$, $n = 26$; Fig. 11D, *Right; Kruskal-Wallis rank sum test*: $\chi^2 = 0.9057$ df = 2, p($\chi^2$) = 0.3413 and *post hoc pairwise comparisons using Wilcoxon rank sum test* with continuity correction: CTR *vs.* Bcl and Bcl *vs.* Wash, p = 0.35). In Fig. 11C, for a better visualization of the Bcl-mediated effect alone or in the presence of CGP, the data across individual cells in control, Bcl, and Wash are presented as values normalized against the control mean value of the population. On average the currents recorded were 65 ± 30% and 100 ± 39% of the control current amplitude upon application of Bcl alone or in the presence of CGP, respectively ($N = 8$, $n = 29, 26$; Fig. 11C).

We next tested the effect of muscarinic cholinergic receptor (mACh-Rs) activation on $I_{Ca}$ modulation. Using the same protocol as for GABA$_B$-R activation, Oxotremorine-M (Oxo-M, 100 μM, selective mACh-R agonist) was pressure-applied while recording Ca²⁺ peak currents in C-, T-, and L-CSF-cNs (Fig. 12). The average $I_{Ca}$ peak amplitude was –53 ± 39 pA.pF⁻¹ in control and decreased to –33 ± 28 pA.pF⁻¹ after Oxo-M application ($N = 9$, $n = 36$; Fig. 12A–C; *Kruskal-Wallis rank sum test*: $\chi^2 = 12.064$, df=2, p($\chi^2$) = 0.0024 and *post hoc pairwise comparisons using Wilcoxon rank sum test* with continuity correction: CTR *vs.* Oxo-M, p = 0.0051 and Oxo-M *vs.* Wash, p = 0.00454). On average, Oxo-M inhibited $I_{Ca}$ by 39 ± 21% ($N = 9$, $n = 36$; Fig. 12D, *Left; Kruskal-Wallis rank sum test*: $\chi^2 = 34.694$, df = 1,

p($\chi^2$) = $3.8 \times 10^{-9}$ and *post hoc pairwise comparisons using Wilcoxon rank sum test* with continuity correction: CTR *vs.* Oxo-M, p = $1.1 \times 10^{-10}$). This inhibition was reversible, with $I_{Ca}$ currents recovering to –52 ± 40 pA.pF⁻¹ ($N = 9$, $n = 36$; CTR *vs.* Wash, p = 0.7412). The $I_{Ca}$ modulation by Oxo-M was consistent across the cc axis (Fig. 12B–D; *Kruskal-Wallis rank sum test*: $\chi^2 = 5.5486$, df = 2, p($\chi^2$) = 0.06239 and *post hoc pairwise comparisons using Wilcoxon rank sum test* with continuity correction: p = 0.70, 0.12 and 0.11 for C *vs.* T, C *vs.* L and T *vs.* L, respectively). In the presence of atropine (Atr, 10 μM, a selective mACh-R antagonist; Fig. 12A, *Right* violet traces *and B-D*), $I_{Ca}$ amplitude was –38 ± 30 pA.pF⁻¹ ($N = 9$, $n = 21$), and Oxo-M failed to inhibit $I_{Ca}$ (amplitude of –36 ± 28 pA.pF⁻¹ and inhibition by 3 ± 11%; $N = 9$, $n = 21$; Fig. 12D, *Right; Kruskal-Wallis rank sum test*: $\chi^2 = 0.24063$, df = 1, p($\chi^2$) = 0.6238 and *post hoc pairwise comparisons using Wilcoxon rank sum test* with continuity correction: Oxo-M *vs.* Wash, p = 0.64) confirming mACh-R-mediated $I_{Ca}$ inhibition. In Fig. 12C, for a better visualization of $I_{Ca}$ inhibition by Oxo-M values across individual cells were normalized against the population control mean value (see above). On average the currents recorded were 62 ± 40% and 96 ± 57% of the control current amplitude upon application of Oxo-M alone or in the presence of Atropine, respectively ($N = 9$, $n = 29$ and 21; Fig. 12C). The comparison of the Bcl and Oxo-M effect on $I_{Ca}$ indicates the GABA$_B$- and mACh-R

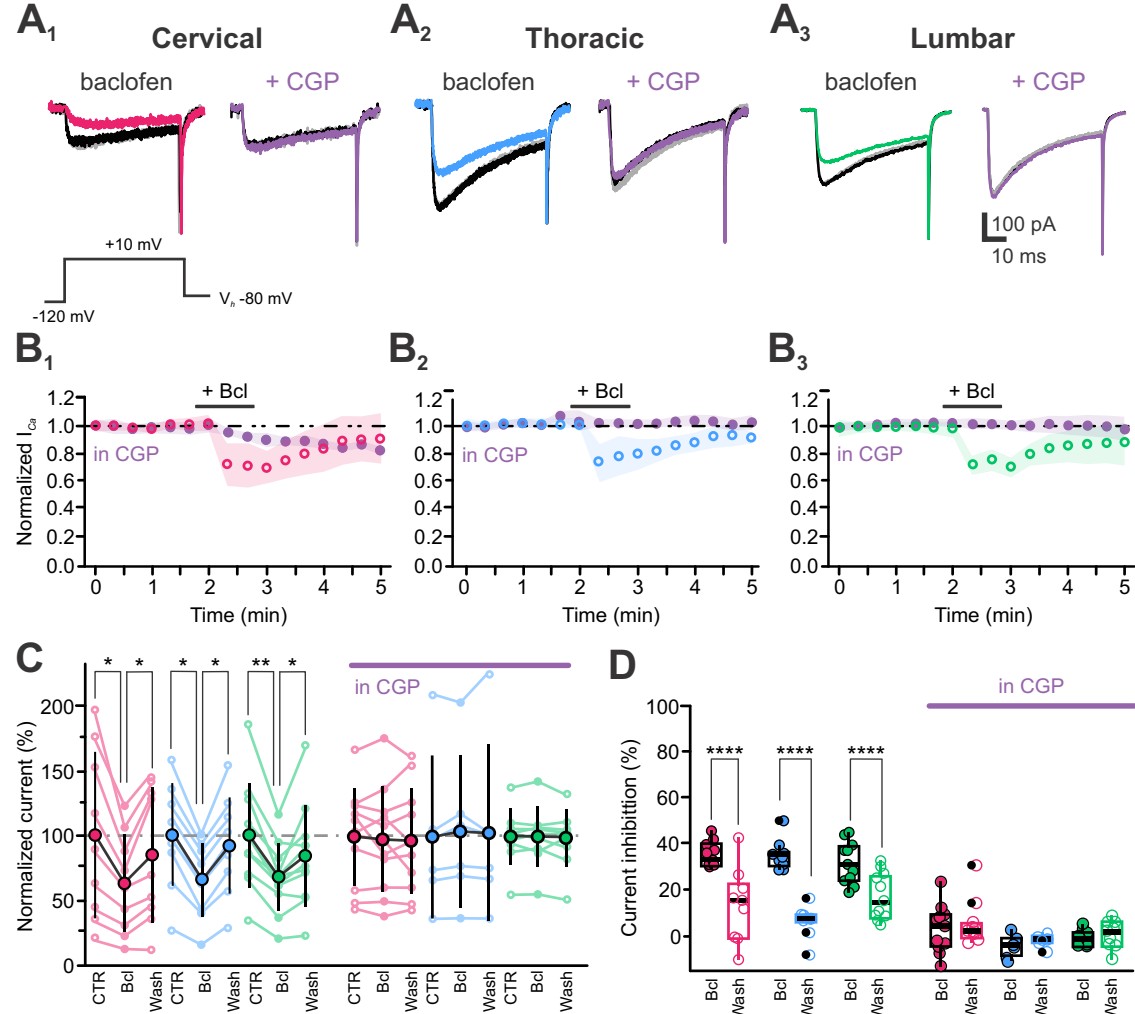

**Fig. 11 | Metabotropic GABA receptors are functional in CSF-cNs and inhibit calcium channels. A** Representative $I_{Ca}$ traces elicited at the current peak with a voltage step to +10 mV from $V_h$ -80 mV and recorded in CSF-cNs from the C ($A_1$), T ($A_2$) and L segments ($A_3$) in control (black traces), in response to baclofen alone (baclofen; *Left*, colored traces) or in the presence of CGP (+ CGP, 2 μM; *Right*, violet traces) and after agonist Wash (gray traces). **B** Average time courses of normalized $I_{Ca}$ peak amplitude recorded in CSF-cNs for the regions of interest ($B_1$, C: $N$ = 2, 3; $n$ = 9, 11; $B_2$, T: $N$ = 3, $n$ = 9, 6 and $B_3$, L: $N$ = 3, $n$ = 11, 9) in control and in the presence of CGP. Baclofen was applied at 100 μM by pressure for 40 s (+ Bcl, black bar). **C** Summary plots for the normalized peak $I_{Ca}$ density (mean ± SD) before (CTR), during (Bcl) and after agonist application (Wash) in the absence (*Left*) or presence of CGP (*Right*, in CGP, violet bar) for the regions of interest (Bcl alone, C: $N$ = 2, $n$ = 9; T: $N$ = 3, n = 9 and L: $N$ = 3, $n$ = 11. Bcl in CGP: C: $N$ = 3, $n$ = 11; T: $N$ = 3, $n$ = 6 and L: $N$ = 3, $n$ = 9). *Friedman test* for Bcl alone: χ²(Bcl)=14.2, 16.2 and 18.7, df = 2, p(χ²) = $8.16 \times 10^{-5}$, $3.00 \times 10^{-5}$ and $8.58 \times 10^{-6}$ for C, T and L segments and *post hoc Pairwise comparisons using Wilcoxon rank sum test* with Benjamini-Hochberg adjustment to compare currents densities within Regions for CTR *vs.* Bcl and Bcl *vs.* Wash: $p$ = 0.012, 0.006 and 0.001/0.01. F*riedman test* for Bcl in CGP: χ²(CGP) = 1.27, 5.33 and 2.00, df = 2, p(χ²) = 0.529, 0.069 and 0.368 for C, T and L segments and *post hoc pairwise comparisons using Wilcoxon rank sum test* with

Benjamini-Hochberg adjustment to compare currents densities within regions for CTR *vs.* Bcl and Bcl *vs.* Wash: $p$ = 0.898, 0.328/0.438 and 0.652 for C, T, and L. **D** Summary boxplots for the $I_{Ca}$ inhibition induced during (Bcl) and after (Wash) agonist application and in the absence (*Left*) and presence of CGP (*Right*; in CGP, violet bar) for the 3 regions of interest (Bcl alone, C: $N$ = 2, $n$ = 9; T: $N$ = 3, $n$ = 9 and L: $N$ = 3, $n$ = 11. Bcl in CGP, C: $N$ = 3, $n$ = 1; T: $N$ = 3, $n$ = 6 and L: $N$ = 3, $n$ = 9. *Kruskal-Wallis rank sum test*: χ²(Bcl)=31.082, df=1, p(χ²) = $2.473 \times 10^{-8}$ and *post hoc pairwise comparisons using Wilcoxon rank sum test* with continuity correction: Bcl vs. Wash: $p$ = $5.7 \times 10^{-10}$; χ²(Region)=1.0398, df=2, p(χ²) = 0.5946 and *post hoc pairwise comparisons using Wilcoxon rank sum test* with continuity correction: $p$ = 0.62 for C *vs.* L, C *vs.* T and T *vs.* L, respectively. *Kruskal-Wallis rank sum test*: χ²(CGP) = 0.90566, df = 1, p(χ²) = 0.3413 and *post hoc pairwise comparisons using Wilcoxon rank sum test* with continuity correction: Bcl in CGP vs. Wash: $p$ = 0.35; χ²(Region)=5.742, df = 2, p(χ²) = 0.05664 and *post hoc pairwise comparisons using Wilcoxon rank sum test* with continuity correction: $p$ = 0.411for C *vs.* L and $p$ = 0.072 for C *vs.* T and T *vs.* L, respectively. In **B**: colored open circles: in control; violet filled circles: in the presence of CGP. mean ± SD; with SD represented as colored shaded area. In **C** large filled colored circles: mean ± SD; small open and filled colored circles: single data points for all the recorded cells at each level. Data are given in the CTR, Bcl and Wash order. In **D**, Data are given in Bcl and Wash order.

activation inhibits Ca$_V$ to the same extent across segments ($I_{Ca}$ inhibition by Bcl: 34 ± 7% and by Oxo-M: 39 ± 21%; *Kruskal-Wallis rank sum test*: χ² = 0.25855, df = 1, p(χ²) = 0.6111 and *post hoc pairwise comparisons using Wilcoxon rank sum test* with continuity correction: Bcl *vs.* Oxo-M, $p$ = 0.62).

We finally tested $I_{Ca}$ modulation *via* metabotropic glutamatergic receptors (mGlu-Rs) by applying glutamate (100 μM, 40 s) but did not observe any reduction in current amplitude across the spinal cord segments ($I_{Ca}$ amplitude –78 ± 41, –78 ± 40 and –78 ± 40 pA.pF$^{-1}$ in CTR,

Glu, and Wash, respectively; *Kruskal-Wallis rank sum test*: χ² = $2.2831 \times 10^{-4}$, df = 2, p(χ²) = 0.9989 and *post hoc pairwise comparisons using Wilcoxon rank sum test* with continuity correction: CTR *vs.* Glu, Glu *vs.* Wash and Wash *vs.* CTR, $p$ = 0.99) indicating that Ca$_V$ in CSF-cNs appear not modulated by mGlu-R activation and see Supplementary Fig. 3G).

In summary, our results show that all tested spinal CSF-cNs express functional GABA$_B$- and mAch-Rs (and see Supplementary Fig. 3E and H),

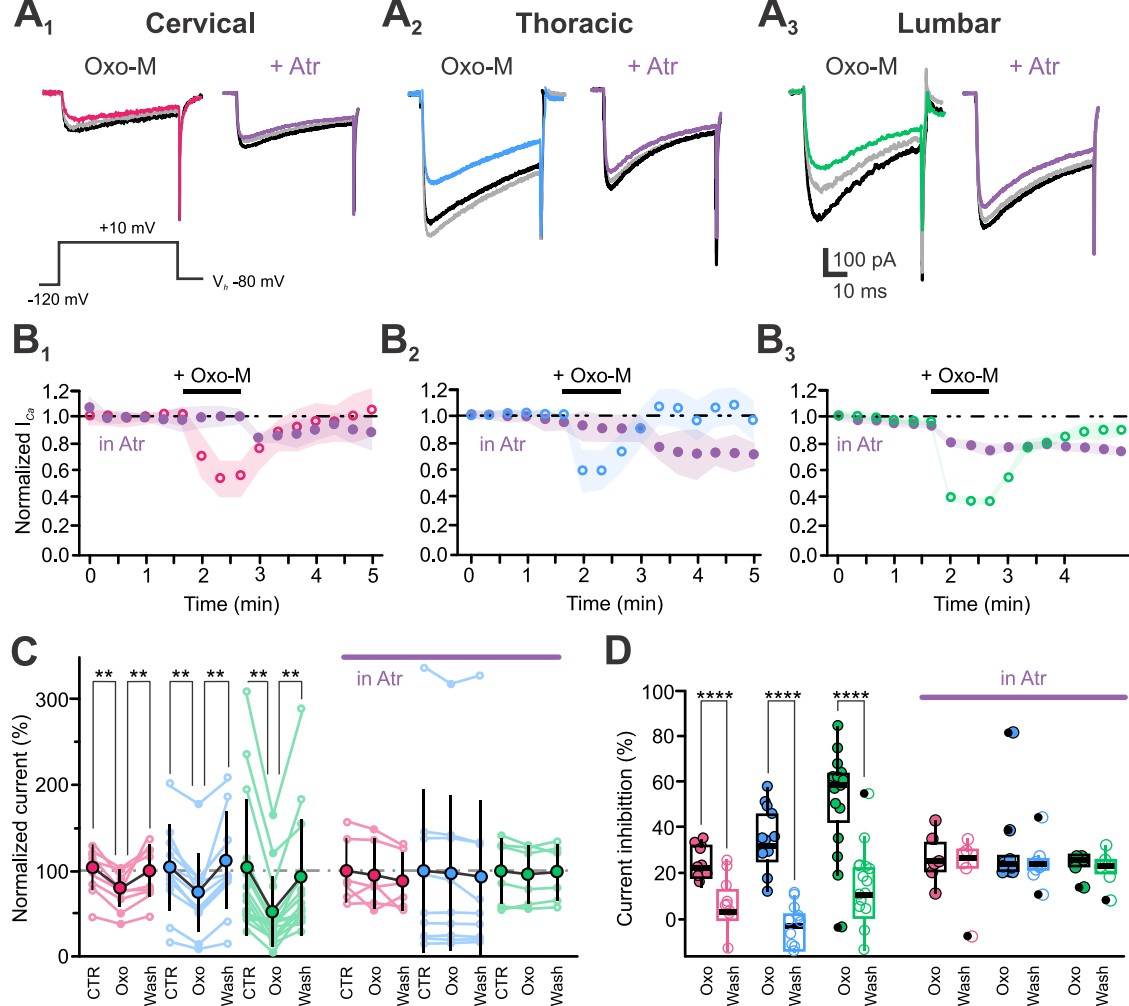

**Fig. 12 | In CSF-cNs muscarinic receptor activation reversibly inhibits calcium channels. A** Representative $I_{Ca}$ traces elicited at the current peak with a voltage step to +10 mV from $V_h$ -80 mV and recorded in CSF-cNs from the C ($A_1$), T ($A_2$) and L segments ($A_3$) in control (black traces), in response to Oxo-M alone (baclofen; *Left*, colored traces) or in the presence of atropine (+ Atr, 10 μM; *Right*, violet traces) and after agonist Wash (grey traces). **B** Average time courses of normalized $I_{Ca}$ peak amplitude recorded in CSF-cNs for the regions of interest ($B_1$, C: N = 2, 3, n = 9, 6; $B_2$, T: N = 3, n = 11, 9 and $B_3$, L: N = 3 n = 16, 6) in control and in the presence of Atr. Oxo-M was applied at 100 μM by pressure for 40 s (+Oxo-M, black bar).
**C** Summary plots for the normalized peak $I_{Ca}$ density (mean ± SD) before (CTR), during Oxo-M application and after agonist washout (Wash) in the absence (*Left*) or presence of atropine (*Right*, in Atr, violet bar) for the regions of interest. Single data points across cells were normalized to the mean value in control and expressed in percent (Oxo-M alone, C: N = 2, n = 9; T: N = 3, n = 11 and L: N = 3, n = 16. Oxo-M in Atr, C: N = 2, n = 6 and 93 ± 108% (T: N = 3, n = 9 and L: N = 3, n = 6). *Friedman test* for Oxo-M alone: χ²(Oxo)=14.2, 16.2 and 18.7, df = 2, p(χ²) = 8.16 × 10⁻⁵, 3 × 10⁻⁵ and 8.58 × 10⁻⁶ for C, T and L segments and *post hoc pairwise comparisons using Wilcoxon rank sum test* with Benjamini-Hochberg adjustment to compare currents densities within Regions for CTR *vs.* Oxo and Oxo *vs.* Wash: p = 0.012,

0.006 and 0.001/0.01. *Friedman test* for Oxo-M in Atr: χ²(Atr)=, df = 2, p(χ²) = 0.0694, 0.0970 and 0.115 for C, T and L segments and *post hoc pairwise comparisons using Wilcoxon rank sum test* with Benjamini-Hochberg adjustment to compare currents densities within Regions for CTR *vs.* Oxo and Oxo *vs.* Wash: p = 0.438, 0.246/0.301 and 0.468/0.657 for C, T, and L. **D** Summary boxplots for the $I_{Ca}$ inhibition induced during (Oxo-M) and after (Wash) agonist application and in the absence (*Left*) and presence (*Right*; in Atr, violet bar) of atropine for the 3 regions of interest (Oxo-M alone, C: N = 2, n = 9;T: N = 3, n = 11 and L:N = 3, n = 116. Oxo-M in Atr, C: N = 2, n = 6; T: N = 3, n = 9 and L: N = 3, n = 6). *Kruskal-Wallis rank sum test:* χ²(Oxo) = 34.694, df = 1, p(χ²) = 3.858 × 10⁻⁹ and *post hoc pairwise comparisons using Wilcoxon rank sum test* with continuity correction: Oxo *vs.* Wash: p = 1.1 × 10⁻¹⁰; χ²(Region) = 5.5486, df = 2, p(χ²) = 0.06239 and *post hoc pairwise comparisons using Wilcoxon rank sum test* with continuity correction: p = 0.70, 0.12 and 0.11 s for C *vs.* T, C *vs.* L and T vs. L, respectively. In **B**: colored open circles: in control; violet filled circles: in the presence of Atr. mean ± SD; with SD represented as colored shaded area. In **C** large filled colored circles: mean ± SD; small open and filled colored circles: single data points for all the recorded cells at each level. Data are given in the CTR, Oxo and Wash order. In **D** Data are given in Oxo and Wash order.

whose activation inhibits post-synaptic $Ca_V$ to a similar level in all spinal CSF-cNs. Notably, we did not observe any modulation by mGlu-Rs, which have been shown to inhibit $Ca^{2+}$ currents in other neuronal populations.

## Discussion
CSF-cNs have been extensively studied in zebrafish larvae[5,16,17,26,27,32,33], in lamprey[6,22,23], and, to some extent, in mice[7,8,12,13,18,19,21,41], rats[9,11] and non-human primates (NHP)[4,5,10]. However, to date CSF-cN properties were largely under-characterized with data reported for different animal models, ages or different SC segments, but not systematically.

Here, we provide an in-depth characterization of CSF-cN properties along the mouse SC from cervical to lumbar segments and find that CSF-cNs possess a conserved and uniform morphology across species and mouse SC segments, respectively. We indicate that they express, along classical ionotropic synaptic receptors, $Na_V$, $K_V$ and $Ca_V$ (LVA and HVA types), the latter being modulated by metabotropic GABAergic and muscarinic receptors but not by glutamatergic ones. We further report electro-physiological differences between CSF-cNs from different SC segments. Finally, although, they share similar chemosensory properties, our study indicates that they exhibit region-specific firing patterns (tonic *vs.* single-

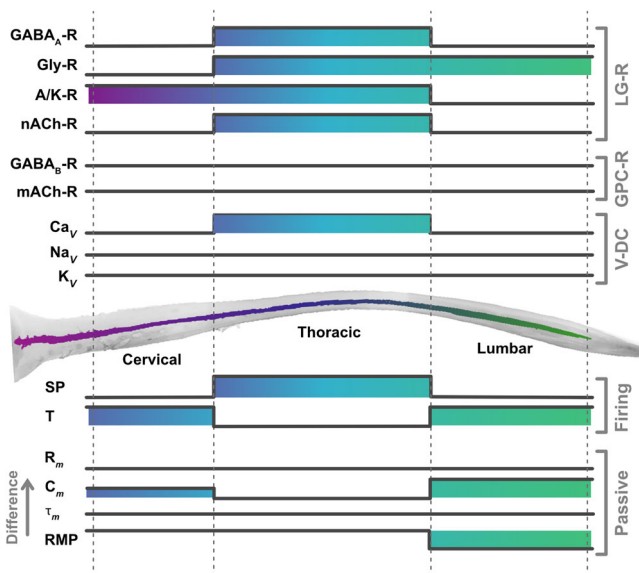

**Fig. 13 | Summary diagram illustrating the electrophysiological differences for CSF-cNs recorded along the spinal cord rostro-caudal axis.** GABA$_A$-, Gly-R, A/K- and nACh-R: GABAergic, glycinergic, AMPA/kainate and nicotinic cholinergic receptors. GABA$_B$- and mACh-R: GABAergic and muscarinic cholinergic metabotropic receptors. Ca$_V$, Na$_V$ and K$_V$: Calcium, Sodium and Potassium voltage-dependent channels. SP and T: single-spike and tonic firing; R$_m$, C$_m$, $\iota_m$ and RMP: membrane resistance, capacitance, time constance and resting potential; LG-R: ligand-gated receptors, GPC-R: G-protein couple receptors; V-DC: voltage-dependent channels. The up- or downward boxes represent differences in the illustrated parameters.

spike) as well as differential calcium signaling and synaptic receptor-mediated excitatory *vs.* inhibitory drive (Fig. 13).

Our results provide a comprehensive and systematic characterization of CSF-cN morphological and electrophysiological properties along the mouse SC axis. They suggest that because of their shared sensory properties, CSF-cNs would detect and integrate information along the cc in a synchronized manner. While, due to the regional differences along the spinal levels of their electrophysiological properties, they would be differentially regulated by local partners to serve region-specific modulation of the physiological functions controlled by the spinal network they are inserted in. Overall, our data set ground for future studies to address this crucial question and demonstrate CSF-cN function as a novel sensory system in mammalian CNS.

## Spinal CSF-cNs form a dense morphologically homogenous neuronal network

Our study confirms and extends prior reports[7,8,13,19], showing that in mice, CSF-cNs are distributed around the central canal (cc) along the entire rostro-caudal axis of the SC, exhibiting a characteristic and conserved morphology[5,7–9,11,31,41]. We show that CSF-cNs form a dense interconnected neuronal population, with approximately 10 to 20 cells per 10 μm of tissue depth across the entire cc axis. Considering a SC length of ~2.5 cm (25,000 μm) in 3-weeks-old mice and a CSF-cN density of ~14 cells/10 μm, based on our confocal images. We estimate that ~35,000 CSF-cNs would be present in the whole SC (52,500 for a 3.75 cm SC in 6-week-old mice). Bjugn and Gundersen[42] estimated that ~6.4 million neurons are found in the whole SC gray matter and therefore spinal CSF-cNs would only represent 0.9% of the total SC neuronal population.

CSF-cNs have been observed in several vertebrate species, where they show similar morphology as well as distribution around and along the cc. In lamprey[6], zebrafish[5,16] and turtle[36], they mainly exhibit a triangular, small soma inserted in the ependymal cell layer with a short dendrite projecting to the cc (but see below for lamprey). Similar features are found for CSF-cNs in postnatal rodents[8,9,13]. With aging, their morphology and localization

changes, and one can find either intra-ependymal neurons, embedded within the ependymal cell layer, or subependymal, located below the cc, with a longer dendrite extending into the cc lumen[7,11,13,29]. In mice, spinal CSF-cNs are predominantly found in the ventral region, comprising about 60% of the total. This ventral localization contrasts with the lateral distribution observed in the medulla[7] (but see also Kútna and colleagues[41]) and suggest an anatomical reorganization along the medullo-spinal axis. We also confirm the presence of PKD2L1-expressing neurons (tdTomato$^+$) in more distal ventral locations along the SC[13]. Note that Tonelli Gombalová and colleagues indicated that in C57 Black6/N mice, in contrast to the J substrain, an important proportion of CSF-cNs is observed in ectopic positions away from the cc due to potential dysfunction of Crb1 and Cyfip2 products in this substrain[43]; our present and past studies were conducted on the C57 Black6/J substrain.

CSF-cNs are also observed in NHP[4,5,10,37] and a recent study indicates that CSF-cNs morphology, density, and distribution[10] are similar to that observed in rodents with PKD2L1$^+$ neurons that are also localized in the ventral region of the SC away from the cc[10]. Interestingly in macaques, CSF-cNs are largely present in subependymal position, exhibit long dendrites and are the only neurons localized around the cc in a hypo-neuronal region enriched with astrocytes and microglia[37]. One crucial question, that is still controversial due to experimental limitations and contradictory reports, concerns the presence of CSF-cNs and whether they would play a similar sensory function in human SC. First, the cc is thought to have collapsed in adult Humans below the cervical region[44] in older subjects but more recent studies have shown that the cc is preserved along the SC[45–47]. Second, there is no evidence for the presence of CSF-cNs in Human and one of the first genetic analyses of Human spinal cord failed to reveal the presence of the gene for PKD2L1 presumably due to experimental limitations[48] (but see ref. 49). Nevertheless, this remains an open question that needs to be addressed in the future to demonstrate whether this unique neuronal population is conserved in Human and if not whether it has been replaced by another system capable of integrating CSF circulating signal in the SC.

Reports from studies in lamprey suggest that intra- and subependymal CSF-cNs represent two distinct subpopulations (type I and II)[6]. This property is also observed in embryonic and postnatal mice, where dorsally located CSF-cNs express Olig2 (CSF-cNs') and latero-ventral ones express Nkx2.2 and Nkx6.1 (CSF-cNs'')[8,28]. A similar organization is observed in zebrafish larvae[17] as well as in postnatal mice[8,28]. It has been suggested that ventral CSF-cNs may have a 'more' immature neuronal phenotype, which aligns with findings in older mice showing a largely conserved immature profile (expression of Nkx2.2, Nkx6.1, doublecortin and low levels of the Neuronal Nuclear protein)[7,8,13,29]. Nevertheless, in older mice, medullo-spinal CSF-cN subpopulations do not cluster specifically around the cc[7,13,29], possibly due to developmental reorganization.

CSF-cN axons form long, bilateral fiber bundles in ventral SC at the median fissure[5,11,13–15] running along several SC segments and extending collaterals within the spinal tissue[14–17] with recurrent synapses onto CSF-cNs[14,15]. Unlike the long range sparse GABAergic projections[50] observed in the hippocampus[51] and the inferior olive[52], CSF-cNs form a unique dense network of ascending fibers. In zebrafish larvae, CSF-cNs have distinct projection paths and postsynaptic targets based on their dorso-ventral location, with ventral CSF-cNs showing longer projections, more branching, and a larger arborization area[17]. These neurons would have different postsynaptic targets[17,25,33,53]. Our study shows that CSF-cN axons project to the median fissure in the SC, forming long ascending fiber bundles in the lumbar to cervical segments. Nakamura and colleagues[14] further demonstrated that CSF-cNs primarily project rostrally, a result previously reported by Gerstmann and colleagues[15], with a number of collaterals that is the highest in the dorsal part of cervical and the ventral lumbar segments. This higher density of CSF-cN projection in cervical and lumbar segments could be related to specific local spinal networks and the associated function. Using selective viral retrograde tracing from the lumbar region, it was confirmed in mice that CSF-cNs project to rostral segments and form functional synaptic contacts with more rostral CSF-cNs[15]. However, in mice

CSF-cNs in ventral or dorsal regions could not be selectively labeled to demonstrate specific axonal projection paths *i.e.* postsynaptic targets relative to their localization around cc and rostro-caudal segments.

At the presynaptic level, it was shown that CSF-cNs project along the cc axis in an ascending manner to contact other CSF-cNs[15] and monosynaptic retrograde tracing indicates that CSF-cNs are mainly contacted by GABAergic neurons although glutamatergic presynaptic partners have also been identified[15]. Nevertheless, the precise phenotype and localization within the spinal tissue of these presynaptic partners remains unknown and further investigations are needed to resolve this issue.

There might be phenotypical and anatomical evidence pointing towards the existence of two CSF-cN subpopulations in mice that would receive inputs from and project to specific presynaptic and postsynaptic partners, respectively. However, to date, no experimental data are available to support this organization and demonstrate its functional relevance in the different medullo-spinal segments. One might hypothesize that these identified subpopulations may differ by their integration in local networks (*i.e.* specific pre- and postsynaptic partners) to serve a modulatory function specific to a given medullo-spinal segment (see below). This hypothesis, that is investigated at the neuroanatomical level by several research groups in the field, is the first fundamental step to allow the characterization of CSF-cN functional connectivity and the better understanding of CSF-cN function in the CNS.

## Along the spinal central canal, CSF-cNs exhibit region-specific electrophysiological properties

CSF-cNs have been primarily studied at the behavioral level in zebrafish[5,16,17,26,27,32,33] and to a lesser extent at the cellular one in mouse medullo-spinal tissues[5,8,12,18,19,21,31,40,54]. Across these models, CSF-cNs share common properties, including high input resistance in the Giga-ohm range and small membrane capacitance, in agreement with their small soma size and limited dendritic branching. Our data show that spinal CSF-cNs exhibit largely conserved passive properties along the rostro-caudal axis, from the *medulla* to the lumbar SC, with some differences across spinal segments that might have a strong impact on CSF-cNs properties.

PKD2L1, a hallmark of medullo-spinal CSF-cNs[5,12,21,22,27], is confirmed in our study as being expressed and functional along the SC. Although we did not perform recordings in CSF-cNs obtained from PKD2L1 knockout mice, we are confident that the unitary current recorded in spinal CSF-cNs is carried by PKD2L1. The unitary current was systematically and only observed in the recorded neurons that exhibited the characteristic morphology of CSF-cNs visualized from the expression of tdTomato fluorescence and confirmed using soluble fluorescent markers (Alexa488 or 594 hydrazide for Pkd2l1-Cre::tdTomato or wild-type mice, respectively) added to the recording intracellular solution. Further, it exhibits the characteristic electrophysiological properties (large unitary current amplitude and low open probability) as well as the pH-sensitivity also reported in medullary CSF-cNs (and see Supplementary Fig. 3A). This non-selective cationic channel is modulated by extracellular pH and can trigger APs, influencing CSF-cN excitability[21]. We found that PKD2L1 is activated by alkaline pH and inhibited by an acidic one, while spinal CSF-cNs also respond to acidic pH *via* ASICs activation. We did not characterize the ASIC isoform expressed in spinal CSF-cNs using pharmacological tools. However, based on the similarities of the current kinetics between our recordings and those reported for medullary CSF-cNs[12,55], one can suggest that spinal CSF-cNs would also express ASIC1a homomers or ASIC1a/2b heteromers. This assumption is further supported by the recent report by Yue and collaborators[35] (see also Supplementary Fig. 3A). We further demonstrate that both mechanisms affect CSF-cN excitability, supporting a conserved sensory function along the cc. Although chemosensitivity has been demonstrated using non-physiological stimuli, further research is needed to identify the circulating signals modulating CSF-cNs as well as the receptors they express to respond to chemical cues present in the CSF.

We observed that T- and C-CSF-cNs exhibit a depolarized RMP at approximately -50 mV, consistent with previous findings for medullary

CSF-cNs[12,21]. In contrast, L-CSF-cNs display a more hyperpolarized RMP of around -65 mV, a result recently reported using the less invasive cell-attached recording technique[31]. This suggests that L-CSF-cNs may have different ionic conductances or reduced PKD2L1 activity, contributing to their hyperpolarized membrane potential. However, our study shows no differences in the expression of major $K_V$ channels or PKD2L1 activity.

In postnatal mice[8] and juvenile rats[9], CSF-cNs were shown to form two major neuronal subpopulations present in different regions around the cc, exhibiting different states of maturity and firing patterns (see above). In older mice, we have shown that two medullary CSF-cN populations can be distinguished based on their maturity state or firing pattern[12,21], an observation confirmed for spinal CSF-cNs in this study. However, in older mice and in contrast to the data obtained in zebrafish or postnatal mice[8], medullo-spinal CSF-cNs are not grouped in specific clusters but rather distributed around the cc. One can argue that during development and maturation CSF-cNs may redistribute around the cc or might undergo developmental processes to change their phenotypical and functional properties. Dedicated developmental studies are required to address this specific point. We also observed that C- and L-CSF-cNs exhibit mainly tonic firing, while thoracic neurons mostly show single-spiking pattern. The single-spike profile with neurons emitting one AP in response to depolarization, but without being able to sustain regular firing, is often associated with immaturity and low expression of voltage-gated sodium channels underlying action potential firing[30,56]. For CSF-cNs, the single-spike firing profile has been classically used to define an immature *i.e.*, low maturity state[8,9,28,36]. However, to date there is no evidence in CSF-cNs of the correlation of single-spike firing and the expression of immaturity markers. One would need to conduct dedicated studies to address this point. When referring to single-spike discharge, alike other reports[8,9], we would rather consider a discharge pattern with a 'depolarization block' where membrane potential depolarization impedes the generation of AP (in Fig. 3A, compare depolarization induced by +10 and +30 pA current injections) presumably due to a lower density of sodium voltage-dependent channels (see Marichal and collaborators[9]). This result suggests different properties and ionic conductances expression for CSF-cNs along the rostro-caudal axis. Due to the recording conditions and dedicated protocols implemented to analyze on one hand AP firing (potassium-based intracellular solution for current-clamp recordings) and on the other isolated ionic conductances (cesium- and TEA-based intracellular solution in voltage-clamp recordings to isolate $Na_V$), we cannot correlate spiking profile and Nav densities. This might also depend on the specific network CSF-cNs are inserted in as well as to the exposure to given bioactive signals. Further, CSF-cNs are present in the so-called spinal niche in contact with the CSF and appear to interact with ependymal cells[35]. One could suggest that they might change functional properties depending on the physio-pathological state as recently demonstrated following SC injury[35]. The observed difference could also be explained by the properties of the networks CSF-cNs are inserted in. However, one has to remain cautious in concluding about AP discharge activity when using whole-cell patch-clamp technique especially in neurons with small somatas. An answer to this point could be addressed in vivo using imaging approaches (calcium and voltage sensitive probes) or electrophysiological recordings.

Marichal and colleagues[9] demonstrated that in juvenile rats CSF-cNs express functional TTX-sensitive $Na^+$ ($Na_V$) and $K^+$ voltage-dependent channels, including delayed-rectifier ($K_{DR}$) and A-type ($K_A$) channels. Our findings align with these reports and the transcriptomic analysis recently published by Yue and collaborators[35], showing that spinal CSF-cNs express $Na_V$ channels at similar densities, mediating only transient and TTX-sensitive currents, as we did not observe any persistent or TTX-resistant $Na^+$ currents. Regarding $K_V$, all CSF-cNs exhibit large $K^+$ currents, characterized by a fast, transient phase followed by a persistent one. In agreement with previous reports[9], our pharmacological analysis, supported by transcriptomic data, reveals that spinal CSF-cNs exhibit 4-AP-sensitive fast and transient currents along with TEA-sensitive persistent ones. This indicates that, as in juvenile rats[9], mouse spinal CSF-cNs express $K_{DR}$ and $K_V1$ channels at similar densities. These $Na_V$ ad $K_V$ channels likely underlie AP

generation in spinal CSF-cNs. However, unlike Marichal and colleagues[9], we did not observe differences in current amplitude or density based on the firing pattern (single spike *vs.* tonic) of CSF-cNs (and see above). It is possible that modulation of these channels by intracellular signaling pathways, potentially activated by agents circulating in the CSF, could differentially modulate CSF-cN excitability. Such modulation might directly affect AP properties or indirectly influence them through PKD2L1, which could subsequently trigger AP firing.

In juvenile rats[9] and more recently in the mouse thoracic SC[19], CSF-cNs were shown to express functional $Ca_V$ channels, including the T-type LVA subtype ($Ca_V3$). We previously reported the absence of $Ca_V3$ channels in mouse medullary CSF-cNs, where N-type channels ($Ca_V2.2$, HVA) were highly expressed. However, in mouse spinal CSF-cNs, we now demonstrate HVA expression along with LVA channel subtypes as indicated by the characteristic 'current shoulder' observed when neurons are held at -80 mV, but not -60 mV, prior to $Ca_V$ activation (and see Yue and collaborators[35]). The presence of T-type $Ca^{2+}$ channel might contribute to CSF-cN excitability as well as $Ca^{2+}$ signaling[31], particularly through rebound burst firing after inhibitory synaptic inputs that hyperpolarize the cells and might enable T-type channel de-inactivation and activation. We also found that thoracic CSF-cNs have the highest expression level of $Ca_V$ channels, as reflected in the larger amplitude of $I_{Ca}$ densities recorded in this region. This suggests that $Ca^{2+}$ signaling is differentially regulated along the SC axis, with higher $Ca_V$ density and calcium signaling in thoracic segments.

We, finally, investigated the modulation of CSF-cN activity and $Ca^{2+}$ signaling by examining $Ca_V$ modulation *via* G protein-coupled receptors (GPCRs). Consistent with previous findings in the brainstem[18], we found that $Ca_V$ channels in all spinal CSF-cNs are reversibly inhibited following $GABA_B$-R (expression of Gb1a and Gb2, two subunits of $GABA_B$-R) activation and also indicate a modulation by mACh-Rs (possibly of the M2-type). Both GABAergic and cholinergic GPCR activations led to similar levels of $Ca_V$ inhibition, likely mediated through the classical $G\beta\gamma$ pathway[57]. Interestingly, we did not observe any $Ca_V$ inhibition via glutamatergic GPCRs, suggesting that either these receptors are not expressed in CSF-cNs or they do not modulate $Ca_V$ in this population (absence of $G\alpha_{i/o}$ coupled mGlu-R isoforms; see Supplementary Fig. 3G). As reported for medullary CSF-cNs[18] and more recently in lumbar segments[31], we found no evidence for functional GIRK channels in spinal CSF-cNs, as $GABA_B$-R activation failed to induce outward currents in all recorded neurons.

Here, we confirm our previous findings in medullary CSF-cNs and show that PKD2L1 is expressed in CSF-cNs along the SC axis, exhibiting spontaneous activity and contributing to CSF-cN excitability[21]. Supporting their chemosensory role[21–23,26,27], we report that spinal CSF-cN excitability is modulated by extracellular acidification through activation of ASICs, presumably of the ASIC1a or 1a/2b, or by alkalinization *via* increased PKD2L1 channel open probability. CSF-cNs are capable of differentiating pH changes by responding to both increases and decreases in pH through specific ionic channels. It is likely that bioactive compounds in the CSF activate intracellular pathways, modulating CSF-cN activity either directly or indirectly through PKD2L1 or other receptors[34,35], though these compounds (such as neurotransmitters, metabolites, or neuropeptides) remain largely unidentified in mice. In zebrafish larvae, PKD2L1 integrates mechanical signals by detecting CSF flow and SC bending, interacting with the Reissner fiber[26,27]. A similar mechanosensitivity was suggested in mouse thoracic CSF-cNs, although strong mechanical stimuli were required to modulate their excitability[19]. We were not able to confirm this result, since in our hands pressure application of agonists or aCSF failed to induce changes in PKD2L1 activity. Identifying the precise sensory stimuli to which CSF-cNs respond remains a critical open question in mice and is essential for understanding their function.

## CSF-cNs show a differential excitatory/inhibitory drive along the central canal axis

CSF-cNs might integrate circulating signals in the CSF to sense the CNS homeostatic state. On the other hand, they appear to be integrated along the

cc within intra-medullary and -spinal networks, which are involved in regulating CNS activity and body physiology. Recent studies[14,15] have begun to explore CSF-cN connectivity, but limited information is available regarding their presynaptic partners. In mice, medullary CSF-cNs express $GABA_A$, glycine, and glutamatergic receptors and primarily exhibit spontaneous GABAergic and glycinergic synaptic activity at low frequencies (~1 Hz)[18] and both GABAergic and glutamatergic synaptic currents were evoked in these neurons[18]. However, the identity and localization of the presynaptic partners remain unknown. Our study shows that along the spinal axis, CSF-cNs express functional $GABA_A$, $GABA_B$, and glycine receptors, with large inhibitory currents elicited by exogenous agonists, suggesting robust inhibitory control. In the cervical segment the inhibitory signaling appears lower. In the lumbar region, CSF-cNs were found to form two subpopulations, one depolarized and the other hyperpolarized by $GABA_A$ receptor activation[31]. Although this has not yet been confirmed in thoracic, cervical and, medullary regions, it is plausible that similar subpopulations exist. Furthermore, we show that spinal CSF-cNs express AMPA/kainate-type glutamatergic receptors, whose exogenous activation induces small, slow-kinetic currents, consistent with reports from the *medulla*[18]. This contrasts with the large, fast AMPA currents seen in other neurons or those electrically evoked in medullary CSF-cNs. Our results further indicate differences in the current densities of glycine and $GABA_A$ receptors-mediated responses that are highest in thoracic CSF-cNs. We also report in lumbar CSF-cNs lower current densities for glutamate (AMPA/kainate)-mediated responses underlying the observed lower excitatory action of AMPA/kainate receptor activation. Due to the recording configuration (whole-cell patch-clamp and artificial intracellular chloride concentration), it is difficult to discuss the functional consequence of the inhibitory signaling onto spinal CSF-cNs (see also Riondel and colleagues[31]). Nevertheless, altogether, these results might imply that the excitatory and inhibitory drives onto CSF-cNs differ along the cc axis and that CSF-cNs would be differentially modulated through synaptic inputs within local spinal networks. The functional consequence of this differential excitatory/inhibitory drive is unknown but opens future lines of research where the underlying connectivity leading to a differential integration within segment specific local networks needs to be characterized. We are currently addressing these aspects in dedicated morphological and functional studies. Additionally, CSF-cNs along the spinal cc axis express muscarinic and nicotinic cholinergic receptors. Despite the small amplitude of these glutamatergic and cholinergic currents, due to the high CSF-cN input resistance, receptor activation can induce significant depolarization and even trigger APs. Thus, activation of glutamatergic and cholinergic ionotropic receptors in CSF-cNs likely modulates their excitability. However, the effects induced by the activation of these two receptors are different suggesting that CSF-cNs might integrate or code differentially the synaptic inputs they receive depending on the network they are inserted in. Within the spinal tissue numerous GABAergic, glutamatergic, and cholinergic interneurons are present and shape spinal activity and outputs (somatic and autonomous motor systems). Recent reports[14,15,35] and our data indicate that CSF-cNs express the major synaptic receptors and would receive inputs from or project to spinal interneurons. Therefore, CSF-cNs might be key players in the SC circuitry in the mammals by bidirectionally interacting with different neuronal populations to regulate CNS activity.

Spontaneous synaptic activity has been observed in medullary CSF-cNs, but sparsely in spinal CSF-cNs, raising questions about the source of neurotransmitters and their release conditions in activating spinal CSF-cNs. Neurotransmitters might be released *via* synaptic contacts with glutamatergic, GABAergic, and cholinergic neurons within the SC, nevertheless, except for recurrent connectivity among CSF-cNs as a source of GABA[14,15], functional contacts with CSF-cNs remain to be definitively demonstrated. Another potential origin could be paracrine release from neurons in the surrounding parenchyma, but this seems unlikely, as no changes in baseline current were observed upon exposure to selective antagonists. However, one cannot rule out a loss or 'dilution' of such paracrine transmission in in vitro models where slice preparation is perfused with aCSF. Additionally,

neurotransmitters have been shown to be present in the CSF and might activate CSF-cNs *via* their protrusions, as suggested by a recent study indicating that CSF-cNs can be activated by κ-opioids released by neighboring cells[35]. Such a route of activation remains to be demonstrated. Nevertheless, there is growing anatomical evidence indicating that CSF-cNs appear integrated within specific spinal networks and one major challenge in the coming years will be to further characterize CSF-cN presynaptic partners and to understand how they are activated. We recently used monosynaptic retrograde neuronal tracing[15] to show that L-CSF-cNs are primarily contacted by GABAergic neurons, with some glutamatergic input as well. Extending this type of characterization to the entire SC and testing functional connectivity in vivo will be crucial.

### Concluding remarks

Altogether, the data collected in this study represent a comprehensive analysis of CSF-cN distribution and highlight electrophysiological regional differences along the SC axis. It contributes to better characterize the properties of spinal CSF-cNs and to understand their functions. We indicate that they represent, at the morphological level, a homogeneous population among species and a uniform one along the medullo-spinal axis. They form an interconnected neuronal network to act as a coordinated sensory system. While, they form synaptic contacts with pre- and postsynaptic neurons that largely remain to be identified and might also be modulated by signals circulating in the CSF or present in the parenchyma. In physiological and pathological conditions, CSF-cNs would integrate sensory signals and in turn modulate, along the medullo-spinal axis, specific networks regulating or controlling either motor[14,15,25,32,33], autonomous or even regenerative functions[35].

As such, we propose that CSF-cNs represent a novel actor of the interoceptive system capable of informing the CNS about its inner state and at the same time modulating specific physio-pathological outputs according to the segment there are located in.

The challenge in the coming years to demonstrate the role of CSF-cNs as interoceptors will be to characterize their connectivity, since, as Kolmer suggested it, "*in many of these cases, precise anatomical knowledge [is] a prerequisite for physiological research*"[58].

## Materials and methods
### Animal ethics / Ethical approval

We have complied with all relevant ethical regulations for animal use and all experiments were conducted in conformity with the rules set by the *EC Council Directive* (2010/63/UE) and the French "*Direction Départementale de la Protection des Populations des Bouches-du-Rhône* (DDPP13)" (Project License Nr: APAFIS 44331,2023071917567777 & 33336, 2021101819022439. WN and License for the Use of Transgenic Animal Models Nr: DUO-5214). Protocols used agree with the rules set by the *Comité d'Ethique de Marseille* (CE71), our local Committee for Animal Care and Research. All animals were housed at constant temperature (21 °C), in an enriched environment, under a standard 12 h light-12 h dark cycle, with food (pellet AO4, UAR, Villemoisson-sur-Orge, France) and water provided *ad libitum*. Every precaution was taken to reduce to the minimal the number of animals used and minimize animal stress during housing and prior to experiments.

### Animal models

We used wild type C57 Black6J (Charles River, MGI ID: 3028467), Pkd2l1-Cre (Pkd2l1tm1(cre); MGI ID: 6451758; a generous gift Emily Leman), and Choline Acetyl Transferase-Cre (ChAT-Cre; Chattm2(cre)Lowl; The Jackson Laboratory, MGI ID: 5475195; RRID:IMSR_JAX:006410) mice. Pkd2l1- and ChAT-Cre animals were cross-breed with Gt(ROSA)26Sortm14 (CAG-tdTomato)Hze (The Jackson Laboratory, MGI ID: 3809524; RRID:IMSR_JAX: 007914) to generate Pkd2l1-Cre and ChAT-Cre::tdTomato mice and selectively express the tdTomato fluorescent protein in the neuronal population of interest. Animals (3–6 Weeks old mice) of both sex were used but not recorded for histology as well as for electrophysiological recordings.

### AAV injections

3–5-week-old mice were deeply anesthetized with isoflurane and placed in a stereotaxic frame. Meloxicam (5 mg.Kg$^{-1}$; Metacam, Boehringer Ingelheim), a non-steroïdian anti-inflammatory compound, was administered intraperitoneally at least 30 minutes before surgery. Prior to a skin incision, lidocaine (5 mg.Kg$^{-1}$; Lurocaïne, Vetoquinol) was administered subcutaneously under the zone. Craniotomy was performed using a motorized drill (78001 Microdrill, RWD life sciences) with drill bits diameters of 1 mm (WPI) at the following coordinates (AP 0.0, ML 1.0 mm; DV 1.6 mm) and 700 nL of AAV1-hSyn-EGFP viral particles at $2 \times 10^{13}$ GC.mL$^{-1}$ were delivered using a Nanoject-III system (Drummond Scientifc Company) at a rate of 4 nL.s$^{-1}$.

### Histology and imaging
**Tissue collection for imaging and microscopy.** Pkd2l1-Cre::tdTomato (Figs. 1 and 2 and Supplementary Fig. 1 and 2) or ChAT-dTomato (Supplementary Fig. 2) animals were injected 30 min prior to the procedure with Meloxicam (5 mg/Kg 30 min prior to procedure). Subsequently mice were anesthetized with intraperitoneal administration of 100 mg/Kg ketamine (Virbac) and 10 mg/Kg xylazine (Rompun, Bayer). Before skin incisions, the local analgesic lidocaine (5 mg/Kg; Lurocaïne, Vetoquinol) was injected subcutaneously. Afterwards, animals were transcardially perfused with 20 mL of ~37 °C PBS 0.1 M followed by 20 mL of ice-cold 4% paraformaldehyde (EMS, 15714S) in PBS 0.1 M. SC were removed by ventral laminectomy and the meninges gently separated prior to overnight (o/n) fixation at 4 °C in the same fixative solution. For clearing experiments, brain plus SC (whole central nervous system, CNS) were dissected and post-fixed o/n at 4 °C in the same fixative solution under gentle shaking.

**Routine histology and confocal microscopy.** SC were cut in three segments corresponding to cervical, thoracic and lumbar regions. We typically focus on the medial levels of each region, deliberately avoiding the most caudal and rostral poles of each segment. Afterwards, each one of the segments was included in a low melting point agarose block (4% in PBS). These blocks were glued to a vibratome (HM 650 V, Microm, Thermo Scientific) stage, and coronal slices (50 μm) of SC were collected as floating sections. After incubation in 1.5 μg/mL of DAPI (D9542, Sigma-Aldrich) in 0.1 M PBS (15 min, RT), sections were washed one last time in PBS and mounted in Vectashield (Vector laboratories, H-1900-10). Slides were kept at 4 °C until use. 8-bits Z-stacks images were acquired on a Zeiss LSM-700 Confocal scanning microscope equipped with ZEN 2009 software (Zeiss) using an EC Plan-Neofluar 10x, 0.30 NA objective, zoom = 0.7x, or a Plan-Apochromat 20x, 0.8 NA objective, zoom=1x or 1.5x. Optical thickness was 4.45 μm for 10x and 0.75 μm for 20x acquisitions.

**Clearing and light sheet microscopy.** For SC clearing, the vDISCO protocol[38,39] was followed with minimal modifications. Since SCs are highly myelinated tissues, to facilitate the access of nanobodies to CSF-cNs, a detergent-based delipidation step derived from the SHANEL[59] protocol was pre-applied before boosting the endogenous tdTomato signal. Briefly, brain plus SCs were incubated in a solution containing 10% wt/vol 3-[(3-cholamidopropyl)dimethylammonio]-1-propanesulfonate (CHAPS; Carl Roth, 1479.1), 25% vol/vol N-Methyldiethanolamine (N-MDEA; Sigma-Aldrich, 471828), 0.05% Sodium Azide in Type 1 diH20 during 24 h at 37 °C. After washing in 0.1 M PBS (3 times; 2 h/each, ea), samples were transferred to the permeabilization solution containing 0.5% vol/vol Triton X-100 (Sigma-Aldrich, T8787), 5 mM Methyl-β-Cyclodextrin (Sigma-Aldrich, 332615), 0.3 M Glycine (Sigma-Aldrich, G7126), 1.5% vol/vol Normal Goat Serum (Vector laboratories, S-1000), 0.05% wt/vol Sodium Azide in 0.1 M PBS, and incubated 48 h at 37 °C. Subsequently, samples were transferred to permeabilization solution containing 1:600 vol/vol FluoTag anti-RFP Alexa647 (, NanoTag Biotechnologies, N0404-AF647-L; RRID:IMSR_JAX:007914) and incubated during 5 days at 37 °C. Afterwards, samples were washed 3 times for

2 h/ea, and once o/n in washing solution containing 0.5% vol/vol Triton X-100 (T8787, Sigma-Aldrich), 1.5% vol/vol Normal Goat Serum (Vector laboratories, S-1000) in 0.1 M PBS at room temperature (RT). Then, samples were washed 4 times (2 h/ea) in 0.1 M PBS at RT and the clearing process was immediately started. To preserve the SC in a linear disposition, samples were gently stretched and pinned with 0.15mm-diameter stainless steel entomology pins to a thin 4% wt/vol low melting agarose block before clearing. For clearing, samples were incubated at RT in the following gradient of Tetrahydrofuran (THF; Sigma-Aldrich, 186562) in Type 1 H20, 1 h/ea step: 50% vol/vol THF, 70% vol/vol THF, 80% vol/vol THF, 100% vol/vol THF, and o/n in 100% THF. After dehydration, samples were incubated in dichloromethane (DCM; Sigma-Aldrich, 270997) 1 h at RT, and finally in Benzyl Alcohol-Benzyl Benzoate 1:2 vol/vol (BABB; Sigma-Aldrich, 305197 and B6630 respectively) at RT until transparency. All incubations were performed under gentle shaking on an orbital shaker. Prior to imaging, samples were transferred to Ethyl cinnamate (ECI;, Sigma-Aldrich; W243000) since it was the medium used for imaging.

**Light-sheet microscopy imaging.** 16-bits image stacks were acquired using an Ultramicroscope Blaze (Miltenyi Biotec) equipped with the following lasers and filter sets [laser (emission/bandwidth)]: 445(480/40), 488(525/50), 515(540/30), 561(595/40), 639(680/30). We used the 445(480/40) and 639(680/30) configurations for acquisition of the autofluorescence signal (to get the whole morphology of the brain plus SC) and boosted tdTomato signal, respectively. The following objectives were used depending on the acquisition needs: 1.1x objective, 0.1 NA, ≤17 mm WD; 4x Objective, 0.35 NA, ≤17 mm WD; 12x Objective, 0.35 NA, ≤11 mm WD.

When Tile scans were obtained, a minimum of 20% overlapping was used to ensure proper stitching of the tiles. Exposure time was 100–200 ms and laser power was adjusted depending on the intensity of the fluorescent signal (to prevent saturation) and on the magnification used. The light-sheet width was kept at 80% for low magnification imaging and at 20% for high magnification imaging. The Z-Stacks shown in Fig. 1A have a voxel size of $8.21 \times 8.21 \times 7.09\,\mu m$ (x, y, z respectively). The Z-Stacks shown in Fig. 1B, D have a voxel size of $1.625 \times 1.625 \times 2\,\mu m$ (x, y, z respectively). The Z-Stack shown in Supplementary Fig. 1C, D has a voxel size of $0.65 \times 0.65 \times 2\,\mu m$ (x, y, z respectively).

## Image processing
Image analysis and rendering was performed on a workstation equipped with an 8 core Intel Xeon Silver 4215 R processor (128 Gb RAM) and a NVIDIA GeForce RTX 3090 (24 Gb) graphics card. FIJI[60] and Chimera X[61,62] were used for image processing, visualization, and rendering. Light-sheet image tiles were stitched with BigStitcher[63] plugin and further processed in FIJI. For Image rendering, a rolling ball background subtraction[64] was used over both channels with a radius size according to the particles of interest (Width of the SC in 445(480/40) and diameter of CSF-cNs axon bundles in 639(680/30)) in the different images. Then, the channel containing the specific signal (AF647) was subsequently treated with Contrast Limited Adaptive Histogram Equalization (CLAHE) in FIJI using a block-size larger (in pixels) than the objects to be preserved in order to compensate fluorescence discrepancies between CSF-cNs somas and thin processes. The background signal corresponding to the surface of the SC was removed by thresholding the autofluorescence signal and creating a reduced selection (in pixels) over the tdTomato (AF647) signal. The function "Clear Outside" was then used in FIJI over this selection to remove surface background fluorescence ensuring that objects in the specific channel (AF647) were preserved. This strategy did not compromise the objects of interest since CSF-cNs and their axon bundles are distant from the surfaces of the SC. Finally, for image rendering, an "Unsharp Mask" filter was used with a radius of 0.9 pixels and a Mask Weight of 0.7. Rendering was performed in Chimera X by loading the previously generated. tiff stacks and adjusting graphical parameters.

"Maximum Intensity Projection" and "3 d projection mode" were used when displaying volumes.

For low-magnifications large field of view images in Fig. 2, individual Z-stacks were acquired as positions and stitched through Grid Collection Stitching plugin in FIJI[65] to avoid edge artifacts derived from Tile acquisition in ZEN software. For high-magnification images in Fig. 2, images were processed with CLAHE in order to avoid fluorescence discrepancies between the somas and the thin axonal processes of CSF-cNs since linear adjustment will inevitably result in saturation at the somatic level.

## Acute spinal cord slice preparation
Coronal SC slices (from lumbar to cervical segments) were prepared as follow[18]. Briefly, wildtype or Pkd2l1-Cre::tdTomato mice were injected with Meloxicam (5 m/Kg; 30 min prior to procedure). Subsequently, they were anaesthetized with intraperitoneal injection of ketamine–xylazine mixture (100 and 15 mg/kg, respectively) and injected on the site of surgery with Lurocaine (5 m/kg), a local analgesic. Next, animals were transcardially perfused with an ice-cold (0-4 °C), oxygenated (95% $O_2$/5% $CO_2$) and low calcium/high magnesium slicing solution containing (in mM): NaCl 75, $NaHCO_3$ 33, $NaH_2PO_4$ 1.25, KCl 3, $CaCl_2$ 0.5, $MgSO_4$ 7, glucose 15, sucrose 58, ascorbic acid 2, Na-pyruvate 2, myo-Inositol 3 (pH 7.3-7.4 and osmolarity of 300-310 mosmole.kg$^{-1}$). Following laminectomy and SC dissection, lumbar to cervical SC slices (250 µm thick) were cut with a vibratome (Leica VT1000S) in ice-cold cutting solution saturated with 95% $O_2$-5% $CO_2$. Slices were subsequently transferred to a recovery chamber and incubated at 35 °C for 15-20 minutes in oxygenated artificial CSF (aCSF) containing (in mM): NaCl 115, $NaHCO_3$ 26, $NaH_2PO_4$ 1.25, KCl 3, $CaCl_2$ 2, $MgSO_4$ 2, glucose 15, ascorbic acid 2, Na-pyruvate 2, myo-Inositol 3 (pH 7.3-7.4 and osmolarity of 300–310 mosmole.kg-1) and then kept at RT to recover for 1 h. After recovery, slices were transferred one by one to the recording chamber.

## Electrophysiological recordings
**CSF-cN visualization and recording of their intrinsic properties.** Lumbar, thoracic, or cervical SC acute slices (not distinguishing the rostro-caudal position) prepared from wild type or Pkd2l1-Cre::tdTomato mice were transferred in a recording chamber mounted on an upright microscope (Zeiss Axioskop 1FS or Scientifica SliceScope Pro 1000) equipped with infra-red differential interference contrast or oblique illumination (IR-DIC or IR-Oblic) and a p1 or p300 Ultra precisExcite LED epifluorescence system (CooLED). CSF-cNs around the cc were visualized using a computer controlled digital camera (HQ2 CoolSnap, Photometrics, SciCam, Scientifica) under epifluorescence illumination (exc. 520 nm/em. 610 nm, tdTomato fluorescence) and/or IR illumination with a 40x or 60x objective. Whole-cell patch-clamp recordings were performed at RT ( ~ 20 °C) in current- and/or voltage-clamp mode using a Axopatch 200 A or Multiclamp 700B amplifiers (Molecular Devices *Inc.*). Patch pipettes (3–7 MΩ) were pulled from borosilicate glass capillaries (OD: 1,5 mm; ID: 0,86 mm; Harvard Apparatus) using the vertical PC-100 puller (Narishige International Ltd) and filled with an internal solution (see detail in Supplementary Table 1). Signals were low-passed filtered at 2–10 kHz and digitized between 10–50 kHz using a Digidata 1322 A, 1440 A, or 1500B interface driven by pClamp 9, 10 or 11 software (Molecular Devices). Series resistance (10–20 MΩ) was monitored throughout the experiment and unstable recording were discontinued (increase by more than 25% from its initial value). For voltage-clamp recordings, the series resistance was compensated by 70-80%. The liquid junction potential was left uncorrected. SC slices were perfused with oxygenated, and pH controlled (95% $O_2$/5% $CO_2$) artificial CSF (aCSF, 2.5 mL/min) and the composition was adapted according to the experiments conducted (see Supplementary Tables 1 and 2).

The recording of CSF-cNs was confirmed based on the characteristic morphology observed (small round soma close or within the ependymal layer and a large dendrite ending in the cc with a round protrusion) from the cytosolic tdTomato as well as from Alexa488 or 594 Hydrazide (added to the

intracellular solution, 10 µM; Invitrogen-Theromefisher, A10436 and A20502, respectively) fluorescence. The presence of spontaneous PKD2L1 channel activity was monitored as a further control. We did not observe differences in the electrophysiological properties of CSF-cNs recorded in acute slices obtained from wild type or Pkd2l1-Cre::tdTomato mice. Resting membrane potential (RMP) was determined under current-clamp mode (I = 0) and CSF-cN spontaneous or induced (step of 200–500 ms, –20 pA with increments of 10 pA, 10–12 sweeps) AP discharges were recorded in current-clamp mode either at RMP or a membrane potential set at –60 mV (current injection of –10 to –15 pA). The intrinsic properties (membrane resistance, $r_m$; time constant, $\tau$ and membrane capacitance, $c_m$) were determined from the current response elicited using a 10 ms voltage step to -10 mV from a holding potential ($V_h$) set at –70 mV in voltage-clamp mode. Note that to account for potential difference in ionic channel and receptor expression as a function of cell size, all current amplitudes are expressed as current density (current/$C_m$).

The characterization of CSF-cN modulation by pH was conducted either in voltage-clamp mode ($V_h$ -80 mV) or current-clamp mode at RMP using standard aCSF complemented with antagonists against $GABA_A$, glycine, and glutamate receptors and a K-based intracellular solution (in mM: KCl 130; NaCl 10; $CaCl_2$ 1; $MgCl_2$ 1, HEPES 10, phosphocreatine 10; Mg-ATP 4; pH 7.35adjusted with KOH and osmolarity of 290 mosmole.kg$^{-1}$). Acidic (pH5 in mM): NaCl 139; KCl 3; $MgCl_2$ 2; $CaCl_2$ 2; glucose 15; HEPES, 10, Na-Citrate 16; pH adjusted with HCl 1 N and 310 mosmole.Kg$^{-1}$) or alkaline (pH9 in mM): NaCl 145; KCl 3; $MgCl_2$ 2; $CaCl_2$ 2; glucose 15; HEPES 10, TAPS 10; pH adjusted with NaOH 1 M and 310 mosmole.Kg$^{-1}$) solutions were applied by pressure for 10 (pH5) or 30 s (pH9) using a patch pipette ( ~ 1 µM tip diameter), positioned at ~50 µm from the recorded cells and connected to pressure application system (Toohey Company, Picospritzer or WPI).

### Recording and characterization of voltage-gated currents.

The expression in CSF-cNs of voltage-gated sodium (Na$^+$; Na$_V$), potassium (K$^+$; K$_V$) and calcium (Ca$^{2+}$; Ca$_V$) channels was assessed in voltage-clamp mode and ionic conductances isolated by specifically adapting the composition of the aCSF and the intracellular solution (see Supplementary Tables 1 and 2).

Potassium, sodium and Calcium voltage-dependent currents were elicited while recording neurons at a $V_h$ of –80 mV or –60 mV and voltage steps ($V_{Step}$) protocols applied to selectively activate a given ionic conductance. Briefly, sodium voltage-gated channels were activated from a $V_h$ of –80 mV and $V_{Step}$ applied from –80 to +60 mV in 10 mV increments applied for 100 ms. Potassium voltage-dependent currents were elicited from $V_h$ -80 mV with a voltage pre-step to –100 mV for 50 ms to ensure full reactivation of fast inactivating currents (i.e. $I_A$) followed by 100 ms $V_{Step}$ ranging from –100 to +60 mV in 10 mV increments before stepping back to -80 mV. To activate $I_{Ca}$, neurons were recorded either from $V_h$ set at -60 mV with voltage steps applied for 100 ms ($V_{Step}$, -40 mV to +30 mV, 10 mV increments) or a voltage-ramp protocol from a $V_h$ of –60 mV or –80 mV ($V_h$ -60 mV: ramp from –60 to +30 mV over 50 ms and 1.8 mV/ms; $V_h$ -80 mV: 50 ms pre-step to –120 mV followed by a ramp from –120 to + 30 mV over 100 ms and 1.5 mV/ms). At the end of each recording, the analyzed conductance was blocked using a specific blocker (see Supplementary Tables 1 and 2) to confirm its nature.

### Recording of ligand-gated-mediated currents.

To assess for the expression of GABAergic (GABA$_A$-Rs), glycinergic (Gly-Rs), glutamatergic (Glu-Rs) and nicotinic cholinergic (nAch-Rs) ionotropic synaptic receptors, CSF-cNs (cervical to lumbar) were recorded at $V_h$ -80 mV and selective agonists applied by pressure (see above). We tested application of γ-amino butyric acid (GABA, 1 mM for 30 ms), glycine (Gly, 1 mM for 100 ms), acetylcholine (ACh, 4 mM for 500 ms), glutamate (Glu, 100 µM for 100–500 ms). The recordings were performed in the presence of a cocktail of selective antagonists to selectively target one specific ionotropic receptor (see text for more details). To identify the glutamatergic

ionotropic receptor subtypes potentially expressed in CSF-cNs, we tested application of α-amino-3-hydroxy-5-methyl-4-isoxazolepropionic acid (AMPA; 100 µM for 100–500 ms), kainate and N-methyl-D-aspartate acid (NMDA; 100 µM for 100–500 ms). We also tested expression of ionotropic purinergic receptors using pressure application of ATP-γ-S (1 mM for 100–500 ms). To confirm the nature of the tested receptors, we applied at the end of the recordings the agonist in the presence of the selective antagonist (see Supplementary Tables 1 and 2).

### Modulation of voltage-gated calcium currents by G-Protein Coupled Receptors.

To assess whether postsynaptic voltage-gated Ca$^{2+}$channels are modulated by G-Protein Coupled Receptors (GPCRs), CSF-cNs were recorded in voltage-clamp mode using a specific intracellular solution (see Table 1). Slices were perfused with oxygenated aCSF supplemented with 0.5 µM TTX, and 20 mM TEA-Cl to selectively isolate Ca$^{2+}$currents. Ca$^{2+}$currents were elicited from $V_h$ -80 mV at the peak amplitude with a 100 ms voltage step to +10 mV preceded by a pre-pulse to -120 mV for 50 ms. Currents were recorded every 20 s over a 5 min period before (control, CTR), during (agonist) and after (Wash) pressure application of a given GPCR agonist: baclofen (Bcl: 100 µM, 40 s) for GABA$_B$ receptors (GABA$_B$-Rs), oxotremorine-M (OxoM; 100 µM, 50 s) for muscarinic cholinergic receptors (mAch-Rs). We also tested $I_{Ca}$ modulation by metabotropic glutamate receptors (mGlu-Rs) using glutamate pressure application (Glu; 100 µM or 1 mM, 50 s).

### Reagents

Unless otherwise stated, all chemicals and drugs were purchased from Sigma-Aldrich. Selective blockers or antagonists were applied to block/inhibit channels and receptors as follows: Na$_V$ with tetrodotoxin (TTX; Alomone labs, T-550); K$_v$ with tetraethylammonium chloride (TEA; Sigma-Aldrich, 86614) and 4-aminopyridine (4-AP; Sigma-Aldrich, 275875); Ca$_V$ with cadmium (Cd$^{2+}$; Sigma-Aldrich, 02908); GABA$_A$ receptors with gabazine (Gbz; SR 95531; Ascent Scientific, ab120042); glycine with strychnine (Stry; Sigma-Aldrich, S8753); glutamate receptors with 6,7-dinitroquinoxaline-2,3-dione disodium salt (DNQX; Tocris Bioscience, 2312); nAch-Rs with D-tubocurarine (D-Tubo; Tocris Bioscience, 2820); GABA$_B$-Rs with CGP54626A (CGP; Tocris Bioscience, 1088); mAch-Rs with atropine (Atr; Sigma-Aldrich, 1044990). For the agonists, the following compounds were used: oxotremorine-M (Oxo-M; Tocris Bioscience, 1067), ATP-γ-S (Abcam Biochemicals) (RS)-Baclofen (Bcl; Tocris Bioscience, 0417), acetylcholine chloride (ACh; Tocris Bioscience, 2809), (S)-α-amino-3-hydroxy-5-methyl-4-isoxazolepropionic acid (AMPA; Tocris Bioscience, 0254), kainic acid (Kainate; Tocris Bioscience, 7065) and N-methyl-D-aspartate acid (NMDA; Tocris Bioscience, 0114), γ-aminobutyric acid (GABA; Tocris Bioscience, 0344), glycine (Gly; Sigma-Aldrich, G7126). Experiments were also performed using pressure application of aCSF in the absence of drugs to test for the artefacts or mechanicals responses. However, no changes in the holding current were observed in any of those control experiments confirming that effects were drug or test solution specific. More details about the experimental procedures and solution used are available in Supplementary Tables 1 and 2.

### Data processing, analysis and statistics

#### Cell counting in Light sheet microscopy images.

Segmentation and cell counting in light sheet microscopy images was performed with FIJI (see Supplementary Fig. 5 for graphical explanation and workflow). Briefly, a rolling ball background subtraction of exactly the size of CSF-cNs soma diameter was used ( ~10 pixels). Afterwards, a difference of Gaussian filter in 2D was used (Sigma1 = 1, Sigma2 = 2) using the CLIJ2[66] version of "Difference of Gaussian Filter". Then, the "H-Maxima Local Maximum detection" plugin (from the SCF-MPI-CBG plugins) was used using a Threshold=1 and a Minimum grey value distance between two local optima=5. To remove the H-Maxima ascribable to the ventral axon bundles of CSF-cNs, a 3D signal of the axon bundles was created over the original images using the CLIJ version of the 3D object counter[67] and

these objects loaded into the 3D manager from the 3D ImageJ suite[68] in order to remove the maxima found within the axon bundles. Afterwards, the remaining particles were filtered by size ( <100 voxels) and sphericity in pixel units ( >0.5) using the 3D manager, since large and non-spherical particles were found to correspond to very low fluorescence signals. The remaining objects were then plotted on an image stack of the same dimensions than the source one and labeled (as 16 bits images) using "3D simple segmentation" of the 3D ImageJ suite. To visually check for correspondence between the labels and CSF-cNs somas the command "Reduce labels to centroids" from CLIJ2 was used. To validate this cell counting strategy, independent acquisitions were made with a 12x objective over a region already imaged with the 4x objective, cells were counted likewise and the total number of cells in this region compared between the two acquisition modes (see Supplementary Fig. 1).

**Cell counting and distribution of CSF-cNs in confocal images**. Five 3 weeks mice were used for CSF-cNs quantification in confocal slices. 10 fields of view (FOV) were acquired per animal and SC segment (total 30 FOV/animal). To count CSF-cNs in confocal coronal slices, Z-stacks of 10–20 images were acquired. Counting was aided by custom FIJI macros available upon reasonable request. In brief, the user was prompted to click in the center of CSF-cNs on a Maximal Projection image while being able to navigate through the Z-stack in order to accurately resolved closely packed cells. Once all the cells were marked, the user was asked to define manually the dorsal and ventral regions on the cc (ventral fiber bundles served as bona fide indicators of SC orientation) and the remaining area in the field of view was automatically classified as lateral. Then, the number of cells in each quadrant and their percentages were automatically calculated, and the cell densities/10 μm of SC were expressed as (Total number of CSF-cNs in the field of view x 10)/(Number of optical slices x Voxel depth). Since we did not observe evidence for CSF-cN lateralization along the rostrocaudal axis of the SC, once the percentage obtained for CSF-cNs in the dorsal and ventral quadrants, the remaining ones was divided by two, and the result was assigned to each of the lateral quadrants (*Left* and *Right*). Moreover, for our study, we have been working with floating sections, and it was therefore not possible to preserve left/right identity. Afterwards the DAPI signal was automatically thresholded to create the contour of the cc, and the Euclidean distance (in microns) of each cell to this contour was calculated using the "Exact Euclidean Distance Transform (3D)" function in FIJI[69] along with the pixel size of the image.

OriginPro was used for statistical analyses and generating plots. Normality was tested for all conditions. Since at least one group was always found to not meet the normality assumption in every possible comparison, *Kruskal-Wallis ANOVA* followed by paired post hoc *Dunn test* was used for cells densities and distributions. Results were considered significant when $p < 0.05$. For comparing distances of CSF-cNs to the cc at different SC levels, Kolmogorov–Smirnov *test* was used. Since K-S *test* is extremely powerful over large data sets, results were considered significant when $p < 0.001$.

**Electrophysiology**. Passive properties were determined, in voltage-clamp mode at $V_h$-70 mV, from the current responses to a −10 mV hyperpolarization step ($V_{Step}$). Membrane resistance ($r_m$) was calculated from the amplitude of the sustained current at the end of the voltage step ($r_m=V_{Step}/I_m$). Membrane capacitance ($c_m$) was estimated as the ratio between the cell decay time constant ($\tau_m$), obtained from the exponential fit of the current decay, divided by the series resistance ($r_s$; $c_m \sim \tau_m/r_s$). Resting membrane potential (RMP) was determined in current-clamp mode at I = 0 just after the whole-cell configuration was achieved. AP discharge pattern was analyzed from depolarizing responses elicited by the injection DC current pulses using the 'threshold detection' routine from Clampfit 9–11 with a threshold set at 0 mV. Voltage and current responses to agonists were analyzed using the Clampfit 9–11 suite (Molecular Devices Inc.), Microsoft Excel 2018-2024 and the R statistics (Version 2024.12.0 + 467©, Posit Software, PBC).

PKD2L1 unitary current analysis was conducted using the WinEDR v4.0.3 software (Strathclyde University) where all points histograms for the close (baseline) and the open (PKD2L1 unitary currents) states were generated on 500 bins over 20–30 s recording periods to determine the respective mean current. Histogram distributions were fitted with gaussian curves defined by the following equation:

$$y(x) = \sum_{i=1\ldots n} \frac{A_i}{\sqrt{2\pi\sigma_i^2}} \cdot \exp\left(-\frac{(X-\mu_i)^2}{2\sigma_i^2}\right)$$

Where $A_i$ corresponds to the area under the curve, $X$ the current amplitude, $\mu_i$ the mean current and $\sigma_i$ to the standard deviation. PKD2L1 unitary current amplitudes were calculated as the difference between the mean current values ($\mu_i$) of the open and close states. The open probability (NPo) was computed from the area under the curve of the open state histogram. Data were analyzed in control, during pH solution application and following wash and compared. For ASICs, the peak amplitude was measure after subtracting the baseline current. Modulation of the spiking properties following exposure to pH solution, was analyzed from the RMP and the instantaneous AP frequency determined using the Clampfit (Molecular Device, Inc.) threshold detection routine.

For $Na_V$ and $K_V$, the current-voltage relationship (IV-curve) for a given conductance was constructed for each cell by plotting the $Na^+$ and $K^+$ current peak amplitude recorded for a given $V_{Step}$, and the mean IV curve generated by pooling the data for all cells recorded from the 3 segments. $Ca_V$ mediated currents were elicited from $V_h$ -60 mV or −80 mV with the voltage-ramp against time protocol. The $Ca_V$ IV-curve was subsequently obtained by converting the data as current against voltage. For each level, to compare recordings elicited from $V_h$ -80 and -60 mV and distinguish HVA and LVA channel activation, the holding current was subtracted from individual converted IV-curves, average curves calculated and subsequently normalized to the peak (current amplitude are presented as arbitrary unit, AU). In the current traces represented to illustrate specific voltage-dependent channel activation against voltage, remaining capacity transients were digitally subtracted.

All data are expressed as mean ± SD or SEM, as specified, and N and n correspond to the number of animal and data samples, respectively. They are represented with boxplots and whisker using the Tukey's method (see Jurčić and colleagues[13]) as well as single data points to illustrate distribution.

Statistical analyses were carried out using the R Studio environment. Since, data did not meet the normality assumption (Shapiro normality *test*), we used non-parametric statistical tests. For results where data are only compared for one variable, we used *Kruskal-Wallis* one-way ANOVA statistical analysis and provide $\chi^2$, degree of freedom (number of parameters-1, df), and $p(\chi^2)$ values, for the analyzed factor followed by a *post hoc pairwise comparisons using Wilcoxon rank sum test* with continuity correction. We compared effect of the effect Control *vs.* 'Drug' (*Condition*) as well as the effect of the region (*Region*). To compare data within a SC segment in different condition (control, CTR *vs.* application of agonists or blockers and wash) as well as between segments along the rostro-caudal axis, the "*linear mixed effect*" (lme version 3.1–128 package[70]) was used. The distinction between CSF-cN response in the different experimental conditions is considered a *Condition* variable, while the distinction between the different neurons along the cc rostro-caudal axis represents the *Region* variable. On one hand, in one neuron for a given *Region*, its response in one *Condition* (control *vs.* drugs application or wash) is assumed dependent of the other recording *Condition*, and the variable represents a dependent factor. On the other hand, the same variables represent independent factors when considered as a function of the *Region* (cervical, thoracic and lumbar). Therefore, the statistical analyses required using a mixed effect model considering in a hierarchical way both dependent (*Condition*) and independent variables (*Region*) as well as the interaction between the variables. The null hypothesis was set as the absence of difference between the data among each group. The *lme* model was tested by using an analysis of the variance (ANOVA) and we

provide the F-statistic (F), the degree of freedom (df) and the p(F) values for the given fixed factor tested (*Condition*, *Region*). Wherever an interaction between the *Region* and *Condition* factors was statistically significant, we reported the p-value. We further conducted a post hoc *Tukey (EMM) post hoc test* to test for Contrasts, pairwise comparisons and interaction as a function of the *Region* and the *Condition*. For data where the sample size was unequal, we used a *Friedman test* followed by *post hoc pairwise comparisons using Wilcoxon rank sum test* with Benjamini-Hochberg adjustment. Statistical differences were considered as significant for $p < 0.05$ and we give in the text and figure legends the $p$ values for each test. In the corresponding figures, asterisks are shown as follow: ns, *, **, *** and **** for p non significative, <0.05, <0.01, <0.001 and <0.0001, respectively. Figures were prepared using R Studio, Adobe illustrator, CorelDraw 2018 and Affinity Designer 2 (v2.6.0).

**Transcriptomic analysis**. Gene expression levels were plotted using custom Python scripts. Gene expression data was parsed from the text file GSE255883_DESeq_norm.txt under accession number GSE255883, filtering for specific gene names. The extracted data was then visualized as heatmaps using Seaborn and Matplotlib to illustrate expression levels across the different samples (refer to Yue et al., 2024 for the specifics).

### Statistics and reproducibility
To ensure data reproducibility, histological experiments were reproduced in 2 to 3 animals and for cells density and distribution, 3 animals were used and up to 50 cells per segments analyzed. For the electrophysiological recordings, experiments were performed in at least 3 different animals and on average at least 10 cells per segments were recorded and analyzed. All details about animals and cells numbers are specified in the results section and in the legend of each figure. To test for significant, the appropriate tests were used and detailed in the Data processing, analysis and statistics section (see above).

### Reporting summary
Further information on research design is available in the Nature Portfolio Reporting Summary linked to this article.

### Data availability

The data of this manuscript are available from the corresponding author upon reasonable request and source data underlying the graphs and charts presented in the study can be found at Figshare depository: https://doi.org/10.6084/m9.figshare.29430869[71].

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

## Acknowledgements

This research was supported by funding obtained from *Aix Marseille Université* (AMU), le *Centre National pour la Recherche Scientifique* (CNRS – INSB) et *l'Agence National pour la Recherche* (ANR-PRCI-MOTAC80C/A134/AN16HRJ NMF, ANR-PRC-MOTOX80C/213Z/AN20HRJNMF & ANR-PRC-SYMPA80C/213Z/AN21HRJNMF; WN). We acknowledge the

'Institut de Neurosciences de la Timone (INT)' technical facilities for their support in the study (*Neuro-Bio-Tools*: Molecular Biology (NeuroVir)/ Histology (ConnectoVir) and *Photonic Neuroimaging* (INPHIM, light-sheet & confocal microscopy) as well as the *Mediterranean Primate Research Centre* (MPRC; UAR 2018), Animal facility).

## Author contributions

All experiments were performed in laboratories at the '*Institut des Neurosciences de la Timone* (INT)' of Aix-Marseille Université & CNRS, UMR7289. Conception and design of the work: C.E., B.E., J.N., R.F.J., S.R. and W.N. Acquisition, analysis, and interpretation of data for the work: C.E., B.E., J.N., M.C., R.F.J, R.P., S.R. and WN. Drafting the work or revisiting it critically for important intellectual content: C.E., B.E., J.N., M.C., R.F.J., S.R., J.T. and W.N. All authors have read and approved the final version of this manuscript and agree to be accountable for all aspects of the work in ensuring that questions related to the accuracy or integrity of any part of the work are appropriately investigated and resolved. All persons designated as authors qualify for authorship, and all those who qualify for authorship are listed.

## Competing interests

The authors declare no competing interests.
