## [Transparent Peer Review file · Communications Biology]

Cerebrospinal Fluid-contacting neurons are sensory neurons with uniform morphological and region-specific electrophysiological properties in the mouse spinal cord

Corresponding Author: Professor Nicolas Wanaverbecq

Version 0:

Reviewer comments:

Reviewer #1

(Remarks to the Author)

Cerebrospinal fluid-contacting neurons (CSF-cNs) are intriguing neurons that are now well characterized in zebrafish where they play a role in sensory-motor control and a similar role has recently been identified in rodents(1). In the present manuscript, the authors have surveyed CsF-cNs along different segments of the spinal cord and provided a lot of convincing data on their intrinsic properties and transmitter sensitivity.

Previous work by the authors and others has shown that CSF-cNs are present at all levels of the spinal chord(2) (3) (4), express voltage-gated Na, K and Ca channels (5-8) are sensitive to pH (6, 9) through expression of ASIC (10) and PKD2L1 (2, 11) (12) and have functional glycine (9), GABAa (6, 9), GABAb (8) and cholinergic (7, 13) receptors. This manuscript reaffirms these previous findings along the spinal cord and as such the findings can't be considered novel.

In the abstract, on line 36, and introduction line 83, it is claimed that CSF-cN functional properties vary along the spinal chord. It is unclear what the authors are referring to, none of the subheadings of the results or discussion indicate differential function. Contrary to this claim subheadings in the results (line 119) and discussion (line 440) states 'Along the spinal central canal, CSF-cNs exhibit similar intrinsic properties'. The authors also show that all identified synaptic receptors are present in each subdivision, with only some variation in the current density for GABAa and iGluR observed. However, the functional relevance of this has not been addressed. The authors should reconsider the claim of functional properties varying along the spinal cord.

Due to the technical issues (low resolution) with light sheet imaging I am not sure what Figure 1 adds to the manuscript that isn't conveyed by Figure 2.

Statistical analyses appear appropriate.

Minor:

- page 3 line 53 you specify rats but don't make it clear that refs 19-21 are from mice. You could possibly change 'rats' to 'rodents' otherwise you should be careful of the species.

- line 63 'regrouped' should just be 'grouped'. Also, it is not evident what is meant by immature vs mature phenotypes. Why is tonic discharge considered mature and single AP firing immature? Perhaps you should be more specific on what is different without referring to the maturation stage of a neuron.

line 71 'finally'... you have an additional point after the finally statement. Rerword.

Cited literature

1. K. Gerstmann, N. Jurčić, E. Blasco, S. Kunz, F. de Almeida Sassi, N. Wanaverbecq, N. Zampieri, The role of intraspinal sensory neurons in the control of quadrupedal locomotion. *Curr Biol* S0960–9822(22)00588 (2022).
2. A. Orts-Del'Immagine, A. Kastner, V. Tillement, C. Tardivel, J. Trouslard, N. Wanaverbecq, Morphology, distribution and phenotype of polycystin kidney disease 2-like 1-positive cerebrospinal fluid contacting neurons in the brainstem of adult

mice. *PLoS One* 9, e87748 (2014).

3. A. Orts-Del'Immagine, J. Trouslard, C. Airault, J.-P. Hugnot, B. Cordier, T. Doan, A. Kastner, N. Wanaverbecq, Postnatal maturation of mouse medullo-spinal cerebrospinal fluid-contacting neurons *Neuroscience* 343, 39–54 (2017).
4. X. Liu, K. Rich, S. M. Nasser, G. Li, S. Hjørnesen, B. Finsen, H. Scherberger, Å. Svenningsen, M. Zhang, A Comparison of PKD2L1-Expressing Cerebrospinal Fluid Contacting Neurons in Spinal Cords of Rodents, Carnivores, and Primates *International Journal of Molecular Sciences* 24, 13582 (2023).
5. Y. L. Petracca, M. M. Sartoretti, D. J. Di Bella, A. Marin-Burgin, A. L. Carcagno, A. F. Schinder, G. M. Lanuza, The late and dual origin of cerebrospinal fluid-contacting neurons in the mouse spinal cord. *Development* 143, 880–891 (2016).
6. N. Marichal, G. García, M. Radmilovich, O. Trujillo-Cenóz, R. E. Russo, Enigmatic central canal contacting cells: immature neurons in “standby mode” *J Neurosci* 29, 10010–10024 (2009).
7. E. Johnson, M. Clark, M. Oncul, A. Pantiru, C. MacLean, J. Deuchars, S. A. Deuchars, J. Johnston, Graded spikes differentially signal neurotransmitter input in cerebrospinal fluid contacting neurons of the mouse spinal cord *iScience* 105914 (2022).
8. N. Jurčić, G. Er-Raoui, C. Airault, J. Trouslard, N. Wanaverbecq, R. Seddik, GABAB receptors modulate Ca²⁺ but not G protein-gated inwardly rectifying K⁺ channels in cerebrospinal-fluid contacting neurones of mouse brainstem. *J Physiol* 597, 631–651 (2019).
9. A. Orts-Del'immagine, N. Wanaverbecq, C. Tardivel, V. Tillement, M. Dallaporta, J. Trouslard, Properties of subependymal cerebrospinal fluid contacting neurones in the dorsal vagal complex of the mouse brainstem. *J Physiol* 590, 3719–3741 (2012).
10. E. Jalalvand, B. Robertson, H. Tostivint, P. Löw, P. Wallén, S. Grillner, Cerebrospinal Fluid-Contacting Neurons Sense pH Changes and Motion in the Hypothalamus. *J Neurosci* 38, 7713–7724 (2018).
11. L. Djenoune, H. Khabou, F. Joubert, F. B. Quan, S. Nunes Figueiredo, L. Bodineau, F. Del Bene, C. Burcklé, H. Tostivint, C. Wyart, Investigation of spinal cerebrospinal fluid-contacting neurons expressing PKD2L1: evidence for a conserved system from fish to primates *Frontiers in Neuroanatomy* 8, (2014).
12. A. Orts-Del'Immagine, R. Seddik, F. Tell, C. Airault, G. Er-Raoui, M. Najimi, J. Trouslard, N. Wanaverbecq, A single polycystic kidney disease 2-like 1 channel opening acts as a spike generator in cerebrospinal fluid-contacting neurons of adult mouse brainstem *Neuropharmacology* 101, 549–565 (2016).
13. L. F. Coms, L. Atkinson, J. Daniel, I. J. Edwards, L. New, J. Deuchars, S. A. Deuchars, Cholinergic Enhancement of Cell Proliferation in the Postnatal Neurogenic Niche of the Mammalian Spinal Cord *STEM CELLS* 33, 2864–2876 (2015).

Reviewer #2

(Remarks to the Author)

CSF-contacting neurons (CSF-cNs) are functionally enigmatic sensory cells facing the central canal of the spinal cord. In this study, Crozat and colleagues examined the morphological and electrophysiological properties of CSF-cNs in the mouse spinal cord by comparing them at the cervical, thoracic, and lumbar levels. The properties were consistent across the spinal levels and aligned with previous observations in spinal and medullar CSF-cNs in mice and juvenile rats (Ref 9, 12, 19, 22), whereas variations in cell density and certain electrophysiological responses were noted between spinal levels. The functional and biological significance of those differences are unclear and not clearly discussed in this study. Overall, the present study provides some advances in the knowledge of spinal CSF-cNs, extending from the previous studies in rodents, especially on the electrophysiological responses at different spinal levels. The experiments were well conducted to obtain the basic electrophysiological properties of CSF-cNs. The manuscript is well organized and the data are interpreted well. Specific comments are as follows:

1. The authors examined the morphological properties of CSF-cNs in clearing tissues and show beautiful images in Fig. 1 and Suppl Fig 1–2. However, unfortunately, they did not bring any novel information for mouse CSF-cNs.
2. *Pkd2l1-Cre::tdTomato* mice likely label other cell types in the gray matter as seen in Fig. 1B–D, 2A and Suppl Fig 1B–D, 2B–D, 3C. Nonspecific labeling of the mice is also reported in previous studies. The authors likely include those cell population in some analyses, but did not state how they are included or excluded from the analyses. Use of PKD2L1 antibody may be more precise for the counting.
3. The number of mice (and sections?) analyzed are not described in the histological analyses in Fig. 2A–D. Please specify them in the figure legend. Do the dots represent the count in each section?
4. Fig. 2A: it is weird that the % of the left and right lateral CSF-cNs are same in the pie chart.
5. The authors used 3 (or 6 or 3–6) weeks old mice (line 1174, 386, 911). Since the number of CSF-cNs are reported to decrease during the development (Ref 31), the rationale to use the present age in histological analyses is unclear. Do the authors aim to examine the properties of CSF-cNs of adult or juvenile mice? Are there any differences in the number between 3 and 6 weeks?
6. Which spinal levels of the cervical/thoracic/lumbar cord did the authors analyze for histology and electrophysiology? The rostro-caudal levels within the cervical, thoracic and lumbar cord would have quite different circuit and functions.
7. Fig. 2D: the definition of “the distance from the central canal” is unclear. Is the distance from the center or the edge of the central canal to the center of the soma? The reason why the distance was examined is also unclear. What does the

difference in the distance functionally means?

8. The authors state that the pH-dependent responses in Suppl Fig. 4 are mediated by PKD2L1 and ASICs in the manuscript. However, without experiments to block PKD2L1 or ASICs (by knockout or inhibitors), the responses are unknown to be mediated by those channels. Please clarify these points.

9. Why are the electrophysiological data of PKD2L1 and ASICs placed in supplemental but not in the main figures? Are they already reported somewhere? Please clarify it.

10. The authors state that the ventral CSF-cNs are immature compared to lateral CSF-cNs. Which population of CSF-cNs did the authors record in the electrophysiological experiments? Dorsal, lateral or ventral CSF-cNs?

11. The channel and receptor expressions are mostly discussed based on the electrophysiological and pharmacological data. It would be supportive if the authors add mRNA expression data of the proposed channels and receptors, e.g. by using the published single cell RNA-seq data.

12. It would be informative if the authors add the data of negative responses to ATP and mGluR in suppl figures.

13. It would be helpful for the readers if the authors summarize the similar and different electrophysiological properties across the spinal levels in a table or diagram in the last part of the figures.

Minor

14. The title may not appropriately convey the information of the present study: "functional" may be replaced with "electrophysiological", since no "functions" are identified in this study. "Intraspinal sensory neuronal population" is vague: this may be replaced with "cerebrospinal fluid-contacting neurons (in the mouse spinal cord)".

15. Please carefully check if suppl Fig. 1–4 are precisely indicated throughout the text and the figures.

16. The data using Chat-Cre mice (Suppl Fig. 3?4?) are not explained in Results.

17. Line 125 etc.: please specify what N = and n = means. The number of mice, slices, cells?

18. Line 145: please explain the abbreviation Vh in the first place.

19. Typos in line 255, 384, 1460.

20. Line 472–474: are the data of different firing pattern between C and L-CSF-cNs shown in figures and Results?

21. Line 635, 638: tdTomato mice are not "flex" mice.

22. Line 642: the data using AAV are not explained in Results.

23. Line 692, 695, 699: what is /ea?

24. Line 718, 719, 720: z-Stack > z-stack

25. Line 754: if wildtype mice were used, how were the CSF-cNs identified? Are the data of Pkd2l1-cre::tdTomato and WT mice combined in the figures? Please specify it.

26. Line 848, 878, 879, 1302, 1316, 1321 etc: Kainate > kainite

27. Line 1303, 1327: Glutamate > glutamate

28. Line 1460, 1568: PKD-tdTomato > PKD2L1-Cre::tdTomato

29. Please carefully present the asterisks and bars throughout the graphs and figure legends as follows:

- Line 1228: there are no * and ** in the graph (Fig. 4C).

- Line 1236–1237: there is only **** in the graph (Fig. 4D).

- Line 1258, 1283, 1293, 1335, 1341, 1406, 1416: please specify what *, **, *** in the figures (Fig. 5C, 6B, 6D, 7E1, 7E2, 9C, 9D) mean. It is unclear which groups do the bars and the asterisks in the graphs compare.

- Line 1322: no * in the graph (Fig. 7C).

- Line 1356: no ** in the graph (Fig. 8B).

- Line 1369, 1375 (Fig. 8D1, D2): * should be added in the graphs?

- Line 1440: *p<0.5? It is unclear which groups do the bars and * in the graph compare (Fig. 10C).

- Line 1450: no * and ** in the graph (Fig. 10D).

- Fig. 1D: it is unclear which groups do the p values compare. * is not consistent with the p values.

- Fig. 7B: it is unclear which groups does *** in the graph compare.

Reviewer #3

(Remarks to the Author)

Manuscript "An intraspinal sensory neuronal population with homogenous morphological and region specific functional properties in the mouse" by Crozat et al., using morphological and in vitro electrophysiological techniques characterized the properties of a kind of neurons that locate around the spinal central canal and their dendritic protrusions contact cerebrospinal fluid (CSF), so-called CSF-contacting neurons (CSF-cNs), in mice. They found that spinal CSF-cNs exhibit a conserved morphology across species and are uniform across SC segments in the mouse. They distribute densely along the entire spinal cord axis and are primarily located ventrally. Functionally, along the spinal cord, they share similar intrinsic and chemosensory properties with mouse medullar CSF-cNs. The neurons express NaV, KV, LVA and HVA Ca²⁺ channels. They also express functional classical inhibitory and excitatory ionotropic synaptic receptors. The authors conclude that the spinal CSF-cNs represent a morphologically homogeneous sensory neuronal population, but functionally different along the spinal cord axis. The study will for sure contribute to the better understanding of CSF-cN functions.

In general, each step of the experiments was carefully designed and conducted, and the results were objective, and the drafting of the manuscript is concise and thorough. The work with tissue clarity technique is beautiful. The work is of interest to others in the community. I don't have major comments but some minor ones.

1. Some abbreviations in the Abstract do not have a complete name, such as PKD2L1, TRP and ASICs. Please give these abbreviations complete names when first used anywhere in the manuscript even in the abstract.
2. Page 3, line 60, concerning the function in detecting spinal cord bending it is mainly observed in zebrafish. This needs to be stressed although the related references were given.
3. Page 4, line 83, "there are" should be "they are".
4. Page 5, lines 93-94, better to mention what animal species for this statement. The authors seemed to have included old articles (although some relatively new). Please update the references especially including those using primates.
5. Page 5, line 102, "(Fig.1B-D, Right ventral view)". Is 1D from ventral view or side view?
6. Page 5, lines 113-114, "A larger proportion of CSF-cNs is located in the ventral CC (Fig. 2A, Right; 2C) and neurons are closer to the CC in cervical and lumbar segments but further away in the thoracic one (Fig. 2D)". Isn't that in the lumbar segment the CSF-cNs further away from CC according to Fig. 2D (the blue dots)?
7. Page 6, line 128, "...compared to T- and C-CSF-cNs (Table 1)", should be "...compared to L- and C-CSF-cNs (Table 1)".
8. Page 6, line 132, "...depolarized CSF-cNs in anterior regions", better to use "rostral regions".
9. Page 7, lines 141-142: "the findings suggest that CSF-cNs share similar intrinsic properties but exhibit differential excitability along the spinal cord". Does this mean that "excitability" is not an "intrinsic property"?
10. Page 8, line 182, can the authors specify which subtypes of ASICs are expressed in mouse CSF-cNs? In lamprey it is primarily ASIC3.
11. Page 11, line 255, there is an extra "(".
12. Page 15, line 354, please specify "mACh-Rs.s", or it is one "s" extra.
13. In the Discussion, the authors need to compare the morphology and the distribution of CSF-cNs between different animal species, especially with those at a higher level, such as primates. There are such articles e.g.:
Liu et al. PKD2L1-expressing cerebrospinal fluid contacting neurons in spinal cords of rodents, carnivores, and primates. *Int. J. Mol. Sci.* 2023, 24, 13582. <https://doi.org/10.3390/ijms241713582>
Tonelli Gombalová Z et al. Majority of cerebrospinal fluid-contacting neurons in the spinal cord of c57bl/6n mice is present in ectopic position unlike in other studied experimental mice strains and mammalian species. *j. comp. neurol.* 2020, 528, 2523–2550. doi: 10.1002/cne.24909.
14. In connection with the last point, the authors also need to do a discussion from a translation point of view. Considering that in majority of humans the central canal is closed and there are no reports as to whether there are CSF-cNs in the human spinal cord, how can the results from the animal studies be translated to human?
15. Page 16, line 384, two refs. with a # 33 (one should be 32).
16. Page 26, line 610, "CS-cNs" should be "CSF-cNs".
17. Page 27, in the section "Histology and imaging", please specify what kind(s) of mice were used for imaging study.
18. Page 9, line 692, specify what "o/n" means.
19. Page 37, line 876-881, "For the agonists:...". This seems not a complete sentence.
20. Page 39, line 928-929, "Results were considered significant when *p<0.05, **p<0.01". Only *P < 0.05" is enough. The "**" is not needed here. Same principle applies to page 42, line 1003.
21. Page 48, line 1170, "thoracic segment (B)" should be "thoracic segment (C)".
22. In the figure legends, please specify what "N" respective "n" stands for. They are specified in Table 1 and 2, but not in other method part or figure legends.
23. Page 54, line 1304, "tilme" should be "time".
24. In general, the figure legends are too long. Is it possible to place the statistical details in the text, but not in the legends?
25. Figure 4, please place the voltage step gradients (each Vstep) used in Figure 4A, similar as Figure 5A.
26. Figure 7B, is "Control" same as "Glu"?
27. Figure 9, "+ CGP" in red color is a bit confusing. It is the cervical recording that shows red color. It is better to place "Baclofen" and "+ CGP" above the traces in A2 and A3 as well.
28. Figure 10, same as Figure 9, please do the same for "Oxo" and "+ Atr".

Version 2:

Reviewer comments:

Reviewer #1

(Remarks to the Author)

The authors have addressed most of my concerns regarding overstating observations/conclusions and the data is technically sound. However, I remain unconvinced that the results, consistent CSF-cn properties along the spinal cord, with only some variations in ionotropic current densities (the functional implications of which have not been addressed), constitute a significant advance bringing new biological insight or a conceptual advance.

Reviewer #2

(Remarks to the Author)

The manuscript has been improved well, but I still have some points that should be improved more as follows:

1. Abstract: the revised abstract is not clear regarding which part is the background, what is the research question, and which part is the result of this study. In addition, the summary of the findings, particularly their importance or novelty, are not well described. This issue relates to the comment of Reviewer #1-1.

2. Introduction: I feel that the research question and novel focusing points of this study in comparison to previous studies are not clearly described. Perhaps this study aims to “comprehensively” examine (and found) anatomical and electrophysiological properties of CSF-cNs, particularly focusing on “regional differences along the spinal levels” in mice. This also relates to the comment of Reviewer #1-1, and they should be clearly stated in the Introduction.

3. Page 7 line 149-: When comparing across spinal segments, T-CSF-cNs displayed a higher Rm and Cm, compared to C- and L-CSF-cNs while the membrane capacitance was the lowest at the thoracic level (Table 1).: I could not see a higher Rm and Cm of T-CSF-cNs in Table 1.

4. Supplemental Fig. 3: please indicate each gene name on the left side of the heatmap. For example, which rows of the heatmap represents ASIC1 and 2 (page 10, line 220)? Other genes are not clear as well.

5. Line 635, 638: tdTomato mice are not “flex” mice.

We agree with Reviewer #2, the tdTomato transgenic mouse model is not flex but floxed (flanked by LoxP sequences). The manuscript has been appropriately corrected.

tdTomato gene is not “floxed” in this mouse line (STOP cassette is floxed). The authors used CAG-loxp-STOP-loxp-tdTomato mice.

6. Line 1460, 1568: PKD-tdTomato > PKD2L1-Cre::tdTomato.

Detailed denomination of transgenic mouse models can be quite long. We therefore provided the extended model denomination in the Methods section “Animal models” (P33, L795-804) associated with shortened ones that were subsequently used in the rest of the manuscript for simplicity.

This is very confusing since “PKD-“ also includes genes of the polycystin family (e.g. Pkd1, Pkd2, etc.). In addition, “PKD-tdTomato” means that the authors use transgenic mice expressing tdTomato directly under the PKD promoter. I recommend using the term “Pkd2l1-Cre” (only the first letter of the gene name should be uppercase) throughout the manuscript. The authors may abbreviate it as Pkd2l1-Cre::tdTomato (or Pkd2l1-tdTomato, though I do not prefer the latter due to the above reason).

Reviewer #3

(Remarks to the Author)

The revised version addressed all my questions. Thus, I don't have any further comments for the revised manuscript.

Version 3:

Reviewer comments:

Reviewer #2

(Remarks to the Author)

The manuscript was well improved though there are typos and grammatical errors as followings. I have no further comments.

Line 41: SC segment > spinal cord segment

Line 95: CSF-cNs act as a novel sensory intrinsic to the CNS > sensory ??

Line 130: However, ; this can be deleted.

Line 463: from different??

Line 583: Pkd2l1&-Cre > Pkd2l1-Cre

Line 633: corelate > correlate

Line 830, 931 etc.: PKD-tdTomato > Pkd2l1-Cre::tdTomato

Point-to point response to Reviewers' comments

We would like to thank the reviewers for their careful reading of our initial manuscript and for their valuable comments. Please find below a point-to-point answer to the comments and requests. We hope to have adequately address the point raised and that in his revised version our manuscript will be suitable for publication.

We apologize for the delay in our response due to unforeseen reason.

Best regards,
Prof. N. Wanaverbecq

1) Reviewer #1 (Remarks to the Author):

Cerebrospinal fluid-contacting neurons (CSF-cNs) are intriguing neurons that are now well characterized in zebrafish where they play a role in sensory-motor control and a similar role has recently been identified in rodents(1). In the present manuscript, the authors have surveyed CsF-cNs along different segments of the spinal cord and provided a lot of convincing data on their intrinsic properties and transmitter sensitivity.

Previous work by the authors and others has shown that CSF-cNs are present at all levels of the spinal cord(2) (3) (4), express voltage-gated Na, K and Ca channels (5-8) are sensitive to pH (6, 9) through expression of ASIC (10) and PKD2L1 (2, 11) (12) and have functional glycine (9), GABAa (6, 9), GABAb (8) and cholinergic (7, 13) receptors. This manuscript reaffirms these previous findings along the spinal cord and as such the **findings can't be considered novel**.

Although we agree with Reviewer #1 that the data presented in our study have been reported to some extent for CSF-cNs. The studies mentioned were conducted in **different animal models** (rat, lamprey or mouse), **at different ages** as well as **in a specific region** along the central canal (medullar or one spinal cord segment).

However, to date there is no anatomical and functional data available that systematically characterize and compare the properties of CSF-cNs in the same animal model (*i.e.* the mouse) and along the central canal axis. Such a characterization has not been conducted prior and until now we did not know whether CSF-cNs show anatomical and electrophysiological properties that would be region-specific or not. **In this regard, our study is novel**. Prior to our study, authors in the field could only extrapolate about properties uniformity or not, now we have groundtruth data demonstrating it

Our claim is actually shared by Reviewers #2 and #3 who acknowledge that our study contributes to the better understanding of CSF-cN properties along the central canal, a crucial and necessary step for future characterization of their role as a novel sensory population intrinsic to the CNS.

Reviewer 2: "Overall, the present study provides some advances in the knowledge of spinal CSF-cNs, extending from the previous studies in rodents, especially on the electrophysiological responses at different spinal levels."

Reviewer 3: "The study will for sure contribute to the better understanding of CSF-cN functions."

Major comments:

- 1) In the abstract, on line 36, and introduction line 83, it is claimed that CSF-cN functional properties vary along the spinal cord.

It is unclear what the authors are referring to, **none of the subheadings of the results or discussion indicate differential function**. Contrary to this claim, subheadings in the results (line 119) and discussion (line 440) states '*Along the spinal central canal, CSF-cNs exhibit similar intrinsic properties*'.

In the revised version of the manuscript, we have checked for consistency between Title, Headings and Subheadings and our claims for different electrophysiological properties in CSF-cNs along the central canal axis. Wherever necessary, we have clarified this point and discussed it.

For simplicity, we indicate here the modified **Heading** and **Subheading**, the Page/Lines in the given section where claims for differences are mentioned and the conclusion of the corresponding section. We have also clarified this in the figure captions.

Title

“Cerebrospinal Fluid-contacting neurons are sensory neurons with uniform morphological and region-specific electrophysiological properties in the mouse spinal cord.”

Abstract

Abstract has been modified to comply with the guidelines and the 150 words limits.

P2, L27-40:

“Cerebrospinal Fluid-contacting neurons (CSF-cNs) are GABAergic neurons found along the medullo-spinal central canal in vertebrates, but their properties in mice remain largely uncharacterized. They express Polycystin Kidney Disease 2-Like 1 channels (PKD2L1), members of the Transient Receptor Potential superfamily, with sensory properties.

*CSF-cNs are distributed throughout the spinal cord with a uniform morphology and a primarily ventral localization. **CSF-cNs exhibit region-specific intrinsic properties, expression of voltage-dependent and ligand-gated conductances and detect variation in extracellular pH through activation of PKD2L1 and Acid-sensing Ion Channels.** They possess GABA_B and muscarinic receptors, not glutamatergic metabotropic ones, to modulate Ca²⁺ channels.*

***CSF-cNs are sensory neurons with a uniform morphology and electro-physiological properties specific to the network they are inserted in.** The future challenge in the field, will be to demonstrate CSF-cNs integrate circulating signals along the central canal while differently integrate synaptic inputs to modulate body function through specific local spinal network.”*

Introduction

P4, L86-89:

“Electrophysiological recordings show that, although they share similar sensory functions and the same expression of ionic conductances and ligand-gated receptors, they present differences in their passive and firing properties as well as in the expression density of the identified channels and receptors.”

P4-5, L91-95:

“On the other hand, they exhibit differential electrophysiological features in the SC segments and these region-specific properties might serve the specific role they play in a given spinal network. Anatomical and functional evidence suggest that CSF-cNs serve as a novel sensory or interoceptive system intrinsic to the CNS.”

Results

P6, L140, 141:

Heading: “Spinal CSF-cNs intrinsic and sensory properties along the central canal axis”

Subheading: “CSF-cNs have region-specific passive properties and firing patterns”

P7, L149-150; P7-8, L165-167;

Conclusion of this section (P10, L221-225):

“Overall, our findings suggest that CSF-cNs are highly resistive neurons showing passive properties as well as firing patterns that differ along the cc. They further exhibit shared chemosensory functions by responding to variations in extracellular pH. These differences were observed between CSF-cNs recorded in different segments but not associated to a specific localization around the cc.”

P10, L227-228:

Heading: *“Spinal CSF-cNs express functional sodium, potassium and calcium voltage-dependent channels”*

P12, L285-286; P13, L297-299

Conclusion of this section (P13, L300-305):

“Spinal CSF-cNs express functional NaV, along with KV of the delayed rectifier- (IKD) and A-type (IA), which present similarly expression densities along the cc axis and would be responsible for the AP generation. Additionally, all spinal CSF-cNs express both HVA and LVA CaV, with higher current densities in T-CSF-cNs. These findings are supported by a recent transcriptomic analysis of gene expression in spinal CSF-cNs¹¹ (see Supplementary Fig. 3B-D).”

P13, L307:

Heading: *“CSF-cNs express ionotropic synaptic receptors with a density that differs along the cc”*

P14, 321-324; P14, L335-337; P14-15 L340-343; P15, L351-352

P15-16, L365-367:

Conclusion of this section

“To summarize, our data show that all spinal CSF-cNs express functional GABAA, glycine, AMPA/kainate glutamatergic, and nACh receptors but at different densities depending on the spinal cord segment considered. Activation of AMPA/kainate and nACh receptors modulates CSF-cN excitability and can trigger AP firing.”

Discussion

P19, L461-463

P19-20, L464-468:

*“Overall, spinal CSF-cNs represent a morphologically uniform neuronal population capable of detecting sensory signals that exhibits differential electrophysiological properties along the SC axis potentially to modulate the specific spinal network they are inserted in (**Fig. 13**). Although differences can be observed between CSF-cNs from different segments, our study does not point to an organization of CSF-cNs in specific cluster around the cc.”*

We also provide a summary figure illustrating the electrophysiological differences along the cc.

P23, L554-555

Heading: *“Along the spinal central canal, CSF-cNs exhibit region-specific electrophysiological properties”*

P24-25, L588-592; P25, 603-605; P27, L64-648; P27, L658-662

P29, L694:

Heading: *“CSF-cNs show a differential excitatory/inhibitory drive along the central canal axis.”*

P29, L705-706; P30, L713-717; P30, L719-722

- 2) *The authors also show that all identified synaptic receptors are present in each subdivision, with only some variation in the current density for GABA_A and iGluR observed. However, the **functional relevance of this has not been addressed**. The authors should reconsider the claim of functional properties varying along the spinal cord.*

In our study, we have analyzed the functional expression of 'classical' synaptic receptors by a pressure application approach of specific agonists and reproduced the recordings in the presence of selective antagonists.

To take into account possible differences in cell size (*i.e.* membrane capacitance) that would have an effect on the current amplitude and mask possible differences in the density of receptor's expression, we have normalized the current amplitude to the membrane capacitance and expressed data as current density in all our analyses. These analyses demonstrate statistical differences for the current densities of glycine- and GABA_A-mediated responses (**Figure 8**, in our revised manuscript) as well as for the glutamate (AMPA/kainate)-mediated ones (**Figure 9**, in our revised manuscript). For the glutamate-mediated responses, this difference further indicates a region-specific modulation of CSF-cN excitability by glutamate since the lower current density observed for glutamate (AMPA/kainate)-mediated response underlies the lower glutamate-mediated excitability in the lumbar segment (**Figure 9D-E**, in our manuscript).

We agree that we report differences in the electrophysiological properties (current densities), not the functional consequences, however having differential receptor expression, *i.e.* current amplitude, would suggest differential function. Our results therefore indicate that CSF-cNs along the central canal axis would be differentially modulated by the specific activation of synaptic receptor subsets and suggest that the excitatory and inhibitory drive onto CSF-cNs differs along the central canal axis (*i.e.* stronger inhibitory vs. excitatory drive in lumbar CSF-cNs)..

This opens future lines of research to further characterize the underlying connectivity leading to a differential integration within segment-specific local networks. We are currently addressing these aspects in dedicated morphological and functional studies.

The same analysis was carried out for voltage-dependent conductances and we also observed significant differences in the expression of Ca_v with thoracic CSF-cNs exhibiting larger calcium current densities and potentially calcium signaling (**Figure 7**, in our revised manuscript). An aspect that also needs to be further investigated.

Considering that in mammals, and actually in most species studied so far, CSF-cNs remain largely under characterized at the functional level, any new piece of evidence contributes to their better understanding.

We have stressed this point in the Discussion of the revised manuscript. It now reads:

P30, L713-726:

“Our results further indicate differences in the current densities of glycine and GABA_A receptors-mediated responses that is highest in thoracic CSF-cNs. We also report in lumbar CSF-cNs lower current densities for glutamate (AMPA/kainate)-mediated responses underlying the observed lower excitatory action of AMPA/kainate receptor activation. Due to the recording configuration (whole-cell patch-clamp and artificial intracellular chloride concentration), it is difficult to discuss the functional consequence of the inhibitory signaling onto spinal CSF-cNs (see also Riondel and Colleagues³⁶). Nevertheless, altogether, these results might imply that the excitatory and inhibitory drives onto CSF-cNs differ along the cc axis and that CSF-cNs would be differentially modulated through synaptic inputs within local spinal networks. The functional consequence of this differential excitatory/inhibitory drives is unknown but opens future lines of research where the underlying connectivity leading to a differential integration within segment specific local networks needs to be characterized. We are currently addressing these aspects in dedicated morphological and functional studies.”

P30, L731-733:

“However, the effects induced by the activation of these two receptors are different suggesting that CSF-cNs might integrate or code differentially the synaptic inputs they receive depending on the network they are inserted in.”

- 3) *Due to the technical issues (low resolution) with light sheet imaging I am not sure what Figure 1 adds to the manuscript that isn't conveyed by Figure 2.*

Although we agree with Reviewer #1's comment that the lack of cellular/subcellular resolution prevents a detailed analysis of CSF-cN morphology or distribution, we believe **Figure 1** plays an important role in demonstrating the mesoscopic distribution of CSF-cNs in the intact mouse CNS. While several studies have shown CSF-cNs at different spinal levels in the mouse SC, a comprehensive view of the entire CSF-cN system is lacking in the literature. **Figure 1**, in its current form, summarizes this data, illustrating that the CSF-cN system extends from rhombencephalic levels to the lumbosacral spinal cord within the mouse CNS. Indeed, the quantification of **Figure 2** is more accurate since we are able to differentiate individual cells, however, **Figure 1** supports the conclusion drawn from **Figure 2** since to infer the distribution of CSF-cNs in the whole CNS by routine histology techniques would be a much more time-consuming task. Moreover, we are currently implementing other clearing techniques with improved resolution (due to the absence of tissue shrinkage) that will enable the use of light-sheet microscopy to study CSF-cN distribution and connectivity. Notably, these experiments are ongoing and beyond the scope of the current manuscript.

- 4) *Statistical analyses appear appropriate.*

We thank Reviewer #1 to acknowledge the well conducted statistical analysis.

Minor:

- 5) *page 3 line 53, you specify rats but don't make it clear that refs 19-21 are from mice. You could possibly change 'rats' to 'rodents' otherwise you should be careful of the species.*

Modified. The sentence has been changed to:

P3, L53-55:

“In rodents, CSF-cNs exhibit spontaneous action potential (AP) firing, mediated by sodium and potassium voltage-dependent channels [9], and they also express voltage-dependent calcium channels (CaV 2.2 and CaV 3.1-3.3) [22,23,24].”

and the reference 9 was moved after “*voltage-dependent channels*”.

- 6) *Line 63 'regrouped' should just be 'grouped'.*

Correction made (L67)

- 7) *Also, it is not evident what is meant by **immature vs mature phenotypes**.*

Classically, the level of neuronal maturity is based on an expression switch from immaturity markers (doublecortin (DCX), HuC/D, PSA-NCAM and Nkx6.1 and Nkx2.2) during development and postnatal period to those of maturity (e.g. neuronal nuclear protein, NeuN). This maturation is accompanied by functional changes such as action potential firing pattern and insertion in synaptic networks.

Our claim is based on previous reports from different groups (Marichel *et al.*, J.Neurosci. 2009; Reali *et al.*, J.Physiol. 2011; Petracca *et al.*, 2016, DiBella *et al.*, 2019) in the field as well as publications from our group (Orts Del'Immagine *et al.*, 2016 & 2017) where it was shown that in CSF-cNs DCX, PSA-NCAM and HuC/D expression is observed along with low or no expression of NeuN. We also reported that CSF-cNs would maintain Nkx6.1 and DCX expression in mice as old as year of age (Orts Del'Immagine *et al.*, Neurosci. 2017). In parallel, there is functional evidence that subsets of CSF-cNs exhibit Calcium-mediated or single-

sodium spike firing in juvenile rats (Marichal *et al.*, 2009; see comments below) as well as depolarizing effect of GABAergic signalling in turtle (Reali *et al.*, J.Physiol. 2011) also observed in 8 weeks-old mice (Riondel *et al.*, 2024).

Further, in our previous reports (Orts Del'Imagine *et al.*, 2016 & 2017; Riondel *et al.*, 2024), we also indicated that, in contrast to what is observed in juvenile animals, although different CSF-cN phenotypes (*i.e.* immature vs. mature) can be observed in older mice, these neurons are not organized in specific clusters but rather distributed around the central canal.

Altogether, these features indicate that in older mice a subset of CSF-cNs present features of immature neurons. In the revised manuscript, we have accordingly edited the text to clarify this point:

P3-4, L64-74:

“In zebra fish larvae and juvenile rodents, morphological [6,8,21,32] and phenotypical [8,32–34] differences suggest that CSF-cNs are grouped in two subpopulations. In the cc ventral region, they exhibit immature phenotypes (expression of doublecortine, PSA-NCAM and homeoboxes Nkx2.2 and 6.1) [5,8,21,32,33] and fire single AP [8,32], a feature often associated to immaturity [35]. Dorso-lateral CSF-cNs have a more mature phenotype (expression on the neuronal nuclear protein, NeuN) and tonic AP discharge [8,21,32,33]. In older mice, CSF-cNs still show the expression of these immaturity markers, two distinct AP firing patterns and, in lumbar segments, two subpopulations can be distinguished based on their responses to GABAergic signaling, either inducing depolarization or hyperpolarization [36]. Nevertheless, and in contrast to what has been reported for juvenile mice, there is no correlation between CSF-cN phenotypical and firing properties and their localization around the cc.”

- 8) Why is tonic discharge considered mature and single AP firing immature? Perhaps you should be more specific on what is different without referring to the maturation stage of a neuron.**

The single-spike profile with neurons emitting one AP in response to depolarization, but without being able to sustain regular firing is often associated with immaturity and low expression of voltage-gated sodium channels underlying action potential firing. Articles from the Russo group (Marichel *et al.*, J.Neurosci. 2009; Reali *et al.*, J.Physiol. 2011) indeed suggest that single-spiking corresponds to an immature stage. To support their claims, they cite publications from Spitzer and collaborators (Spitze & Ribera, 1998 and Ben-Ari & Spitzer, Trends Neurosci 2010), where the claim that single-spiking corresponds to an immature neuronal stage.

For CSF-cNs the single-spike firing profile has been classically used to define their immature *i.e.* low maturity state (Marichal *et al.*, J.Neurosci. 2009; Reali *et al.*, J.Physiol. 2011; Petracca *et al.*, 2016, DiBella *et al.*, 2019). Based on these reports and the ‘association’ of single spiking with immaturity, we suggest, in agreement with earlier reports, that a subset of CSF-cNs would have an immature phenotype.

Nevertheless, we discussed this claim in the revised version of the manuscript.

P3-4, L64-68:

“In zebra fish larvae and juvenile rodents, morphological [6,8,21,32] and phenotypical [8,32–34] differences suggest that CSF-cNs are grouped in two subpopulations. In the cc ventral region, they exhibit immature phenotypes (expression of doublecortine, PSA-NCAM and homeoboxes Nkx2.2 and 6.1) [5,8,21,32,33] and fire single AP [8,32], a feature often associated to immaturity [35].”

P25-26, L603-615:

“We also observed that C- and L-CSF-cNs exhibit mainly tonic firing while thoracic neurons mostly show single-spiking pattern. The single-spike profile with neurons emitting one AP in response to depolarization, but without being able to sustain regular firing, is often associated with immaturity and low expression of voltage-gated sodium channels underlying action

potential firing [57,58]. For CSF-cNs, the single-spike firing profile has been classically used to define an immature i.e. low maturity state [8,9,32,42]. However, to date there is no evidence in CSF-cNs of the correlation of single-spike firing and the expression of immaturity marker. One would need to conduct dedicated studies to address this point. When referring to single-spike discharge, alike other reports^{8,9}, we would rather consider a discharge pattern with a 'depolarization block' where membrane potential depolarization impedes the generation of AP (in Fig. 3A, compare depolarization induced by +10 and +30 pA current injections) presumably due to a lower density of sodium voltage-dependent channels (see Marichal and collaborators [9])."

9) line 71 'finally'... you have an additional point after the 'finally' statement. Reword. The sentence has been modified as follow.

P4, L77-85;

*"In zebrafish larvae, CSF-cNs selectively activate motor neurons and interneurons, influencing swimming behavior [29,37,38]. In mice, they play a role in controlling posture, balance [16], and locomotion [15]. **Further**, in zebrafish larvae, they appear to respond to bacterial toxins, suggesting a potential role in the body's immune defense [39]. **Finally**, it was recently shown that, following SC injury, CSF-cN constitutive activation through κ -opioid signaling is halted leading to disinhibition of ependymal cell proliferation to promote scar formation [11] and would even have regenerative properties [40,41]."*

2) Reviewer #2 (Remarks to the Author):

CSF-contacting neurons (CSF-cNs) are functionally enigmatic sensory cells facing the central canal of the spinal cord. In this study, Crozat and colleagues examined the morphological and electrophysiological properties of CSF-cNs in the mouse spinal cord by comparing them at the cervical, thoracic, and lumbar levels.

The properties were consistent across the spinal levels and aligned with previous observations in spinal and medullar CSF-cNs in mice and juvenile rats (Ref 9, 12, 19, 22), whereas variations in cell density and certain electrophysiological responses were noted between spinal levels.

The functional and biological significance of those differences are unclear and not clearly discussed in this study.

Overall, the present study provides some advances in the knowledge of spinal CSF-cNs, extending from the previous studies in rodents, especially on the electrophysiological responses at different spinal levels.

The experiments were well conducted to obtain the basic electrophysiological properties of CSF-cNs. The manuscript is well organized and the data are interpreted well.

Specific comments are as follows:

- 1) *The authors examined the morphological properties of CSF-cNs in clearing tissues and show beautiful images in Fig. 1 and Suppl Fig 1–2. However, unfortunately, they did not bring any novel information for mouse CSF-cNs.*

We acknowledge that the limited resolution of light-sheet microscopy, combined with the small size of CSF-cNs and their axons, limits detailed analysis of CSF-cN morphology and distribution using this technique. However, the population analysis of CSF-cN distribution in Figure 1 should be understood at a mesoscopic level.

Our main aim with this figure was to present the overall distribution of CSF-cNs in the intact mouse CNS, rather than providing detailed quantification or morphological analysis. Currently, our lab is making all possible efforts to implement clearing techniques and light-sheet microscopy acquisitions, combined with registered confocal acquisitions, to finely characterize the connectivity patterns and morphological features of CSF-cNs at spinal and supraspinal levels. These results, however, fall beyond the scope of the current manuscript.

- 2) *Pkd2l1-Cre::tdTomato mice likely label other cell types in the gray matter as seen in Fig. 1B–D, 2A and Suppl Fig 1B–D, 2B–D, 3C. Nonspecific labeling of the mice is also reported in previous studies. The authors likely include those cell populations in some analyses, but did not state how they are included or excluded from the analyses. Use of PKD2L1 antibody may be more precise for the counting.*

We appreciate the reviewer's comment. As observed both in our study and in others (Nakamura et al., 2023; Yu et al., 2024), the PKD-tdTomato mouse also labels cell types other than CSF-cNs, likely oligodendrocytes (Nakamura et al., 2023), within the central nervous system (CNS). We conducted a systematic analysis of these labeled cells throughout the entire mouse CNS and found: *i)* These cells are absent at ages younger than 4 weeks, and *ii)* at around 4-6 weeks, when these cells do appear, their fluorescence levels are significantly lower than those of actual CSF-cNs (as confirmed by PKD2L1 immunoreactivity). See the **Figure 1** below (Cervical SC, 6 weeks; Magenta - tdTomato; Green - α PKD2L1). Additionally, CSF-cNs near the central canal (CC) can often be morphologically identified because of their protrusions visible into the CC lumen. Based on this, we visually defined CSF-cNs as cells where the protrusion can be identified, along with those showing higher fluorescence values where the protrusion cannot be seen. It is important to note that in Figures 1 and 2 in the manuscript, all images correspond to 3-week-old mice, so there is no ectopic tdTomato expression in non-CSF-cN cells. The "ectopic" appearance of CSF-cNs is supported by our

previous findings, where PKD2L1 expression was observed in cells distant from the cc (Jurčić N *et al.*, 2021).

We also acknowledge the reviewer's suggestion to use anti-PKD2L1 antibodies to count CSF-cNs instead of relying on tdTomato fluorescence in the PKD2L1-tdTomato mouse. Unfortunately, there is no simple solution to this issue. While CSF-cNs are defined by PKD2L1 expression, immunoreactivity is often considered a semiquantitative technique, as it depends on factors such as antigen expression levels, fixation conditions, and antibody accessibility to specific epitopes. Antibody binding can even be affected by the molecular environment, especially for membrane-inserted ion channels (Lorincz and Nusser, 2008; Ramirez-Franco *et al.*, 2024). Indeed, although rare, we have also observed some morphologically identified CSF-cNs where PKD2L1 immunolabeling is barely detectable (see **Figure 2** below; arrow). For this reason, we choose to quantify CSF-cNs based on tdTomato fluorescence in the 3-6 weeks PKD-tdTomato mice.

Figure 1: Cervical SC, 6 weeks. Magenta-tdTomato; Green- α PKD2L1

Figure 2: Cervical SC, 6 weeks. Magenta-tdTomato; Green- α PKD2L1; scale bar=25µm

- 3) *The number of mice (and sections?) analyzed are not described in the histological analyses in Fig. 2A–D.*

Please refer to the Method section “Cell counting and distribution of CSF-cNs in confocal images” for this point as we specify:

P44, L1072-1075

“Five 6 weeks old mice were used for CSF-cNs quantification in confocal slices. 10 fields of view (FOV) were acquired per animal and SC segment (total 30 FOV/animal). To count CSF-cNs in confocal coronal slices, Z-stacks of 10-20 images were acquired. Counting was aided by custom FIJI macros available upon reasonable request.”

Please specify them in the figure legend. Do the dots represent the count in each section?

We thank the reviewer since adding this info to the figure legends will surely help to improve the clarity of the figure. In Figure 2B and 2C each one of the dots represents a slice. In Figure 2D, each dot represents an individual cell. This information has been added to the Figure legend.

P54, L1417-1418:

“Dots represent each slice and each individual cell in Figure 2B, 2C and Figure 2D, respectively.”

- 4) *Fig. 2A: it is weird that the % of the left and right lateral CSF-cNs are the same in the pie chart.*

We thank the reviewer for this comment. To classify CSF-cNs as ventral, dorsal, or lateral, a custom script in FIJI (ImageJ) was created. The user manually defines the dorsal and ventral regions, and the remaining area in the field of view is automatically classified as lateral. Since we observed no evidence of lateralization of CSF-cNs along the rostrocaudal axis of the spinal cord, the remaining percentage of CSF-cNs was divided by two, and the result was assigned to each of the lateral quadrants (Left and Right). Moreover, since it is impossible to preserve left/right identity when working with floating sections, this information would be meaningless in our preparation. This is now explained in the methods section and we added a sentence in the related method section.

In the revised manuscript, it now reads:

P44-45, L1077-1088:

“Once all the cells were marked, the user was asked to define manually the dorsal and ventral regions on the CC (ventral fiber bundles served as bona fide indicators of SC orientation) and the remaining area in the field of view was automatically classified as lateral. Then, the number of cells in each quadrant and their percentages were automatically calculated, and the cell densities/10 μ m of SC were expressed as $(\text{Total number of CSF-cNs in the field of view} \times 10)/(\text{Number of optical slices} \times \text{Voxel depth})$. Since we did not observe evidence for CSF-cN lateralization along the rostrocaudal axis of the SC, once the percentage obtained for CSF-cNs in the dorsal and ventral quadrants, the remaining ones was divided by two, and the result was assigned to each of the lateral quadrants (Left and Right). Moreover, for our study, we have been working with floating sections and it was therefore not possible to preserve left/right identity.”

- 5) *The authors used 3 (or 6 or 3–6) weeks old mice (line 1174, 386, 911). Since the number of CSF-cNs are reported to decrease during the development (Ref 31), the **rationale to use the present age in histological analyses is unclear.***

Do the authors aim to examine the properties of CSF-cNs of adult or juvenile mice?
Are there any differences in the number between 3 and 6 weeks?

We do not aim to examine the properties of juvenile and adult mice and as a note mice are not considered to be adult at 6 weeks but rather from 3-6 months:

<https://www.jax.org/research-and-faculty/research-labs/the-harrison-lab/gerontology/life-span-as-a-biomarker#>

This is also the reason why we avoid using the term ‘adult’ in the description of our results. The **rationale** to use 3 and 6 week-old mice was first to parallel our electrophysiological experiments (**Figure 1** and **Figure 2** of the manuscript) where recordings were performed in this age range. Second, 6 week-old mice were also tested to assess potential differences in the anatomical properties. However, our analysis did not highlight such differences. In the revised version of our manuscript, we now clarify this point.

We added a comment in the Results section:

P6, L129-131:

“However, since our electrophysiological analyses were carried out in mice aged from 3-6 weeks, 6 week-old mice were also tested to assess potential differences in the anatomical properties but it did not highlight such differences.”

We modified the text earlier referred at line 383

P20, L475-478:

“The SC length 3 weeks-old mouse is ~2.5 cm (25,000 μ m) and based on our confocal images the density of CSF-cNs is ~14 cells/10 μ m. We therefore estimate that ~35,000 CSF-cNs would be present in the whole SC (52,500 for a 3.75 cm SC in 6 week-old mice).”

We modified the text earlier referred at line 911

P44, L1073

“Five 3 weeks mice were used for CSF-cNs quantification in confocal slices. 10 fields of view (FOV) were acquired per animal and SC segment (total 30 FOV/animal).”

We modified the text earlier referred at line 1174

Here we present data from a 3 week-old mice

P49, L1405 - Legend of Figure 2A:

- 6) Which spinal levels of the cervical/thoracic/lumbar cord did the authors analyze for histology and electrophysiology? The rostro-caudal levels within the cervical, thoracic and lumbar cord would have quite different circuits and functions.**

For imaging, we typically focus on the medial levels of each region, deliberately avoiding the most caudal and rostral poles of each segment. A detailed analysis of the connectivity patterns of CSF-cNs at various levels and sublevels of the spinal cord, as well as in the rhombencephalic regions, is currently underway in the lab, and these results fall outside the scope of this manuscript.

For electrophysiological studies, the recordings were carried out on acute slices obtained from a given region (cervical, thoracic and lumbar) without distinction of the rostro-caudal level (avoiding the caudal and rostral poles). Although we agree with Reviewer #2’s comment about the difference in the circuits and functions at each level, we did not observe differences in the CSF-cN electrophysiological properties within a region.

All our data are presented as boxplots with whiskers and we also illustrate single data points for each recorded cell. Since, the data distribution appears homogeneous (*i.e.* without data clusters) in a given segment, we are confident that CSF-cN properties are homogenous within a region. We clarify this point in the revised version of the manuscript.

P34, L831-833:

“SC were cut in three segments corresponding to cervical, thoracic and lumbar regions. We typically focus on the medial levels of each region, deliberately avoiding the most caudal and rostral poles of each segment.”

P39, L938-940:

“Lumbar, thoracic, or cervical SC acute slices (not distinguishing the rostro-caudal position) prepared from wild type or PKD-tdTomato mice were transferred in a recording chamber...#

- 7) *Fig. 2D: the definition of “the distance from the central canal” is unclear. Is the distance from the center or the edge of the central canal to the center of the soma? The reason why the distance was examined is also unclear. What does the difference in the distance functionally mean?*

Definition: We thank the Reviewer #2 for this comment since the accurate definition of this distance is important. In the revised manuscript this part of the Methods section now reads as follow

P44, L1078-1094:

“In brief, the user was prompted to click in the center of CSF-cNs on a Maximal Projection image while being able to navigate through the Z-stack in order to accurately resolve closely packed cells. [...] Afterwards the DAPI signal was automatically thresholded to create the contour of the cc, and the Euclidean distance (in microns) of each cell to this contour was calculated using the “Exact Euclidean Distance Transform (3D)” function in FIJI [57] along with the pixel size of the image.”

In brief, we use the DAPI signal of ependymal cells to automatically define the central canal, and the distance measured is from CSF-cN somatic center to the inner edge of the ependymal cell layer (**Figure 3**).

From the original image (1) the CSF-cN channel (2) is used to quantify the center of CSF-cNs by clicking on the cells (3). In parallel, the DAPI channel (4) is used for thresholding (5) and the line corresponding to the ependymal cell layer (6) is used to quantify the euclidean distance of each point (CSF-cNs) to the line (Central Canal).

This is based on:

Distance Transform 3D plugin (FIJI):

<https://imagej.net/plugins/distance-transform-3d>

Points to curve distance FIJI macro:

<https://gist.github.com/lacan/74f550a21ea97f46c74f1a110583586d>

Points to curve distance FIJI macro:

<https://gist.github.com/lacan/74f550a21ea97f46c74f1a110583586d>

Figure 3: Analysis routine used to define CSF-cN distance to the CC

Reason: we performed this analysis for the **systematic** follow up of the article previously published: *Evidence for PKD2L1-positive neurons distant from the central canal in the ventromedial spinal cord and medulla of the adult mouse*. Jurčić N et al., Eur J Neurosci. 2021 Aug;54(3):4781-4803. PMID: 34097332.

- 8) *The authors state that the pH-dependent responses in Suppl Fig. 4 are mediated by PKD2L1 and ASICs in the manuscript. However, without experiments to block PKD2L1 or ASICs (by knockout or inhibitors), the responses are unknown to be mediated by those channels. Please clarify these points.*

In our previous reports obtained from medullar CSF-cNs, we extensively demonstrated that they selectively express the PKD2L1 channel (comparison between wild type and PKD2L1-KO models: Orts Del'Immagine et al., Neurophram 2016) as well as homomers ASIC1a or heteromers ASIC1a/2b based on the sensitivity to *Psalmotioxin* (Orts Del'Immagine et al., J.Physiol. 2012; Orts Del'Immagine et al., Neurophram 2016).

In the present study, we did not carry out such an analysis. However, based on the selective Cre-dependent expression of tdTomato protein under the promoter of PKD2L1, the characteristic morphology of cells recorded around the cc and the unique characteristics (amplitude, open probability and similar pH sensitivity) of the unitary currents, we can safely claim it is carried by the PKD2L1 channel. Further, only neurons around the cc with the expected morphology (confirmed by the cell dialysis with AlexaFluor 488 a cytosolic fluorescent probe added to the intracellular pipette solution).

Regarding ASIC expression and activation, the current kinetics are also similar to those demonstrated in the above mentioned studies and here we can also claim they are carried by ASICs. Finally, in a recent study in mice, Yue and Collaborators (Nature, 2024) conducted a transcriptomic analysis of spinal CSF-cNs data and their data support our claim (see Response to Reviewer #3, Point 10; below). We now added a supplementary Figure on gene expression in CSF-cNs adapted from the mentioned publication.

In the new version of the manuscript, we now clarify this point:

Result/CSF-cN modulation by extracellular pH through the activity of PKD2L1 and ASICs.

P9-10, L216-220:

*“The current properties observed in our study are in line with those reported in previous reports^{25–27} and suggest that they are mediated by PKD2L1 channels and ASIC. An observation that is further supported by the transcriptomic analysis recently carried out by Yue and Collaborators¹¹ where gene for PKD2L1 and ASICs 1 and 2 can be found (see **Supplementary Figure 3A** and Discussion).”*

Discussion/Along the spinal central canal, CSF-cNs exhibit similar intrinsic properties

P24, L566-575:

*“Although we did not perform recordings in CSF-cNs obtained from PKD2L1 knock out mice, we are confident that the unitary current recorded in spinal CSF-cNs is carried by PKD2L1. The unitary current was systematically and only observed in the recorded neurons that exhibited the characteristic morphology of CSF-cNs visualized from the expression of tdTomato fluorescent and confirmed using soluble fluorescent markers (Alexa488 or 594 hydrazide for PKD-tdTomato or wild type mice, respectively) added to the recording intracellular solution mice. Further, it exhibits the characteristic electrophysiological properties (large unitary current amplitude and low open probability) as well as the pH-sensitivity also reported in medullar CSF-cNs (and see **Supplementary Fig. 3A**).*

P24, L578-583:

[...]“We did not characterize the ASIC isoform expressed in spinal CSF-cNs using pharmacological tools. However, based on the similarities of the current kinetics between our recordings and those reported for medullar CSF-cNs(13,54), one can suggest that spinal CSF-cNs would also express ASIC1a homomers or ASIC1a/2b heteromers. This assumption is further supported by the recent report by Yue and collaborators(11) (see also **Supplementary Fig.3A**).”

- 9) *Why are the electrophysiological data of PKD2L1 and ASICs placed in supplemental but not in the main figures? Are they already reported somewhere? Please clarify it.*

The analysis of CSF-cNs sensitivity was reported in the mouse by our group for medullar CSF-cNs but so far not along the spinal cord cc axis. The reason for presenting these data in a supplementary Figure was only based on the figure number limits defined in the manuscript guidelines. In agreement with the Editor this Figure can be included in the main manuscript as **Figure 4**.

- 10) *The authors state that the ventral CSF-cNs are immature compared to lateral CSF-cNs. Which population of CSF-cNs did the authors record in the electrophysiological experiments? Dorsal, lateral or ventral CSF-cNs?*

The claim for the presence of CSF-cNs showing different levels of maturity between the dorso-lateral and ventral regions around the cc is based on the literature (see above comment to Reviewer #1 Point 7 and 8) and dreferrs to postnatal mice. **It was indeed shown in postnatal mice** that ventral CSF-cNs would express immaturity markers and exhibit single-spike firing, two features related to neuronal immaturity. In contrast, dorso-lateral CSF-cNs express maturity markers and present tonic firing (Petracca *et al.*, 2016, DiBella *et al.*, 2019).

In our previous reports, we also analyzed CSF-cN maturity profile at the phenotypical and functional level (Orts Del'Immagine *et al.*, 2016 & 2017; Riondel *et al.*, 2024) **but in older mice**. We show that although two subpopulations (immature vs. mature) of CSF-cNs can be found in older mice based on protein markers, these two groups would not cluster in specific regions around the cc.

In the present study, we recorded neurons around the cc and in the different quadrants. However, although we did observe differences in the firing pattern, channel and receptor expression between CSF-cNs from different segments we could not correlate these differences to a given localization around the cc.

This was stated in the original manuscript and we added a comment in the revised mansucript at the beginning of the Result section

Introduction

P4, L72-74:

“Nevertheless, and in contrast to what has been reported for juvenile mice, there is no correlation between CSF-cN phenotypical and firing properties and their localization around the cc.”

Results

P7, L151-156:

“Here and in the following experiments, recordings have been conducted on CSF-cNs localized in the different quadrant (dorsal, lateral and ventral) around the cc to test whether we could reveal localization-specific properties. However, our data do not indicate that CSF-cNs would group functional clusters. We therefore pool together our data according to their presence in a given segment, not a specific localization around the cc.”

P8, L168-169:

“Nevertheless, in contrast with the situation observed in juvenile rodents, we could not associate a specific discharge patterns with CSF-cN localization around cc.”

P10, L223-225:

“These differences were observed between CSF-cNs recorded in different segments but not associated to a specific localization around the cc.”

Discussion

P19-20, L464-468:

“Overall, spinal CSF-cNs represent a morphologically uniform neuronal population capable of detecting sensory signals that exhibits differential electrophysiological properties along the SC axis potentially to modulate the specific spinal network they are inserted in (Fig. 13). Although differences can be observed between CSF-cNs from different segments, our study does not point to a an organization of CSF-cNs in specific cluster around the cc.”

P22, L520-522:

“Nevertheless in older mice unlike in zebrafish or postnatal mice, medullo-spinal CSF-cN subpopulations do not cluster specifically around the cc [7,13,31], possibly due to developmental reorganization.”

P25, L599-601:

“However, in older mice and in contrast to the data obtained in zebrafish or postnatal mice8, medullo-spinal CSF-cNs are not regrouped in specific clusters but rather distributed around the cc. One can argue that during development and maturation CSF-cNs may redistribute around the cc or might undergo developmental process to change their phenotypical and functional properties. Dedicated developmental studies are required to address this specific point.”

- 11) The channel and receptor expressions are mostly discussed based on the electrophysiological and pharmacological data. **It would be supportive if the authors add mRNA expression data** of the proposed channels and receptors, e.g. by using the published single cell RNA-seq data.**

Classically, the characterization for the expression of sets of metabotropic and ligand-gated receptors as well as voltage-dependent channels is conducted using electrophysiological recordings with specific activation protocol and combined using pharmacological approaches. Nevertheless, to the attention of Reviewer #2 and the Editor, we provide, as a further support, a transcriptomic analysis (see **Supplementary Figure 3**) adapted from the recent publication by Yue *et al.* (2024) for the expression of the gene (*i.e.* proteins) related to the channels and receptors identified in our study.

This figure could be added as a supplementary figure in our revised manuscript upon the decision of the Editor.

- 12) It would be informative if the authors add the data of negative responses to ATP and mGluR in suppl figures.**

As requested by Reviewer #2, we provide below a figure illustrating the negative responses observed following application of ATP (Purinergic receptor; **Supplementary Figure 4**) and of glutamate (mGluRs; **Supplementary Figure 5**). These data can be added as a supplementary figure, if necessary.

However, we would like to inform Reviewer #2 and the Editor that these data are part of a follow-up study (finalization of a manuscript), where we characterize the effect of these compounds and others in the activation of metabotropic receptors, the absence of modulation calcium channels but activation of calcium release from intracellular stores.

- 13) It would be helpful for the readers if the authors summarize the similar and different electrophysiological properties across the spinal levels in a table or diagram in the last part of the figures.**

A new figure summarizing (**Figure 13**) the electrophysiological properties of CSF-cNs along the spinal cord axis has been prepared to be included as a last figure in the revised manuscript.

Minor:

- 14) The title may not appropriately convey the information of the present study: “functional” may be replaced with “electrophysiological”, since no “functions” are identified in this study. “Intraspinal sensory neuronal population” is vague: this may be replaced with “cerebrospinal fluid-contacting neurons (in the mouse spinal cord)”.

We agree with Reviewer #2’s comment and will follow his/her advice. In the revised manuscript, the title now reads:

“Cerebrospinal Fluid-contacting neurons are sensory neurons with uniform morphological and region-specific electrophysiological properties in the mouse spinal cord”

- 15) Please carefully check if **suppl Fig. 1–4** are precisely indicated throughout the text and the figures.

We thank Reviewer #2 for the careful reading of our manuscript and pointing out these mistakes. We have accordingly updated and corrected the reference to the supplementary figures.

- 16) The data using ChAT-Cre mice (Suppl Fig. 3?4?) are not explained in Results.

Our comment below also applies to Point 22.

We thank Reviewer #2 for her/his insightful comment. Although Supplementary Figure 4 is just a positive control of our clearing protocol and light-sheet imaging acquisition protocol, we have now rephrased the results section as follows. Instead of:

“[...] we faced technical limitations and reached microscopy optical limits to resolve and identify single CSF-cN somatas (see **Supplementary Figure 2** in the revised manuscript).”

It now reads:

P6, L118-124:

“[...] we faced technical limitations and reached microscopy optical limits to resolve and identify single CSF-cN somatas (see **Supplementary Fig. 2**). To validate our clearing technique and light-sheet microscopy acquisitions, we conducted two additional experiments. First, we performed viral delivery into the ventricular system of ChAT-tdTomato mice to infect CSF-cNs, allowing us to compare the size of cholinergic neurons and CSF-cNs within the same preparation (Supplementary Figure 4A and B). Second, we cleared the spinal cords of ChaT-tdTomato mice and imaged them using the same settings previously applied to CSF-cNs. Under these conditions, we observed that the technical limitations of light-sheet imaging affect CSF-cNs but not other neuronal populations, likely due to their size differences (**Supplementary Fig. 2A and B**)”.

Change accordingly in the text

- 17) Line 125 etc.: please specify what $N =$ and $n =$ means. The number of mice, slices, cells? This information has been specified in the methods section.

P47, L1141-1142:

“All data are expressed as $\text{mean} \pm \text{SD}$ or SEM, as specified, and N and n correspond to the number of animal and data samples, respectively.”

In the revised manuscript, we have added in the Result section at their first mention what these parameters refers to.

P7, L146-147:

“Consistent with previous studies in the brainstem(12,22), spinal CSF-cNs have a high input resistance (R_m , $3.6 \pm 2.3 \text{ G}\Omega$; $N=191$, $n=659$; N and n for the number of animals used and the sample size, respectively)”.

We also explain these parameters explicitly at their first mention in the legends for **Figure 3**.

P56, L1432-1433:

“(N=10 for the number of animals, n=21 for the sample size) “

18) Line 145: please explain the abbreviation Vh in the first place.

The abbreviation has been explained at the first place mentioned with Vh standing for holding potential

P8, L172

19) Typos in line 255, 384, 1460.

Typos have been corrected:

Line 255 (P12, L290): bracket deleted. Reads now: *“[...] Ca_v selective blocker. In juvenile rat [...]”*

Line 384 (P20, L474): the comma after *“population”* has been deleted

“[...] We show that CSF-cNs form a dense interconnected neuronal population with approximately 10 to 20 cells per $10 \mu\text{m}$ of tissue depth across the entire CC axis. [...]”

Line 1460 (P74 L1879): the ‘l’ in Material was changed from italic to regular font.

“[...] “Maximal projection” [...]”

20) Line 472–474: are the data of different firing patterns between C and L-CSF-cNs shown in figures and Results?

To describe the data of cells with different firing patterns, we have initially prepared Table 2 that shows the proportion of single spike vs. tonic firing for each segment and presented the data in the text (original manuscript).

“AP discharge patterns were also assessed. In line with previous findings in juvenile rats(9) and postnatal mice(8), spinal CSF-cNs exhibit either tonic or single spike discharges. Following a current injection step (+20 pA, 200-500 ms duration and membrane potential at -60 mV, DC current: -10 to -15 pA), 58% of the neurons showed tonic AP firing (38 cells out of 65 recorded), while 42% fired a single AP (27 cells out of 65 recorded; Table 2). Comparative analysis revealed that most L-CSF-cNs and C-CSF-cNs had single AP discharges while T-CSF-cNs exhibited a primarily tonic pattern (Table 2).”

In the revised version we have prepared a new figure (**Figure 3**) and accordingly modified the manuscript. It now reads:

P7-8, L161-167:

*“AP discharge patterns were also assessed (**Fig. 3**). In line with previous findings in juvenile rats(9) and mice [8], spinal CSF-cNs in older mice exhibit either tonic or single spike discharges. Following a positive current injection step (+10 pA, 200-500 ms duration) from RMP, 58% of the neurons showed tonic AP firing (38 cells out of 65 recorded), while 42% fired a single AP (27 cells out of 65 recorded; **Fig. 3**). Comparative analysis between regions revealed that most L-CSF-cNs and C-CSF-cNs had tonic AP discharges while T-CSF-cNs exhibited a primarily single-spike pattern (**Fig. 3B**).*

21) Line 635, 638: *tdTomato mice are not “flex” mice.*

We agree with Reviewer #2, the tdTomato transgenic mouse model is not *flex* but **floxed** (flanked by LoxP sequences). The manuscript has been appropriately corrected.

22) Line 642: *the data using AAV are not explained in Results.*

See answer to point 16 above.

23) Line 692, 695, 699: *what is /ea?*

For more clarity, we now define this abbreviation when first used.

P35, L851: “After washing in 0.1M PBS (3 times; 2h/each; ea), samples were transferred”

24) Line 718, 719, 720: *z-Stack > z-stack*

P35, L838: Corrected to ‘Z-stack’

25) Line 754: *if wildtype mice were used, how were the CSF-cNs identified? Are the data of Pkd2l1-cre::tdTomato and WT mice combined in the figures? Please specify it.*

Our group has been studying CSF-cNs in mice since the early 2000. If we initially conducted our recordings in medullar acute slices obtained from transgenic mice selectively expressing fluorescent protein (eGFP: PKD2L1-IRES-Cre::floxed-eGFP) in CSF-cNs, we have now gained sufficient expertise to also record them in wild type mice based on their localisation around the central canal and shape.

Nevertheless, as stated in the Method section “CSF-cN visualization and recording of their intrinsic properties”

P39, L943-946:

“CSF-cNs around the CC were visualized using a computer controlled digital camera (HQ2 CoolSnap, Photometrics, SciCam, Scientifica) under epifluorescence illumination (exc. 520 nm/em. 610 nm, tdTomato fluorescence) and/or IR illumination with a 40x or 60x objective.” Finally, all our recordings were performed by adding a Alexa488 or 594 Hydrazide compound to the intracellular solution.”

P39-40, L959-965:

“The recording of CSF-cNs was confirmed based on the characteristic morphology observed (small round soma close or within the ependymal layer and a large dendrite ending in the CC with a round protrusion) from the cytosolic tdTomato as well as from Alexa488 or 594 Hydrazide (added to the intracellular solution, 10 μM; Invitrogen) fluorescence. The presence of spontaneous PKD2L1 channel activity was monitored as a further control.”

As mentioned in the Methods section and in the manuscript, and depending on the mouse availability, recordings were performed with either wild type or PKD-tdTomato mice and data obtained with the different animals were combined since the expression of the tdTomato fluorescent protein would not modify the electrophysiological properties of CSF-cNs.

To clarify this point and answer to Reviewer #2 possible concern, we added in the Methods section the sentence:

P40, L963-965:

“We did not observe a difference in the electrophysiological properties of CSF-cNs recorded in acute slices obtained from wild type or PKD-tdTomato mice.”

26) Line 848, 878, 879, 1302, 1316, 1321 etc: *Kainate > kainite*

Reviewer #2 must ask to write ‘Kainate’ with a lower case ‘k’ (kainate, not kainite). If so we have accordingly modified this point in the revised version of the manuscript.

27) Line 1303, 1327: *Glutamate > glutamate Corrected*

28) Line 1460, 1568: PKD-tdTomato > PKD2L1-Cre::tdTomato.

Detailed denomination of transgenic mouse models can be quite long. We therefore provided the extended model denomination in the Methods section “Animal models” (P33, L795-804) associated with shortened ones that were subsequently used in the rest of the manuscript for simplicity.

“We used wild type C57 Black6J (Charles River), PKD2L1-Cre (PKD-Cre: Pkd2l1^{tm1(cre)}; MGI ID: 6451758; a generous gift Emily Leman), and Choline Acetyl Transferase-Cre (ChAT-Cre: Chat^{tm2(cre)Lowl}; The Jackson Laboratory, MGI ID: 5475195; RRID:IMSR_JAX:006410) mice. PKD- and ChAT-Cre animals were cross-breed with floxed-tdTomato mice (Gt(ROSA)26^{Sortm14} (CAG-tdTomato)^{Hze}, The Jackson Laboratory, MGI ID: 3809524; RRID:IMSR_JAX:007914) to generate PKD-Cre (PKD-tdTomato)- and ChAT-Cre::floxed-tdTomato (ChAT-tdTomato) mice and to selectively express the tdTomato fluorescent protein in the neuronal population of interest. Animals of either sex were used for histology as well as for electrophysiological recordings (3-6 Weeks old mice).”

29) Please carefully present the asterisks and bars throughout the graphs and figure legends as follows:

We apologize for this oversight, and went carefully through the manuscript to verify and correct for the proper presentation of the asterisks. The text of the figure legends and the figures have been accordingly modified for more clarity (see revised manuscript and figures). We decided to give in the text and legend the exact p-values and accordingly illustrate the asterisks on the graphs.

- Line 1228: there are no * and ** in the graph (Fig. 4C).

In Figure 4C (**Figure 6C**, in the revised version) We compared the difference between K_V pic current densities for the Early (E, \$; Transient) and Persistent (P, #) components using a linear multi effect statistical test (lm function in R statistics). We set Level (Cerv, Tho and Lumb) and Current (Pers and Trans) as factors to compare the current densities (Dens).

The test gives no statistical differences for the comparison of current densities between Levels nor within a Level for current types (Pers and Trans).

We corrected this error that is due to edition (cut and paste) issues.

- Line 1236–1237: there is only **** in the graph (Fig. 4D).

Same error as above. Corrected.

- Line 1258, 1283, 1293, 1335, 1341, 1406, 1416: please specify what *, **, *** in the figures (Fig. 5C, 6B, 6D, 7E1, 7E2, 9C, 9D) mean. **It is unclear which groups do the bars and the asterisks in the graphs compare.**

Figure 5C (now 7C): we compare the Ca_V current densities between CSF-cNs recorded in the cervical, thoracic and lumbar segments.

Figure 6B (now 8B): we compare the GABA-mediated current densities between CSF-cNs recorded in the cervical, thoracic and lumbar segments.

Figure 6D (now 8D): we compare the glycine-mediated current densities between CSF-cNs recorded in the cervical, thoracic and lumbar segments.

Figure 7E1 (now 9E1): we compare the amplitude of the glutamate-mediated depolarization between CSF-cNs recorded in the cervical, thoracic and lumbar segments.

Figure 7E2 ((now 9E2): : we compare the number of action potential triggered by the application of glutamate in CSF-cNs recorded in the cervical, thoracic and lumbar segments.

Figure 9C/D (now 11C/D): we compare the Ca_v current densities in control, in the presence of baclofen and after washout of the agonist between CSF-cNs recorded in the cervical, thoracic and lumbar segments.

We have modified this figure to now present the currents normalized to the mean control value of the population. We also re-analyzed the data using a Friedman statistical test, more appropriate to compare for each segment the effect of baclofen.

For the panel 9D (now **11D**), on the left part of the graph we compare the percentage of inhibition in the presence of baclofen and after washout. Here we analyzed the data using a kruskal walis test for the Bcl effect (the condition) as well as the comparison in the 3 segments of interest (the level).

The same comparison has been illustrated in **Figure 10C and D (now 12C, D)**.

- Line 1322: no * in the graph (Fig. 7C)

Fig. 9C - This is a 'cut and paste' error; Legend corrected

- Line 1356: no ** in the graph (Fig. 8B now Fig)

- Line 1369, 1375 (Fig. 8D1, D2): * should be added in the graphs?

Now Fig. 10B, D1 and D2 - We have clarified these points on the graph and in the legend.

- Line 1440: * $p < 0.05$? It is unclear which groups do the bars and * in the graph compare (Fig. 10C).

We have modified the figures for more clarity

- Line 1450: no * and ** in the graph (Fig. 10D).

We have modified the figures for more clarity

- Fig. 1D: it is unclear which groups do the p values compare. * is not consistent with the p values.

The Kolmogorov-Smirnov (K-S) test responds differently to sample size. In very large datasets, the K-S test can detect even minute differences that may not be biologically or practically relevant. Therefore, a more stringent threshold (e.g., $p < 0.001$) is necessary to avoid identifying trivial effects as significant. For highly powered tests like K-S in large datasets, adjusting the threshold helps prevent overestimation of significance.

P45, L1099-1101:

*"For comparison of CSF-cN distances to the CC at the different SC levels, Kolmogorov-Smirnov test was used. Since K-S test is extremely powerful over large data sets, results were considered significant when * $p < 0.001$."*

- Fig. 7B: it is unclear which groups does *** in the graph compare.

We performed a lnm statistical test followed by a pairwise posthoc analysis (R Statistics; see Methods section, "Statistical Analysis"), where data were compared between Conditions (glutamate without or without DNQX) and between Region (cervical, thoracic and lumbar). The '***' on the top of the graph is referring to the statistical difference between recordings performed in the different Conditions in the presence of 'glutamate' application alone and 'glutamate application in the presence of DNQX'. The figure (**Figure 9B** in the revised version) has been accordingly modified to clarify this point.

We now distinguish the statistical difference between glutamate alone and glutamate with DNQX with violet asterisks and bar and the difference for glutamate responses between segment with black asterisks and bars.

3) Reviewer #3 (Remarks to the Author):

Manuscript “An intraspinal sensory neuronal population with homogenous morphological and region specific functional properties in the mouse” by Crozat et al., using morphological and in vitro electrophysiological techniques characterized the properties of a kind of neurons that locate around the spinal central canal and their dendritic protrusions contact cerebrospinal fluid (CSF), so-called CSF-contacting neurons (CSF-cNs), in mice.

They found that spinal CSF-cNs exhibit a conserved morphology across species and are uniform across SC segments in the mouse. They distribute densely along the entire spinal cord axis and are primarily located ventrally. Functionally, along the spinal cord, they share similar intrinsic and chemosensory properties with mouse medullar CSF-cNs. The neurons express NaV, KV, LVA and HVA Ca²⁺ channels. They also express functional classical inhibitory and excitatory ionotropic synaptic receptors.

The authors conclude that the spinal CSF-cNs represent a morphologically homogeneous sensory neuronal population, but functionally different along the spinal cord axis.

The study will for sure contribute to the better understanding of CSF-cN functions.

In general, each step of the experiments was carefully designed and conducted, and the results were objective, and the drafting of the manuscript is concise and thorough. The work with tissue clarity technique is beautiful. **The work is of interest to others in the community. I don't have major comments but some minor ones.**

- 1) *Some abbreviations in the Abstract do not have a complete name, such as PKD2L1, TRP and ASICs. Please give these abbreviations complete names when first used anywhere in the manuscript even in the abstract.*

The abstract has been updated and abbreviations replaced with full denomination. We went through the manuscript and gave full names at the first use of an abbreviation.

- 2) *Page 3, line 60, concerning the function in detecting spinal cord bending, is mainly observed in zebrafish. This needs to be stressed although the related references were given.*

Corrected. It now reads

P3, L58-61:

“These neurons express Polycystin Kidney Disease 2-Like 1 (PKD2L1), a channel sensitive to pH, and osmolarity changes (chemosensitivity) [13,25–28] that was also shown, in zebrafish larvae, to respond to CSF flow, and SC bending (mechanosensitivity) [29–31].”

- 3) *Page 4, line 83, “there are” should be “they are”.*

Corrected

- 4) *Page 5, lines 93-94, better to mention what animal species for this statement. The authors seemed to have included old articles (although some relatively new). Please update the references especially including those using primates.*

This part has been modified for more clarity. It now reads

P5, L102-104:

“CSF-contacting neurons (CSF-cNs) are found along the entire cc in lamprey⁶, zebrafish larvae [5,21], turtle [42] rat [9], mouse [7,11,13–16,36] and macaques [4,10,43] where they exhibit a consistent morphology.”

- 5) *Page 5, line 102, “(Fig.1B-D, Right ventral view)”. Is 1D from ventral view or side view?*

We thank Reviewer 3 for this insightful comment. Indeed, the right view of **Figure 1B-D** is ~90° rotated regarding the views presented in the left panels. Due to inherent difficulties in SC manipulation during the clearing process and the light sheet imaging (specifically in whole CNS specimens) some torsion may exist thus giving the impression of incoherence between the right panels (Panel 1D, right). In any case this is not a ventral view.

We have now changed “ventral view” by “caudo-rostral view”, which fits for all the right panels in **Figure 1**.

- 6) Page 5, lines 113-114, “A larger proportion of CSF-cNs is located in the ventral CC (Fig. 2A, Right; 2C) and neurons are closer to the CC in cervical and lumbar segments but further away in the thoracic one (Fig. 2D)”. **Isn't that in the lumbar segment the CSF-cNs further away from CC according to Fig. 2D (the blue dots)?**

We thank Reviewer #3 for this comment.

In Figure 2D, the blue dots represent data for the thoracic segment. What can be seen is that thoracic CSF-cNs (blue dots) are slightly, although significantly, further away from the CC than cervical or lumbar CSF-cNs. Our statement is therefore correct.

- 7) Page 6, line 128, “...compared to T- and C-CSF-cNs (Table 1)”, should be “...compared to L- and C-CSF-cNs (Table 1)”.

We agree and have corrected this mistake.

- 8) Page 6, line 132, “...depolarized CSF-cNs in anterior regions”, better to use “rostral regions”.

Modified accordingly.

- 9) Page 7, lines 141-142: “the findings suggest that CSF-cNs share similar intrinsic properties but exhibit differential excitability along the spinal cord”. Does this mean that “excitability” is not an “intrinsic property”.

We agree with Reviewer #3 that excitability as intended here corresponds to the capability for a neuron to generate action potential and with which pattern. Since this response is triggered by an external stimulus (*i.e.* DC current injection) and not by synaptic inputs it corresponds to an intrinsic property mediated by active conductances.

We therefore modified the revised manuscript to distinguish intrinsic properties between passive (Rm, Cm ...) and active ones (spiking capabilities).

In the revised manuscript this reads as follow:

P10, L221-225:

“Overall, our findings suggest that CSF-cNs are highly resistive neurons showing passive properties as well as firing patterns that differ along the cc. They further exhibit shared chemosensory functions by responding to variations in extracellular pH. These differences were observed between CSF-cNs recorded in different segments but not associated to a specific localization around the cc.”

- 10) Page 8, line 182, can the authors specify which subtypes of ASICs are expressed in mouse CSF-cNs? In lamprey it is primarily ASIC3.

In our previous reports (Orts Del'Immagine *et al.*, 2012 and 2016), we indicated that medullar CSF-cNs expressed the ASIC1a homomers and ASIC1a/2b heteromers (sensitivity to psalmotoxin). In the present study we did not use pharmacological tools to determine the ASIC subtype expressed in mouse spinal CSF-cNs. However, based on the similar current kinetics observed in our study, we can assume that spinal CSF-cNs would also express ASIC1a homomers and ASIC1a/2b heteromers. This assumption is supported by the transcriptomic data recently published by Yu *et al.* (2024). We present below a raster plot based on the mentioned study that confirms the presence of ASIC1 and 4 genes (score of 9 and 8.1, respectively for samples 3 and 4 and see the **Supplementary Figure 3** about gene expression in spinal CSF-cNs adapted from the mentioned publication).

In the revised version of the manuscript, we know state:

P9-10, L216_220:

“The current properties observed in our study are in line with those reported in previous reports(25–27) and suggest that they are mediated by PKD2L1 channels and ASIC. An observation that is further supported by the transcriptomic analysis recently carried out by Yue and Collaborators(11) where gene for PKD2L1 and ASICs 1 and 2 can be found (see Supplementary Figure 3A and Discussion).”

11) Page 11, line 255, there is an extra “(“ *The extra ’)*’ was deleted.

12) Page 15, line 354, please specify “mACH-Rs.s”, or it is one “s” extra. *The extra ‘s’ was deleted.*

13) *In the Discussion, the authors need to compare the morphology and the distribution of CSF-cNs between different animal species, especially with those at a higher level, such as primates.*

There are such articles e.g.:

Liu et al. PKD2L1-expressing cerebrospinal fluid contacting neurons in spinal cords of rodents, carnivores, and primates. *Int. J. Mol. Sci.* 2023, 24, 13582. <https://doi.org/10.3390/ijms241713582>

Tonelli Gombalová Z et al. Majority of cerebrospinal fluid-contacting neurons in the spinal cord of c57bl/6n mice is present in ectopic position unlike in other studied experimental mice strains and mammalian species. *j. comp. neurol.* 2020, 528, 2523–2550. doi: 10.1002/cne.24909.

We have addressed and discussed this point in the revised manuscript. It now reads

P20-21, L481-502:

“CSF-cNs have been observed in several vertebrate species, where they show similar morphology as well as distribution around and along the cc and. In lampre[[6], zebra fish [5,20] and turtle [42], they mainly exhibit a triangular, small soma inserted in the ependymal cell layer with a short dendrite projecting to the cc (but see below for lamprey). Similar features are found for CSF-cNs in postnatal rodents [8,9,14]. With aging, their morphology and localization changes, and one can find either intra-ependymal neurons, embedded within the ependymal cell layer, or subependymal, located below the cc, with a longer dendrite extending into the cc lumen [7,12,14,33]. In mice, spinal CSF-cNs are predominantly found in the ventral region, comprising about 60% of the total. This ventral localization contrasts with the lateral distribution observed in the medulla [7] (but see also Kútna and colleagues [34]) and suggest an anatomical reorganization along the medullo-spinal axis. We also confirm the presence of PKD2L1-expressing neurons (tdTomato+) in more distal ventral locations along the SC [14]. Note that Tonelli Gombalová and colleagues indicated that in C57 Black6/N mice, in contrast to the J substrain, an important proportion of CSF-cNs is observed in ectopic positions away from the cc due to potential dysfunction of Crb1 and Cyfip2 products in this substrain [48]; our present and past studies were conducted on the C57 Black6/J substrain.

CSF-cNs are also observed in NHP [4,5,10,43] and a recent study indicates that CSF-cNs morphology, density and distribution [10] are similar to that observed in rodents with PKD2L1+ neurons that are also localized in the ventral region of the SC away from the cc [10]. Interestingly in Macaques, CSF-cNs are largely present in subependymal position, exhibit long dendrites and are the only neurons localized around the cc in a hypo-neuronal region enriched with astrocytes and microglia [43].”

14) *In connection with the last point, the authors also need to do a discussion from a translation point of view. Considering that in the majority of humans the central canal is closed and there are no reports as to whether there are CSF-cNs in the human spinal cord, how can the results from the animal studies be translated to human?*

We have addressed and discussed this point in the revised manuscript. It now reads

P21, L502-512:

“One crucial question, that is still controversial due to experimental limitations and contradictory reports, concerns the presence of CSF-cNs and whether they would play a similar sensory function in human SC. First, the cc is thought to have collapsed in adult Humans below the cervical region [49] in older subjects but more recent studies have shown that the cc is preserved along the SC [50–52]. Second, there is no evidence for the presence of CSF-cNs in Human and one of the first genetic analysis of human spinal cord failed to reveal the presence of the gene for PKD2L1 presumably due to experimental limitation [53]. Nevertheless, this remains an open question that needs to be addressed in the future to demonstrate whether this unique neuronal population is conserved in Human and if not whether it has been replaced by another system capable of integrating CSF circulating signal in the SC.”

15) Page 16, line 384, two refs. with a # 33 (one should be 32).

Corrected

16) Page 26, line 610, “CS-cNs” should be “CSF-cNs”.

Corrected

17) Page 27, in the section “Histology and imaging”, please specify what kind(s) of mice were used for imaging study.

According to Reviewer #3’s suggestion, we have now replaced the sentence [...] “Animals were injected” [...] by

P34, L819-820

[...]“PKD-tdTomato (Fig. 1 and 2 and Supplementary Fig. 1 and 2) or ChAT-dTomato (Supplementary Fig. 4) animals were injected” [...]

18) Page 9, line 692, specify what “o/n” means. We specified overnight for o/n

19) Page 37, line 876-881, “For the agonists:...”. This seems not to be a complete sentence.

We changed the sentence to

P43, L1041:

“For the agonists, the following compounds were used:” .

20) Page 39, line 928-929, “Results were considered significant when $*p < 0.05$, $**p < 0.01$ ”. Only $*P < 0.05$ ” is enough. The “*” is not needed here. Same principle applies to page 42, line 1003.

Corrected

P48, L1172-1175:

*“Statistical differences were considered as significant for $p < 0.05$ and we give in the text and figure legends the p values for each test. In the corresponding figures, asterisks are shown as follow: ns, *, **, *** and **** for p non significative, < 0.05 , < 0.01 , < 0.001 and < 0.0001 , respectively.”*

21) Page 48, line 1170, “thoracic segment (B)” should be “thoracic segment (C)”.

Corrected

22) In the figure legends, please specify what “N” respective “n” stands for. They are specified in Table 1 and 2, but not in other method parts or figure legends.

This information has been given in the Method section “Processing, analysis and statistics, Electrophysiology”

Our feeling is that adding this information in the figure legends will make them even longer than they are now. We have detailed that when we describe statistical analysis in the Methods section, in the Results section and the legends when it appears for the first time, In the revised manuscript, we have added in the Result section at their first mention what these parameters refers to.

Results

P7, L146-147:

“Consistent with previous studies in the brainstem(12,22), spinal CSF-cNs have a high input resistance (R_m , $3.6 \pm 2.3 \text{ G}\Omega$; $N=191$, $n=659$; N and n for the number of animals used and the sample size, respectively)”.

Methods

P47, L1141-1142:

“All data are expressed as mean \pm SD or SEM, as specified, and N and n correspond to the number of animal and data samples, respectively.”

We also explain these parameters explicitly at their first mention in the legends for **Figure 3**.

P56, L1432-1433:

“(N=10 for the number of animals, n=21 for the sample size) “

23) Page 54, line 1304, “time” should be “time”.

Corrected

24) *In general, the figure legends are too long. Is it possible to place the statistical details in the text, but not in the legends?*

In the journal Guidelines to prepare Figures and legends, it is recommended to detail the statistics in the legends. However we agree that it ends with very long texts for the legends. On the other hand, we decided to avoid too long results and statistical descriptions in the text to ensure easier reading. Alternatives would be to:

- detail the results and statistical description in the text
- provide extended figure legends as a supplementary file,
- or prepare a supplementary statistical document summarizing the statistics figure by figure.

For this issue, we will comply with the editor and publishing policy/decision.

25) *Figure 4, please place the voltage step gradients (each Vstep) used in Figure 4A, similar as Figure 5A.*

Now it is **Fig. 6**. The figure has been accordingly modified and we took this opportunity to modify also **Fig. 5**.

26) *Figure 7B, is “Control” the same as “Glu”?*

Yes it is and we modified the figure to replace “Control” by “glutamate”.

27) *Figure 9, “+ CGP” in red color is a bit confusing. It is the cervical recording that shows red color. It is better to place “Baclofen” and “+ CGP” above the traces in A2 and A3 as well.*

28) *Figure 10, same as Figure 9, please do the same for “Oxo” and “+ Atr”.*

In these two figures as well as in all figures where traces with an antagonist or a channel blocker are presented, we have modified the color used. Now all are violet.

Point-to point response to Reviewers' comments

We would like to thank the reviewevers for their careful reading of our initial manuscript and for their valluable comments. Please find below a point-to-point answer to the comments and requests. We hope to have adequatly address the point raised and that in his revised version our manuscript will be suitable for publicationn.

We apologize for the delay in our response due to unforeseen reasons.

Best regards,
Prof. N. Wanaverbecq

1) Reviewer #1 (Remarks to the Author):

Cerebrospinal fluid-contacting neurons (CSF-cNs) are intriguing neurons that are now well characterized in zebrafish where they play a role in sensory-motor control and a similar role has recently been identified in rodents(1). In the present manuscript, the authors have surveyed CsF-cNs along different segments of the spinal cord and provided a lot of convincing data on their intrinsic properties and transmitter sensitivity.

Previous work by the authors and others has shown that CSF-cNs are present at all levels of the spinal cord(2) (3) (4), express voltage-gated Na, K and Ca channels (5-8) are sensitive to pH (6, 9) through expression of ASIC (10) and PKD2L1 (2, 11) (12) and have functional glycine (9), GABAa (6, 9), GABAb (8) and cholinergic (7, 13) receptors. This manuscript reaffirms these previous findings along the spinal cord and as such the **findings can't be considered novel**.

Although we agree with Reviewer #1 that the data presented in our study have been reported to some extent for CSF-cNs. The studies mentioned were conducted in **different animal models** (rat, lamprey or mouse), **at different ages** as well as **in a specific region** along the central canal (medullar or one spinal cord segment).

However, to date there is no anatomical and functional data available that systematically characterize and compare the properties of CSF-cNs in the same animal model (*i.e.* the mouse) and along the central canal axis. Such a characterization has not been conducted prior and until now we did not know whether CSF-cNs show anatomical and electrophysiological properties that would be region-specific or not. **In this regard, our study is novel**. Prior to our study, authors is the field could only extrapolate about properties uniformity or not, now we have groundthruth data demonstrating it

Our claim is actually shared by Reviewers #2 and #3 who acknowledge that our study contributes to the better understanding of CSF-cN properties along the central canal, a crucial and necessary step for future characterization of their role as a novel sensory population intrinsic to the CNS.

Reviewer 2: "Overall, the present study provides some advances in the knowledge of spinal CSF-cNs, extending from the previous studies in rodents, especially on the electrophysiological responses at different spinal levels."

Reviewer 3: "The study will for sure contribute to the better understanding of CSF-cN functions."

Major comments:

- 1) In the abstract, on line 36, and introduction line 83, it is claimed that CSF-cN functional properties vary along the spinal cord.

It is unclear what the authors are referring to, **none of the subheadings of the results or discussion indicate differential function**. Contrary to this claim, subheadings in the results (line 119) and discussion (line 440) states '*Along the spinal central canal, CSF-cNs exhibit similar intrinsic properties*'.

In the revised version of the manuscript, we have checked for consistency between Title, Headings and Subheadings and our claims for different electrophysiological properties in CSF-cNs along the central canal axis. Wherever necessary, we have clarified this point and discussed it.

For simplicity, we indicate here the modified **Heading** and **Subheading**, the Page/Lines in the given section where claims for differences are mentioned and the conclusion of the corresponding section. We have also clarified this in the figure captions.

Title

“Cerebrospinal Fluid-contacting neurons are sensory neurons with uniform morphological and region-specific electrophysiological properties in the mouse spinal cord.”

Abstract

Abstract has been modified to comply with the guidelines and the 150 words limits.

P2, L27-40:

“Cerebrospinal Fluid-contacting neurons (CSF-cNs) are GABAergic neurons found along the medullo-spinal central canal in vertebrates, but their properties in mice remain largely uncharacterized. They express Polycystin Kidney Disease 2-Like 1 channels (PKD2L1), members of the Transient Receptor Potential superfamily, with sensory properties.

*CSF-cNs are distributed throughout the spinal cord with a uniform morphology and a primarily ventral localization. **CSF-cNs exhibit region-specific intrinsic properties, expression of voltage-dependent and ligand-gated conductances and detect variation in extracellular pH through activation of PKD2L1 and Acid-sensing Ion Channels.** They possess GABA_B and muscarinic receptors, not glutamatergic metabotropic ones, to modulate Ca²⁺ channels.*

***CSF-cNs are sensory neurons with a uniform morphology and electro-physiological properties specific to the network they are inserted in.** The future challenge in the field, will be to demonstrate CSF-cNs integrate circulating signals along the central canal while differently integrate synaptic inputs to modulate body function through specific local spinal network.”*

Introduction

P4, L86-89:

“Electrophysiological recordings show that, although they share similar sensory functions and the same expression of ionic conductances and ligand-gated receptors, they present differences in their passive and firing properties as well as in the expression density of the identified channels and receptors.”

P4-5, L91-95:

“On the other hand, they exhibit differential electrophysiological features in the SC segments and these region-specific properties might serve the specific role they play in a given spinal network. Anatomical and functional evidence suggest that CSF-cNs serve as a novel sensory or interoceptive system intrinsic to the CNS.”

Results

P6, L140, 141:

Heading: “Spinal CSF-cNs intrinsic and sensory properties along the central canal axis”

Subheading: “CSF-cNs have region-specific passive properties and firing patterns”

P7, L149-150; P7-8, L165-167;

Conclusion of this section (P10, L221-225):

“Overall, our findings suggest that CSF-cNs are highly resistive neurons showing passive properties as well as firing patterns that differ along the cc. They further exhibit shared chemosensory functions by responding to variations in extracellular pH. These differences were observed between CSF-cNs recorded in different segments but not associated to a specific localization around the cc.”

P10, L227-228:

Heading: *“Spinal CSF-cNs express functional sodium, potassium and calcium voltage-dependent channels”*

P12, L285-286; P13, L297-299

Conclusion of this section (P13, L300-305):

“Spinal CSF-cNs express functional NaV, along with KV of the delayed rectifier- (IKD) and A-type (IA), which present similarly expression densities along the cc axis and would be responsible for the AP generation. Additionally, all spinal CSF-cNs express both HVA and LVA CaV, with higher current densities in T-CSF-cNs. These findings are supported by a recent transcriptomic analysis of gene expression in spinal CSF-cNs¹¹ (see Supplementary Fig. 3B-D).”

P13, L307:

Heading: *“CSF-cNs express ionotropic synaptic receptors with a density that differs along the cc”*

P14, 321-324; P14, L335-337; P14-15 L340-343; P15, L351-352

P15-16, L365-367:

Conclusion of this section

“To summarize, our data show that all spinal CSF-cNs express functional GABAA, glycine, AMPA/kainate glutamatergic, and nACh receptors but at different densities depending on the spinal cord segment considered. Activation of AMPA/kainate and nACh receptors modulates CSF-cN excitability and can trigger AP firing.”

Discussion

P19-20, L463-467:

*“Overall, spinal CSF-cNs represent a morphologically uniform neuronal population capable of detecting sensory signals that exhibits differential electrophysiological properties along the SC axis potentially to modulate the specific spinal network they are inserted in (**Fig. 13**). Although differences can be observed between CSF-cNs from different segments, our study does not point to an organization of CSF-cNs in specific cluster around the cc.”*

We also provide a summary figure (**Figure 13**) illustrating the electrophysiological differences along the cc.

P23, L554-555

Heading: *“Along the spinal central canal, CSF-cNs exhibit region-specific electrophysiological properties”*

P24-25, L588-592; P25, 603-605; P27, L64-648; P27, L658-662

P29, L694:

Heading: *“CSF-cNs show a differential excitatory/inhibitory drive along the central canal axis.”*

P29, L705-706; P30, L713-717; P30, L719-722

- 2) *The authors also show that all identified synaptic receptors are present in each subdivision, with only some variation in the current density for GABA_A and iGluR observed. However, the **functional relevance of this has not been addressed**. The authors should reconsider the claim of functional properties varying along the spinal cord.*

In our study, we have analyzed the functional expression of 'classical' synaptic receptors by a pressure application approach of specific agonists and reproduced the recordings in the presence of selective antagonists.

To take into account possible differences in cell size (*i.e.* membrane capacitance) that would have an effect on the current amplitude and mask possible differences in the density of receptor's expression, we have normalized the current amplitude to the membrane capacitance and expressed data as current density in all our analyses. These analyses demonstrate statistical differences for the current densities of glycine- and GABA_A-mediated responses (**Figure 8**, in our revised manuscript) as well as for the glutamate (AMPA/kainate)-mediated ones (**Figure 9**, in our revised manuscript). For the glutamate-mediated responses, this difference further indicates a region-specific modulation of CSF-cN excitability by glutamate since the lower current density observed for glutamate (AMPA/kainate)-mediated response underlies the lower glutamate-mediated excitability in the lumbar segment (**Figure 9D-E**, in our manuscript).

We agree that we report differences in the electrophysiological properties (current densities), not the functional consequences, however having differential receptor expression, *i.e.* current amplitude, would suggest differential function. Our results therefore indicate that CSF-cNs along the central canal axis would be differentially modulated by the specific activation of synaptic receptor subsets and suggest that the excitatory and inhibitory drive onto CSF-cNs differs along the central canal axis (*i.e.* stronger inhibitory vs. excitatory drive in lumbar CSF-cNs)..

This opens future lines of research to further characterize the underlying connectivity leading to a differential integration within segment-specific local networks. We are currently addressing these aspects in dedicated morphological and functional studies.

The same analysis was carried out for voltage-dependent conductances and we also observed significant differences in the expression of Ca_v with thoracic CSF-cNs exhibiting larger calcium current densities and potentially calcium signaling (**Figure 7**, in our revised manuscript). An aspect that also needs to be further investigated.

Considering that in mammals, and actually in most species studied so far, CSF-cNs remain largely under characterized at the functional level, any new piece of evidence contributes to their better understanding.

We have stressed this point in the Discussion of the revised manuscript. It now reads:

P30, L713-726:

“Our results further indicate differences in the current densities of glycine and GABA_A receptors-mediated responses that is highest in thoracic CSF-cNs. We also report in lumbar CSF-cNs lower current densities for glutamate (AMPA/kainate)-mediated responses underlying the observed lower excitatory action of AMPA/kainate receptor activation. Due to the recording configuration (whole-cell patch-clamp and artificial intracellular chloride concentration), it is difficult to discuss the functional consequence of the inhibitory signaling onto spinal CSF-cNs (see also Riondel and Colleagues³⁶). Nevertheless, altogether, these results might imply that the excitatory and inhibitory drives onto CSF-cNs differ along the cc axis and that CSF-cNs would be differentially modulated through synaptic inputs within local spinal networks. The functional consequence of this differential excitatory/inhibitory drives is unknown but opens future lines of research where the underlying connectivity leading to a differential integration within segment specific local networks needs to be characterized. We are currently addressing these aspects in dedicated morphological and functional studies.”

P30, L731-733:

“However, the effects induced by the activation of these two receptors are different suggesting that CSF-cNs might integrate or code differentially the synaptic inputs they receive depending on the network they are inserted in.”

- 3) *Due to the technical issues (low resolution) with light sheet imaging I am not sure what Figure 1 adds to the manuscript that isn't conveyed by Figure 2.*

Although we agree with Reviewer #1's comment that the lack of cellular/subcellular resolution prevents a detailed analysis of CSF-cN morphology or distribution, we believe **Figure 1** plays an important role in demonstrating the mesoscopic distribution of CSF-cNs in the intact mouse CNS. While several studies have shown CSF-cNs at different spinal levels in the mouse SC, a comprehensive view of the entire CSF-cN system is lacking in the literature. **Figure 1**, in its current form, summarizes this data, illustrating that the CSF-cN system extends from rhombencephalic levels to the lumbosacral spinal cord within the mouse CNS. Indeed, the quantification of **Figure 2** is more accurate since we are able to differentiate individual cells, however, **Figure 1** supports the conclusion drawn from **Figure 2** since to infer the distribution of CSF-cNs in the whole CNS by routine histology techniques would be a much more time-consuming task. Moreover, we are currently implementing other clearing techniques with improved resolution (due to the absence of tissue shrinkage) that will enable the use of light-sheet microscopy to study CSF-cN distribution and connectivity. Notably, these experiments are ongoing and beyond the scope of the current manuscript.

- 4) *Statistical analyses appear appropriate.*

We thank Reviewer #1 to acknowledge the well conducted statistical analysis.

Minor:

- 5) *page 3 line 53, you specify rats but don't make it clear that refs 19-21 are from mice. You could possibly change 'rats' to 'rodents' otherwise you should be careful of the species.*

Modified. The sentence has been changed to:

P3, L53-55:

“In rodents, CSF-cNs exhibit spontaneous action potential (AP) firing, mediated by sodium and potassium voltage-dependent channels [9], and they also express voltage-dependent calcium channels (CaV 2.2 and CaV 3.1-3.3) [22,23,24].”

and the reference 9 was moved after “voltage-dependent channels”.

- 6) *Line 63 'regrouped' should just be 'grouped'.*

Correction made (L67)

- 7) *Also, it is not evident what is meant by **immature vs mature phenotypes**.*

Classically, the level of neuronal maturity is based on an expression switch from immaturity markers (doublecortin (DCX), HuC/D, PSA-NCAM and Nkx6.1 and Nkx2.2) during development and postnatal period to those of maturity (e.g. neuronal nuclear protein, NeuN). This maturation is accompanied by functional changes such as action potential firing pattern and insertion in synaptic networks.

Our claim is based on previous reports from different groups (Marichel *et al.*, J.Neurosci. 2009; Reali *et al.*, J.Physiol. 2011; Petracca *et al.*, 2016, DiBella *et al.*, 2019) in the field as well as publications from our group (Orts Del'Immagine *et al.*, 2016 & 2017) where it was shown that in CSF-cNs DCX, PSA-NCAM and HuC/D expression is observed along with low or no expression of NeuN. We also reported that CSF-cNs would maintain Nkx6.1 and DCX expression in mice as old as year of age (Orts Del'Immagine *et al.*, Neurosci. 2017). In parallel, there is functional evidence that subsets of CSF-cNs exhibit Calcium-mediated or single-sodium spike firing in juvenile rats (Marichal *et al.*, 2009; see comments bellow) as well as

depolarizing effect of GABAergic signalling in turtle (Reali *et al.*, J.Physiol. 2011) also observed in 8 weeks-old mice (Riondel *et al.*, 2024).

Further, in our previous reports (Orts Del'Immagine *et al.*, 2016 & 2017; Riondel *et al.*, 2024), we also indicated that, in contrast to what is observed in juvenile animals, although different CSF-cN phenotypes (*i.e.* immature vs. mature) can be observed in older mice, these neurons are not organized in specific clusters but rather distributed around the central canal.

Altogether, these features indicate that in older mice a subset of CSF-cNs present features of immature neurons. In the revised manuscript, we have accordingly edited the text to clarify this point:

P3-4, L64-74:

“In zebra fish larvae and juvenile rodents, morphological [6,8,21,32] and phenotypical [8,32–34] differences suggest that CSF-cNs are grouped in two subpopulations. In the cc ventral region, they exhibit immature phenotypes (expression of doublecortine, PSA-NCAM and homeoboxes Nkx2.2 and 6.1) [5,8,21,32,33] and fire single AP [8,32], a feature often associated to immaturity [35]. Dorso-lateral CSF-cNs have a more mature phenotype (expression on the neuronal nuclear protein, NeuN) and tonic AP discharge [8,21,32,33]. In older mice, CSF-cNs still show the expression of these immaturity markers, two distinct AP firing patterns and, in lumbar segments, two subpopulations can be distinguished based on their responses to GABAergic signaling, either inducing depolarization or hyperpolarization [36]. Nevertheless, and in contrast to what has been reported for juvenile mice, there is no correlation between CSF-cN phenotypical and firing properties and their localization around the cc.”

- 8) Why is tonic discharge considered mature and single AP firing immature? Perhaps you should be more specific on what is different without referring to the maturation stage of a neuron.**

The single-spike profile with neurons emitting one AP in response to depolarization, but without being able to sustain regular firing is often associated with immaturity and low expression of voltage-gated sodium channels underlying action potential firing. Articles from the Russo group (Marichel *et al.*, J.Neurosci. 2009; Reali *et al.*, J.Physiol. 2011) indeed suggest that single-spiking corresponds to an immature stage. To support their claims, they cite publications from Spitzer and collaborators (Spitze & Ribera, 1998 and Ben-Ari & Spitzer, Trends Neurosci 2010), where the claim that single-spiking corresponds to an immature neuronal stage.

For CSF-cNs the single-spike firing profile has been classically used to define their immature *i.e.* low maturity state (Marichal *et al.*, J.Neurosci. 2009; Reali *et al.*, J.Physiol. 2011; Petracca *et al.*, 2016, DiBella *et al.*, 2019). Based on these reports and the ‘association’ of single spiking with immaturity, we suggest, in agreement with earlier reports, that a subset of CSF-cNs would have an immature phenotype.

Nevertheless, we discussed this claim in the revised version of the manuscript.

P3-4, L64-68:

“In zebra fish larvae and juvenile rodents, morphological [6,8,21,32] and phenotypical [8,32–34] differences suggest that CSF-cNs are grouped in two subpopulations. In the cc ventral region, they exhibit immature phenotypes (expression of doublecortine, PSA-NCAM and homeoboxes Nkx2.2 and 6.1) [5,8,21,32,33] and fire single AP [8,32], a feature often associated to immaturity [35].”

P25-26, L603-615:

“We also observed that C- and L-CSF-cNs exhibit mainly tonic firing while thoracic neurons mostly show single-spiking pattern. The single-spike profile with neurons emitting one AP in response to depolarization, but without being able to sustain regular firing, is often associated with immaturity and low expression of voltage-gated sodium channels underlying action potential firing [57,58]. For CSF-cNs, the single-spike firing profile has been classically used

to define an immature i.e. low maturity state [8,9,32,42]. However, to date there is no evidence in CSF-cNs of the correlation of single-spike firing and the expression of immaturity marker. One would need to conduct dedicated studies to address this point. When referring to single-spike discharge, alike other reports^{8,9}, we would rather consider a discharge pattern with a 'depolarization block' where membrane potential depolarization impedes the generation of AP (in Fig. 3A, compare depolarization induced by +10 and +30 pA current injections) presumably due to a lower density of sodium voltage-dependent channels (see Marichal and collaborators [9])."

9) line 71 'finally'... you have an additional point after the 'finally' statement. Reword. The sentence has been modified as follow.

P4, L77-85;

*"In zebrafish larvae, CSF-cNs selectively activate motor neurons and interneurons, influencing swimming behavior [29,37,38]. In mice, they play a role in controlling posture, balance [16], and locomotion [15]. **Further**, in zebrafish larvae, they appear to respond to bacterial toxins, suggesting a potential role in the body's immune defense [39]. **Finally**, it was recently shown that, following SC injury, CSF-cN constitutive activation through κ -opioid signaling is halted leading to disinhibition of ependymal cell proliferation to promote scar formation [11] and would even have regenerative properties [40,41]."*

2) Reviewer #2 (Remarks to the Author):

CSF-contacting neurons (CSF-cNs) are functionally enigmatic sensory cells facing the central canal of the spinal cord. In this study, Crozat and colleagues examined the morphological and electrophysiological properties of CSF-cNs in the mouse spinal cord by comparing them at the cervical, thoracic, and lumbar levels.

The properties were consistent across the spinal levels and aligned with previous observations in spinal and medullar CSF-cNs in mice and juvenile rats (Ref 9, 12, 19, 22), whereas variations in cell density and certain electrophysiological responses were noted between spinal levels.

The functional and biological significance of those differences are unclear and not clearly discussed in this study.

Overall, the present study provides some advances in the knowledge of spinal CSF-cNs, extending from the previous studies in rodents, especially on the electrophysiological responses at different spinal levels.

The experiments were well conducted to obtain the basic electrophysiological properties of CSF-cNs. The manuscript is well organized and the data are interpreted well.

Specific comments are as follows:

- 1) *The authors examined the morphological properties of CSF-cNs in clearing tissues and show beautiful images in Fig. 1 and Suppl Fig 1–2. However, unfortunately, they did not bring any novel information for mouse CSF-cNs.*

We acknowledge that the limited resolution of light-sheet microscopy, combined with the small size of CSF-cNs and their axons, limits detailed analysis of CSF-cN morphology and distribution using this technique. However, the population analysis of CSF-cN distribution in Figure 1 should be understood at a mesoscopic level.

Our main aim with this figure was to present the overall distribution of CSF-cNs in the intact mouse CNS, rather than providing detailed quantification or morphological analysis. Currently, our lab is making all possible efforts to implement clearing techniques and light-sheet microscopy acquisitions, combined with registered confocal acquisitions, to finely characterize the connectivity patterns and morphological features of CSF-cNs at spinal and supraspinal levels. These results, however, fall beyond the scope of the current manuscript.

- 2) *Pkd2l1-Cre::tdTomato mice likely label other cell types in the gray matter as seen in Fig. 1B–D, 2A and Suppl Fig 1B–D, 2B–D, 3C. Nonspecific labeling of the mice is also reported in previous studies. The authors likely include those cell populations in some analyses, but did not state how they are included or excluded from the analyses. Use of PKD2L1 antibody may be more precise for the counting.*

We appreciate the reviewer's comment. As observed both in our study and in others (Nakamura et al., 2023; Yu et al., 2024), the PKD-tdTomato mouse also labels cell types other than CSF-cNs, likely oligodendrocytes (Nakamura et al., 2023), within the central nervous system (CNS). We conducted a systematic analysis of these labeled cells throughout the entire mouse CNS and found: *i)* These cells are absent at ages younger than 4 weeks, and *ii)* at around 4-6 weeks, when these cells do appear, their fluorescence levels are significantly lower than those of actual CSF-cNs (as confirmed by PKD2L1 immunoreactivity). See the **Figure 1** below (Cervical SC, 6 weeks; Magenta - tdTomato; Green - α PKD2L1). Additionally, CSF-cNs near the central canal (CC) can often be morphologically identified because of their protrusions visible into the CC lumen. Based on this, we visually defined CSF-cNs as cells where the protrusion can be identified, along with those showing higher fluorescence values where the protrusion cannot be seen. It is important to note that in Figures 1 and 2 in the manuscript, all images correspond to 3-week-old mice, so there is no ectopic tdTomato expression in non-CSF-cN cells. The "ectopic" appearance of CSF-cNs is supported by our

previous findings, where PKD2L1 expression was observed in cells distant from the cc (Jurčić N *et al.*, 2021).

We also acknowledge the reviewer's suggestion to use anti-PKD2L1 antibodies to count CSF-cNs instead of relying on tdTomato fluorescence in the PKD2L1-tdTomato mouse. Unfortunately, there is no simple solution to this issue. While CSF-cNs are defined by PKD2L1 expression, immunoreactivity is often considered a semiquantitative technique, as it depends on factors such as antigen expression levels, fixation conditions, and antibody accessibility to specific epitopes. Antibody binding can even be affected by the molecular environment, especially for membrane-inserted ion channels (Lorincz and Nusser, 2008; Ramirez-Franco *et al.*, 2024). Indeed, although rare, we have also observed some morphologically identified CSF-cNs where PKD2L1 immunolabeling is barely detectable (see **Figure 2** below; arrow). For this reason, we choose to quantify CSF-cNs based on tdTomato fluorescence in the 3-6 weeks PKD-tdTomato mice.

Figure 1: Cervical SC, 6 weeks. Magenta-tdTomato; Green- α PKD2L1

Figure 2: Cervical SC, 6 weeks. Magenta-tdTomato; Green- α PKD2L1; scale bar=25µm

- 3) *The number of mice (and sections?) analyzed are not described in the histological analyses in Fig. 2A–D.*

Please refer to the Method section “Cell counting and distribution of CSF-cNs in confocal images” for this point as we specify:

P44, L1072-1075

“Five 6 weeks old mice were used for CSF-cNs quantification in confocal slices. 10 fields of view (FOV) were acquired per animal and SC segment (total 30 FOV/animal). To count CSF-cNs in confocal coronal slices, Z-stacks of 10-20 images were acquired. Counting was aided by custom FIJI macros available upon reasonable request.”

Please specify them in the figure legend. Do the dots represent the count in each section?

We thank the reviewer since adding this info to the figure legends will surely help to improve the clarity of the figure. In Figure 2B and 2C each one of the dots represents a slice. In Figure 2D, each dot represents an individual cell. This information has been added to the Figure legend.

P56, L1427-1428:

“Dots represent each slice and each individual cell in Figure 2B, 2C and Figure 2D, respectively.”

- 4) *Fig. 2A: it is weird that the % of the left and right lateral CSF-cNs are the same in the pie chart.*

We thank the reviewer for this comment. To classify CSF-cNs as ventral, dorsal, or lateral, a custom script in FIJI (ImageJ) was created. The user manually defines the dorsal and ventral regions, and the remaining area in the field of view is automatically classified as lateral. Since we observed no evidence of lateralization of CSF-cNs along the rostrocaudal axis of the spinal cord, the remaining percentage of CSF-cNs was divided by two, and the result was assigned to each of the lateral quadrants (Left and Right). Moreover, since it is impossible to preserve left/right identity when working with floating sections, this information would be meaningless in our preparation. This is now explained in the methods section and we added a sentence in the related method section.

In the revised manuscript, it now reads:

P44-45, L1079-1089:

“Once all the cells were marked, the user was asked to define manually the dorsal and ventral regions on the CC (ventral fiber bundles served as bona fide indicators of SC orientation) and the remaining area in the field of view was automatically classified as lateral. Then, the number of cells in each quadrant and their percentages were automatically calculated, and the cell densities/10 μ m of SC were expressed as $(\text{Total number of CSF-cNs in the field of view} \times 10)/(\text{Number of optical slices} \times \text{Voxel depth})$. Since we did not observe evidence for CSF-cN lateralization along the rostrocaudal axis of the SC, once the percentage obtained for CSF-cNs in the dorsal and ventral quadrants, the remaining ones was divided by two, and the result was assigned to each of the lateral quadrants (Left and Right). Moreover, for our study, we have been working with floating sections and it was therefore not possible to preserve left/right identity.”

- 5) *The authors used 3 (or 6 or 3–6) weeks old mice (line 1174, 386, 911). Since the number of CSF-cNs are reported to decrease during the development (Ref 31), the **rationale to use the present age in histological analyses is unclear.***

Do the authors aim to examine the properties of CSF-cNs of adult or juvenile mice?

Are there any differences in the number between 3 and 6 weeks?

We do not aim to examine the properties of juvenile and adult mice and as a note mice are not considered to be adult at 6 weeks but rather from 3-6 months:

<https://www.jax.org/research-and-faculty/research-labs/the-harrison-lab/gerontology/life-span-as-a-biomarker#>

This is also the reason why we avoid using the term ‘adult’ in the description of our results. The **rationale** to use 3 and 6 week-old mice was first to parallel our electrophysiological experiments (**Figure 1** and **Figure 2** of the manuscript) where recordings were performed in this age range. Second, 6 week-old mice were also tested to assess potential differences in the anatomical properties. However, our analysis did not highlight such differences. In the revised version of our manuscript, we now clarify this point.

We added a comment in the Results section:

P6, L129-131:

“However, since our electrophysiological analyses were carried out in mice aged from 3-6 weeks, 6 week-old mice were also tested to assess potential differences in the anatomical properties but it did not highlight such differences.”

We modified the text earlier referred at line 383

P20, L475-478:

“The SC length 3 weeks-old mouse is ~2.5 cm (25,000 μ m) and based on our confocal images the density of CSF-cNs is ~14 cells/10 μ m. We therefore estimate that ~35,000 CSF-cNs would be present in the whole SC (52,500 for a 3.75 cm SC in 6 week-old mice).”

We modified the text earlier referred at line 911

P44, L1074

“Five 3 weeks mice were used for CSF-cNs quantification in confocal slices. 10 fields of view (FOV) were acquired per animal and SC segment (total 30 FOV/animal).”

We modified the text earlier referred at line 1174

Here we present data from a 3 week-old mice

P56, L1430 - Legend of **Figure 2A**:

- 6) *Which spinal levels of the cervical/thoracic/lumbar cord did the authors analyze for histology and electrophysiology? The rostro-caudal levels within the cervical, thoracic and lumbar cord would have quite different circuits and functions.*

For imaging, we typically focus on the medial levels of each region, deliberately avoiding the most caudal and rostral poles of each segment. A detailed analysis of the connectivity patterns of CSF-cNs at various levels and sublevels of the spinal cord, as well as in the rhombencephalic regions, is currently underway in the lab, and these results fall outside the scope of this manuscript.

For electrophysiological studies, the recordings were carried out on acute slices obtained from a given region (cervical, thoracic and lumbar) without distinction of the rostro-caudal level (avoiding the caudal and rostral poles). Although we agree with Reviewer #2’s comment about the difference in the circuits and functions at each level, we did not observe differences in the CSF-cN electrophysiological properties within a region.

All our data are presented as boxplots with whiskers and we also illustrate single data points for each recorded cell. Since, the data distribution appears homogeneous (*i.e.* without data clusters) in a given segment, we are confident that CSF-cN properties are homogenous within a region. We clarify this point in the revised version of the manuscript.

P34, L830-832:

“SC were cut in three segments corresponding to cervical, thoracic and lumbar regions. We typically focus on the medial levels of each region, deliberately avoiding the most caudal and rostral poles of each segment.”

P39, L937-938:

“Lumbar, thoracic, or cervical SC acute slices (not distinguishing the rostro-caudal position) prepared from wild type or PKD-tdTomato mice were transferred in a recording chamber...”

- 7) *Fig. 2D: the definition of “the distance from the central canal” is unclear. Is the distance from the center or the edge of the central canal to the center of the soma? The reason why the distance was examined is also unclear. What does the difference in the distance functionally mean?*

Definition: We thank the Reviewer #2 for this comment since the accurate definition of this distance is important. In the revised manuscript this part of the Methods section now reads as follow

P44, L1077-1093:

“In brief, the user was prompted to click in the center of CSF-cNs on a Maximal Projection image while being able to navigate through the Z-stack in order to accurately resolve closely packed cells. [...] Afterwards the DAPI signal was automatically thresholded to create the contour of the cc, and the Euclidean distance (in microns) of each cell to this contour was calculated using the “Exact Euclidean Distance Transform (3D)” function in FIJI [57] along with the pixel size of the image.”

In brief, we use the DAPI signal of ependymal cells to automatically define the central canal, and the distance measured is from CSF-cN somatic center to the inner edge of the ependymal cell layer (**Figure 3**).

From the original image (1) the CSF-cN channel (2) is used to quantify the center of CSF-cNs by clicking on the cells (3). In parallel, the DAPI channel (4) is used for thresholding (5) and the line corresponding to the ependymal cell layer (6) is used to quantify the euclidean distance of each point (CSF-cNs) to the line (Central Canal).

This is based on:

Distance Transform 3D plugin (FIJI):

<https://imagej.net/plugins/distance-transform-3d>

Points to curve distance FIJI macro:

<https://gist.github.com/lacan/74f550a21ea97f46c74f1a110583586d>

Points to curve distance FIJI macro:

<https://gist.github.com/lacan/74f550a21ea97f46c74f1a110583586d>

Figure 3: Analysis routine used to define CSF-cN distance to the CC

Reason: we performed this analysis for the **systematic** follow up of the article previously published: *Evidence for PKD2L1-positive neurons distant from the central canal in the ventromedial spinal cord and medulla of the adult mouse*. Jurčić N et al., Eur J Neurosci. 2021 Aug;54(3):4781-4803. PMID: 34097332.

- 8) *The authors state that the pH-dependent responses in Suppl Fig. 4 are mediated by PKD2L1 and ASICs in the manuscript. However, without experiments to block PKD2L1 or ASICs (by knockout or inhibitors), the responses are unknown to be mediated by those channels. Please clarify these points.*

In our previous reports obtained from medullar CSF-cNs, we extensively demonstrated that they selectively express the PKD2L1 channel (comparison between wild type and PKD2L1-KO models: Orts Del'Immagine et al., Neurophram 2016) as well as homomers ASIC1a or heteromers ASIC1a/2b based on the sensitivity to *Psalmotioxin* (Orts Del'Immagine et al., J.Physiol. 2012; Orts Del'Immagine et al., Neurophram 2016).

In the present study, we did not carry out such an analysis. However, based on the selective Cre-dependent expression of tdTomato protein under the promoter of PKD2L1, the characteristic morphology of cells recorded around the cc and the unique characteristics (amplitude, open probability and similar pH sensitivity) of the unitary currents, we can safely claim it is carried by the PKD2L1 channel. Further, only neurons around the cc with the expected morphology (confirmed by the cell dialysis with AlexaFluor 488 a cytosolic fluorescent probe added to the intracellular pipette solution).

Regarding ASIC expression and activation, the current kinetics are also similar to those demonstrated in the above mentioned studies and here we can also claim they are carried by ASICs. Finally, in a recent study in mice, Yue and Collaborators (Nature, 2024) conducted a transcriptomic analysis of spinal CSF-cNs data and their data support our claim (see Response to Reviewer #3, Point 10; below). We now added a supplementary Figure on gene expression in CSF-cNs adapted from the mentioned publication.

In the new version of the manuscript, we now clarify this point:

Result/CSF-cN modulation by extracellular pH through the activity of PKD2L1 and ASICs.

P9-10, L216-220:

*“The current properties observed in our study are in line with those reported in previous reports^{25–27} and suggest that they are mediated by PKD2L1 channels and ASIC. An observation that is further supported by the transcriptomic analysis recently carried out by Yue and Collaborators¹¹ where gene for PKD2L1 and ASICs 1 and 2 can be found (see **Supplementary Figure 3A and Discussion**).”*

Discussion/Along the spinal central canal, CSF-cNs exhibit similar intrinsic properties

P24, L565-574:

*“Although we did not perform recordings in CSF-cNs obtained from PKD2L1 knock out mice, we are confident that the unitary current recorded in spinal CSF-cNs is carried by PKD2L1. The unitary current was systematically and only observed in the recorded neurons that exhibited the characteristic morphology of CSF-cNs visualized from the expression of tdTomato fluorescent and confirmed using soluble fluorescent markers (Alexa488 or 594 hydrazide for PKD-tdTomato or wild type mice, respectively) added to the recording intracellular solution mice. Further, it exhibits the characteristic electrophysiological properties (large unitary current amplitude and low open probability) as well as the pH-sensitivity also reported in medullar CSF-cNs (and see **Supplementary Fig. 3A**).*

P24, L577-582:

[...]“We did not characterize the ASIC isoform expressed in spinal CSF-cNs using pharmacological tools. However, based on the similarities of the current kinetics between our recordings and those reported for medullar CSF-cNs(13,54), one can suggest that spinal CSF-cNs would also express ASIC1a homomers or ASIC1a/2b heteromers. This assumption is further supported by the recent report by Yue and collaborators(11) (see also **Supplementary Fig.3A**).”

- 9) *Why are the electrophysiological data of PKD2L1 and ASICs placed in supplemental but not in the main figures? Are they already reported somewhere? Please clarify it.*

The analysis of CSF-cNs sensitivity was reported in the mouse by our group for medullar CSF-cNs but so far not along the spinal cord cc axis. The reason for presenting these data in a supplementary Figure was only based on the figure number limits defined in the manuscript guidelines. In agreement with the Editor this Figure can be included in the main manuscript as **Figure 4**.

- 10) *The authors state that the ventral CSF-cNs are immature compared to lateral CSF-cNs. Which population of CSF-cNs did the authors record in the electrophysiological experiments? Dorsal, lateral or ventral CSF-cNs?*

The claim for the presence of CSF-cNs showing different levels of maturity between the dorso-lateral and ventral regions around the cc is based on the literature (see above comment to Reviewer #1 Point 7 and 8) and dreferrs to postnatal mice. **It was indeed shown in postnatal mice** that ventral CSF-cNs would express immaturity markers and exhibit single-spike firing, two features related to neuronal immaturity. In contrast, dorso-lateral CSF-cNs express maturity markers and present tonic firing (Petracca *et al.*, 2016, DiBella *et al.*, 2019).

In our previous reports, we also analyzed CSF-cN maturity profile at the phenotypical and functional level (Orts Del'Immagine *et al.*, 2016 & 2017; Riondel *et al.*, 2024) **but in older mice**. We show that although two subpopulations (immature vs. mature) of CSF-cNs can be found in older mice based on protein markers, these two groups would not cluster in specific regions around the cc.

In the present study, we recorded neurons around the cc and in the different quadrants. However, although we did observe differences in the firing pattern, channel and receptor expression between CSF-cNs from different segments we could not correlate these differences to a given localization around the cc.

This was stated in the original manuscript and we added a comment in the revised manuscript at the beginning of the Result section

Introduction

P4, L72-74:

“Nevertheless, and in contrast to what has been reported for juvenile mice, there is no correlation between CSF-cN phenotypical and firing properties and their localization around the cc.”

Results

P7, L151-156:

“Here and in the following experiments, recordings have been conducted on CSF-cNs localized in the different quadrant (dorsal, lateral and ventral) around the cc to test whether we could reveal localization-specific properties. However, our data do not indicate that CSF-cNs would group functional clusters. We therefore pool together our data according to their presence in a given segment, not a specific localization around the cc.”

P8, L168-169:

“Nevertheless, in contrast with the situation observed in juvenile rodents, we could not associate a specific discharge pattern with CSF-cN localization around cc.”

P10, L223-225:

“These differences were observed between CSF-cNs recorded in different segments but not associated to a specific localization around the cc.”

Discussion

P19-20, L463-467:

“Overall, spinal CSF-cNs represent a morphologically uniform neuronal population capable of detecting sensory signals that exhibits differential electrophysiological properties along the SC axis potentially to modulate the specific spinal network they are inserted in (Fig. 13). Although differences can be observed between CSF-cNs from different segments, our study does not point to a an organization of CSF-cNs in specific cluster around the cc.”

P22, L519-521:

“Nevertheless in older mice unlike in zebrafish or postnatal mice, medullo-spinal CSF-cN subpopulations do not cluster specifically around the cc [7,13,31], possibly due to developmental reorganization.”

P25, L598-600:

“However, in older mice and in contrast to the data obtained in zebrafish or postnatal mice8, medullo-spinal CSF-cNs are not regrouped in specific clusters but rather distributed around the cc. One can argue that during development and maturation CSF-cNs may redistribute around the cc or might undergo developmental process to change their phenotypical and functional properties. Dedicated developmental studies are required to address this specific point.”

- 11) The channel and receptor expressions are mostly discussed based on the electrophysiological and pharmacological data. It would be supportive if the authors add mRNA expression data of the proposed channels and receptors, e.g. by using the published single cell RNA-seq data.**

Classically, the characterization for the expression of sets of metabotropic and ligand-gated receptors as well as voltage-dependent channels is conducted using electrophysiological recordings with specific activation protocol and combined using pharmacological approaches. Nevertheless, to the attention of Reviewer #2 and the Editor, we provide, as a further support, a transcriptomic analysis (see **Supplementary Figure 3**) adapted from the recent publication by Yue *et al.* (2024) for the expression of the gene (*i.e.* proteins) related to the channels and receptors identified in our study.

For the Transcriptomic analysis and on Editors' request, we added the accession number, the URL and a reference to the original publication for the Methodes published by Yue and Colleagues (Nature, 2024).

We added a section in the Method section.

P48, L1176-1182:

Transcriptomic analysis

Gene expression levels were plotted using custom Python scripts. Gene expression data was parsed from the text file GSE255883_DESeq_norm.txt under accession number GSE255883 (<https://www.ncbi.nlm.nih.gov/geo/query/acc.cgi?acc=GSE255883>), filtering for specific gene names. The extracted data was then visualized as heatmaps using Seaborn and Matplotlib to illustrate expression levels across the different samples (refer to Yue et al., 2024 for the specifics).

- 12) It would be informative if the authors add the data of negative responses to ATP and mGluR in suppl figures.**

As requested by Reviewer #2, we provide below a figure illustrating the negative responses observed following application of ATP (Purinergic receptor; **Supplementary Figure 4**) and of glutamate (mGluRs; **See below**). We propose, in agreement with the Editors, to add the ATP

data as supplementary figure but not the mGluR one. Indeed, these data are part of a follow-up study (finalization of a manuscript), where we characterize the effect of glutamate, ATP and others compounds in the activation of metabotropic receptors leading to the absence of modulation calcium channels but activation of calcium release from intracellular stores.

Figure 5 – Along the spinal cord, CSF-cNs appear to lack calcium channel modulation by metabotropic glutamatergic receptors.

A) Representative I_{Ca} traces elicited at the peak with a VStep to +10 mV from V_h -80 mV and recorded in CSF-cNs from the cervical (A1), thoracic (A2) and lumbar (A3) in control (black traces), in response to pressure application of glutamate (colored traces) and after agonist Wash (grey traces). **B)** Average time courses of I_{Ca} peak amplitude recorded in CSF-cNs for the regions of interest (B1, C: N=1, n=6; B2, T: N=2, n=11 and B3, L: N=1, n=7). Glutamate was applied at 100 μ M by pressure (+ Glu, for 30 s; black bar) and data were normalized to the baseline/control period before agonist application. Colored open circles: mean \pm SD; with SD represented as a colored shaded area. **C)** Summary plots for the normalized peak I_{Ca} density before (CTR), during glutamate application and after agonist washout (Wash) for the regions of interest. Single data points across cells were normalized to the mean value in control and expressed in percent. 100 \pm 42, 104 \pm 40 and 98 \pm 42% (C: N=1, n=6); 100 \pm 30, 97 \pm 29 and 99 \pm 27% (T: N=2, n=11) and 100 \pm 33, 101 \pm 32 and 100 \pm 34% (L: N=3, n=7). Friedman test: χ^2 (Glu)=7.00, 0.54 and 2.57, df=2, p (χ^2)=0.056, 0.761 and 0.276 for C, T and L segments and post-hoc Pairwise comparisons using Wilcoxon rank sum test with Benjamini-Hochberg adjustment to compare currents densities within Regions for CTR vs. Glu and Glu vs. Wash: p =0.094/0.234, 0.619 and 0.813. **D)** Summary boxplots for the I_{Ca} inhibition induced during (Glu) and after (Wash) agonist application for the 3 regions of interest: -5 \pm 6 and 2 \pm 8% (C: N=1, n=6); 2 \pm 8 and 1 \pm 8% (T: N=2, n=11) and -1 \pm 4 and 0 \pm 3% (L: N=3, n=7). Kruskal-Wallis rank sum test: χ^2 (Glu)=5.137 \times 10⁻⁴, df=1, p (χ^2)=0.9974 and posthoc Pairwise comparisons using Wilcoxon rank sum test with continuity correction: Glu vs. Wash: p =0.99. χ^2 (Region)=0.1062, df=2, p (χ^2)=0.9483 and posthoc Pairwise comparisons using Wilcoxon rank sum test with continuity correction: p =0.99 for C vs. T, C vs. L and T vs. L, respectively. In **B)**: colored open circles: in control; mean \pm SD; with SD represented as colored shaded area. In **C)** large filled colored circles: mean \pm SD; small open and filled colored circles: single data points for all the recorded cells at each level. Data are given in the CTR, Glu and Wash order. In **D)**, Data are given in Glu and Wash order.

13) It would be helpful for the readers if the authors summarize the similar and different electrophysiological properties across the spinal levels in a table or diagram in the last part of the figures.

A new figure summarizing (**Figure 13**) the electrophysiological properties of CSF-cNs along the spinal cord axis has been prepared to be included as a last figure in the revised manuscript.

Minor:

- 14) The title may not appropriately convey the information of the present study: “functional” may be replaced with “electrophysiological”, since no “functions” are identified in this study. “Intraspinal sensory neuronal population” is vague: this may be replaced with “cerebrospinal fluid-contacting neurons (in the mouse spinal cord)”.

We agree with Reviewer #2’s comment and will follow his/her advice. In the revised manuscript, the title now reads:

“Cerebrospinal Fluid-contacting neurons are sensory neurons with uniform morphological and region-specific electrophysiological properties in the mouse spinal cord”

- 15) Please carefully check if **suppl Fig. 1–4** are precisely indicated throughout the text and the figures.

We thank Reviewer #2 for the careful reading of our manuscript and pointing out these mistakes. We have accordingly updated and corrected the reference to the supplementary figures.

- 16) The data using ChAT-Cre mice (Suppl Fig. 3?4?) are not explained in Results.

Our comment below also applies to Point 22.

We thank Reviewer #2 for her/his insightful comment. Although Supplementary Figure 4 (now **Supplementary Fig. 2**) is just a positive control of our clearing protocol and light-sheet imaging acquisition protocol, we have now rephrased the results section as follows. Instead of:

“[...] we faced technical limitations and reached microscopy optical limits to resolve and identify single CSF-cN somatas (see **Supplementary Fig. 2** in the revised manuscript).”

It now reads:

P5-6, L117-124:

“[...] we faced technical limitations and reached microscopy optical limits to resolve and identify single CSF-cN somatas (see **Supplementary Fig. 2**). To validate our clearing technique and light-sheet microscopy acquisitions, we conducted two additional experiments. First, we performed viral delivery into the ventricular system of ChAT-tdTomato mice to infect CSF-cNs, allowing us to compare the size of cholinergic neurons and CSF-cNs within the same preparation (Supplementary Figure 4A and B). Second, we cleared the spinal cords of ChAT-tdTomato mice and imaged them using the same settings previously applied to CSF-cNs. Under these conditions, we observed that the technical limitations of light-sheet imaging affect CSF-cNs but not other neuronal populations, likely due to their size differences (**Supplementary Fig. 2A and B**).”

Change accordingly in the text

- 17) Line 125 etc.: please specify what N = and n = means. The number of mice, slices, cells? This information has been specified in the methods section.

P47, L1140-1143:

“All data are expressed as mean±SD or SEM, as specified, and N and n correspond to the number of animal and data samples, respectively.”

In the revised manuscript, we have added in the Result section at their first mention what these parameters refers to.

P7, L145-147:

“Consistent with previous studies in the brainstem(12,22), spinal CSF-cNs have a high input resistance (R_m , 3.6 ± 2.3 G Ω ; $N=191$, $n=659$; N and n for the number of animals used and the sample size, respectively)”.

We also explain these parameters explicitly at their first mention in the legends for **Figure 3**.

P57, L1439-1440:

“($N=10$ for the number of animals, $n=21$ for the sample size) “

18) Line 145: please explain the abbreviation V_h in the first place.

The abbreviation has been explained at the first place mentioned with V_h standing for holding potential

P8, L172

19) Typos in line 255, 384, 1460.

Typos have been corrected:

Line 255 (P12, L290): bracket deleted. Reads now: *“[...] Ca_v selective blocker. In juvenile rat [...]”*

Line 384 (P20, L474): the comma after “population” has been deleted

“[...] We show that CSF-cNs form a dense interconnected neuronal population with approximately 10 to 20 cells per 10 μ m of tissue depth across the entire CC axis. [...]”

Line 1460 (P74 L1879): the ‘l’ in Material was changed from italic to regular font.

“[...] “Maximal projection” [...]”

20) Line 472–474: are the data of different firing patterns between C and L-CSF-cNs shown in figures and Results?

To describe the data of cells with different firing patterns, we have initially prepared Table 2 that shows the proportion of single spike vs. tonic firing for each segment and presented the data in the text (original manuscript).

“AP discharge patterns were also assessed. In line with previous findings in juvenile rats(9) and postnatal mice(8), spinal CSF-cNs exhibit either tonic or single spike discharges. Following a current injection step (+20 pA, 200-500 ms duration and membrane potential at -60 mV, DC current: -10 to -15 pA), 58% of the neurons showed tonic AP firing (38 cells out of 65 recorded), while 42% fired a single AP (27 cells out of 65 recorded; Table 2). Comparative analysis revealed that most L-CSF-cNs and C-CSF-cNs had single AP discharges while T-CSF-cNs exhibited a primarily tonic pattern (Table 2).”

In the revised version we have prepared a new figure (**Figure 3**) and accordingly modified the manuscript. It now reads:

P7-8, L161-167:

*“AP discharge patterns were also assessed (**Fig. 3**). In line with previous findings in juvenile rats9 and mice [8], spinal CSF-cNs in older mice exhibit either tonic or single spike discharges. Following a positive current injection step (+10 pA, 200-500 ms duration) from RMP, 58% of the neurons showed tonic AP firing (38 cells out of 65 recorded), while 42% fired a single AP (27 cells out of 65 recorded; **Fig. 3**). Comparative analysis between regions revealed that most*

L-CSF-cNs and C-CSF-cNs had tonic AP discharges while T-CSF-cNs exhibited a primarily single-spike pattern (Fig. 3B).

21) Line 635, 638: *tdTomato mice are not “flex” mice.*

We agree with Reviewer #2, the tdTomato transgenic mouse model is not *flex* but **floxed** (flanked by LoxP sequences). The manuscript has been appropriately corrected.

22) Line 642: *the data using AAV are not explained in Results.*

See answer to **Point 16** above.

23) Line 692, 695, 699: *what is /ea?*

For more clarity, we now define this abbreviation when first used.

P35, L850: *“After washing in 0.1M PBS (3 times; 2h/each; ea), samples were transferred”*

24) Line 718, 719, 720: *z-Stack > z-stack*

P35, L837: **Corrected to ‘Z-stack’**

25) Line 754: *if wildtype mice were used, how were the CSF-cNs identified? Are the data of Pkd2l1-cre::tdTomato and WT mice combined in the figures? Please specify it.*

Our group has been studying CSF-cNs in mice since the early 2000. If we initially conducted our recordings in medullar acute slices obtained from transgenic mice selectively expressing fluorescent protein (eGFP: PKD2L1-IRES-Cre::floxed-eGFP) in CSF-cNs, we have now gained sufficient expertise to also record them in wild type mice based on their localisation around the central canal and shape.

Nevertheless, as stated in the Method section *“CSF-cN visualization and recording of their intrinsic properties”*

P39, L942-945:

“CSF-cNs around the CC were visualized using a computer controlled digital camera (HQ2 CoolSnap, Photometrics, SciCam, Scientifica) under epifluorescence illumination (exc. 520 nm/em. 610 nm, tdTomato fluorescence) and/or IR illumination with a 40x or 60x objective.” Finally, all our recordings were performed by adding a Alexa488 or 594 Hydrazide compound to the intracellular solution.”

P39-40, L958-964:

“The recording of CSF-cNs was confirmed based on the characteristic morphology observed (small round soma close or within the ependymal layer and a large dendrite ending in the CC with a round protrusion) from the cytosolic tdTomato as well as from Alexa488 or 594 Hydrazide (added to the intracellular solution, 10 µM; Invitrogen) fluorescence. The presence of spontaneous PKD2L1 channel activity was monitored as a further control.”

As mentioned in the Methods section and in the manuscript, and depending on the mouse availability, recordings were performed with either wild type or PKD-tdTomato mice and data obtained with the different animals were combined since the expression of the tdTomato fluorescent protein would not modify the electrophysiological properties of CSF-cNs.

To clarify this point and answer to Reviewer #2 possible concern, we added in the Methods section the sentence:

P40, L962-964:

“We did not observe a difference in the electrophysiological properties of CSF-cNs recorded in acute slices obtained from wild type or PKD-tdTomato mice.”

26) Line 848, 878, 879, 1302, 1316, 1321 etc: *Kainate > kainite*

Reviewer #2 must ask to write ‘Kainate’ with a lower case ‘k’ (kainate, not kainite). If so we have accordingly modified this point in the revised version of the manuscript.

27) Line 1303, 1327: *Glutamate > glutamate Corrected*

28) Line 1460, 1568: *PKD-tdTomato > PKD2L1-Cre::tdTomato.*

Detailed denomination of transgenic mouse models can be quite long. We therefore provided the extended model denomination in the Methods section “Animal models” (P33, L795-804) associated with shortened ones that were subsequently used in the rest of the manuscript for simplicity.

P33, L795-803:

“We used wild type C57 Black6J (Charles River), PKD2L1-Cre (**PKD-Cre**: *Pkd2l1^{tm1(cre)}*; MGI ID: 6451758; a generous gift Emily Leman), and Choline Acetyl Transferase-Cre (**ChAT-Cre**: *Chat^{tm2(cre)Lowl}*; The Jackson Laboratory, MGI ID: 5475195; RRID:IMSR_JAX:006410) mice. PKD- and ChAT-Cre animals were cross-breed with floxed-tdTomato mice (*Gt(ROSA)26^{Sortm14}* (CAG-tdTomato)^{Hze}, The Jackson Laboratory, MGI ID: 3809524; RRID:IMSR_JAX:007914) to generate PKD-Cre (**PKD-tdTomato**)- and ChAT-Cre::floxed-tdTomato (**ChAT-tdTomato**) mice and to selectively express the tdTomato fluorescent protein in the neuronal population of interest. Animals of either sex were used for histology as well as for electrophysiological recordings (3-6 Weeks old mice).”

29) Please carefully present the asterisks and bars throughout the graphs and figure legends as follows:

We apologize for this oversight, and went carefully through the manuscript to verify and correct for the proper presentation of the asterisks. The text of the figure legends and the figures have been accordingly modified for more clarity (see revised manuscript and figures).

We decided to give in the text and legend the exact p-values and accordingly illustrate the asterisks on the graphs.

- Line 1228: there are no * and ** in the graph (Fig. 4C).

In Figure 4C (**Figure 6C**, in the revised version) We compared the difference between K_V pic current densities for the Early (E, \$; Transient) and Persistent (P, #) components using a linear multi effect statistical test (lm function in R statistics). We set Level (Cerv, Tho and Lumb) and Current (Pers and Trans) as factors to compare the current densities (Dens).

The test gives no statistical differences for the comparison of current densities between Levels nor within a Level for current types (Pers and Trans).

We corrected this error that is due to edition (cut and paste) issues.

- Line 1236–1237: there is only **** in the graph (Fig. 4D).

Same error as above. Corrected.

- Line 1258, 1283, 1293, 1335, 1341, 1406, 1416: please specify what *, **, *** in the figures (Fig. 5C, 6B, 6D, 7E1, 7E2, 9C, 9D) mean. **It is unclear which groups do the bars and the asterisks in the graphs compare.**

Figure 5C (now 7C): we compare the Ca_V current densities between CSF-cNs recorded in the cervical, thoracic and lumbar segments.

Figure 6B (now 8B): we compare the GABA-mediated current densities between CSF-cNs recorded in the cervical, thoracic and lumbar segments.

Figure 6D (now 8D): we compare the glycine-mediated current densities between CSF-cNs recorded in the cervical, thoracic and lumbar segments.

Figure 7E1 (now 9E1): we compare the amplitude of the glutamate-mediated depolarization between CSF-cNs recorded in the cervical, thoracic and lumbar segments.

Figure 7E2 ((now **9E2**): : we compare the number of action potential triggered by the application of glutamate in CSF-cNs recorded in the cervical, thoracic and lumbar segments.

Figure 9C/D (now **11C/D**): we compare the Ca_v current densities in control, in the presence of baclofen and after washout of the agonist between CSF-cNs recorded in the cervical, thoracic and lumbar segments.

We have modified this figure to now present the currents normalized to the mean control value of the population. We also re-analyzed the data using a Friedman statistical test, more appropriate to compare for each segment the effect of baclofen.

For the panel 9D (now **11D**), on the left part of the graph we compare the percentage of inhibition in the presence of baclofen and after washout. Here we analyzed the data using a kruskal walis test for the Bcl effect (the condition) as well as the comparison in the 3 segments of interest (the level).

The same comparison has been illustrated in **Figure 10C and D** (now **12C, D**).

- Line 1322: no * in the graph (Fig. 7C)

Fig. 9C - This is a 'cut and paste' error; Legend corrected

- Line 1356: no ** in the graph (Fig. 8B now Fig)

- Line 1369, 1375 (Fig. 8D1, D2): * should be added in the graphs?

Now Fig. 10B, D1 and D2 - We have clarified these points on the graph and in the legend.

- Line 1440: * $p < 0.05$? It is unclear which groups do the bars and * in the graph compare (Fig. 10C).

We have modified the figures for more clarity

- Line 1450: no * and ** in the graph (Fig. 10D).

We have modified the figures for more clarity

- Fig. 1D: it is unclear which groups do the p values compare. * is not consistent with the p values.

The Kolmogorov-Smirnov (K-S) test responds differently to sample size. In very large datasets, the K-S test can detect even minute differences that may not be biologically or practically relevant. Therefore, a more stringent threshold (e.g., $p < 0.001$) is necessary to avoid identifying trivial effects as significant. For highly powered tests like K-S in large datasets, adjusting the threshold helps prevent overestimation of significance.

P45, L1098-1100:

*"For comparison of CSF-cN distances to the CC at the different SC levels, Kolmogorov-Smirnov test was used. Since K-S test is extremely powerful over large data sets, results were considered significant when * $p < 0.001$."*

- Fig. 7B: it is unclear which groups does *** in the graph compare.

We performed a lnm statistical test followed by a pairwise posthoc analysis (R Statistics; see Methods section, "Statistical Analysis"), where data were compared between Conditions (glutamate without or without DNQX) and between Region (cervical, thoracic and lumbar). The '***' on the top of the graph is referring to the statistical difference between recordings performed in the different Conditions in the presence of 'glutamate' application alone and 'glutamate application in the presence of DNQX'. The figure (**Figure 9B** in the revised version) has been accordingly modified to clarify this point.

We now distinguish the statistical difference between glutamate alone and glutamate with DNQX with violet asterisks and bar and the difference for glutamate responses between segment with black asterisks and bars.

3) Reviewer #3 (Remarks to the Author):

Manuscript “An intraspinal sensory neuronal population with homogenous morphological and region specific functional properties in the mouse” by Crozat et al., using morphological and in vitro electrophysiological techniques characterized the properties of a kind of neurons that locate around the spinal central canal and their dendritic protrusions contact cerebrospinal fluid (CSF), so-called CSF-contacting neurons (CSF-cNs), in mice.

They found that spinal CSF-cNs exhibit a conserved morphology across species and are uniform across SC segments in the mouse. They distribute densely along the entire spinal cord axis and are primarily located ventrally. Functionally, along the spinal cord, they share similar intrinsic and chemosensory properties with mouse medullar CSF-cNs. The neurons express NaV, KV, LVA and HVA Ca²⁺ channels. They also express functional classical inhibitory and excitatory ionotropic synaptic receptors.

The authors conclude that the spinal CSF-cNs represent a morphologically homogeneous sensory neuronal population, but functionally different along the spinal cord axis.

The study will for sure contribute to the better understanding of CSF-cN functions.

In general, each step of the experiments was carefully designed and conducted, and the results were objective, and the drafting of the manuscript is concise and thorough. The work with tissue clarity technique is beautiful. **The work is of interest to others in the community. I don't have major comments but some minor ones.**

- 1) *Some abbreviations in the Abstract do not have a complete name, such as PKD2L1, TRP and ASICs. Please give these abbreviations complete names when first used anywhere in the manuscript even in the abstract.*

The abstract has been updated and abbreviations replaced with full denomination. We went through the manuscript and gave full names at the first use of an abbreviation.

- 2) *Page 3, line 60, concerning the function in detecting spinal cord bending, is mainly observed in zebrafish. This needs to be stressed although the related references were given.*

Corrected. It now reads

P3, L58-61:

“These neurons express Polycystin Kidney Disease 2-Like 1 (PKD2L1), a channel sensitive to pH, and osmolarity changes (chemosensitivity) [13,25–28] that was also shown, in zebrafish larvae, to respond to CSF flow, and SC bending (mechanosensitivity) [29–31].”

- 3) *Page 4, line 83, “there are” should be “they are”.*

Corrected

- 4) *Page 5, lines 93-94, better to mention what animal species for this statement. The authors seemed to have included old articles (although some relatively new). Please update the references especially including those using primates.*

This part has been modified for more clarity. It now reads

P5, L102-104:

“CSF-contacting neurons (CSF-cNs) are found along the entire cc in lamprey⁶, zebrafish larvae [5,21], turtle [42] rat [9], mouse [7,11,13–16,36] and macaques [4,10,43] where they exhibit a consistent morphology.”

- 5) *Page 5, line 102, “(Fig.1B-D, Right ventral view)”. Is 1D from ventral view or side view?*

We thank Reviewer 3 for this insightful comment. Indeed, the right view of **Figure 1B-D** is ~90° rotated regarding the views presented in the left panels. Due to inherent difficulties in SC manipulation during the clearing process and the light sheet imaging (specifically in whole CNS specimens) some torsion may exist thus giving the impression of incoherence between the right panels (Panel 1D, right). In any case this is not a ventral view.

We have now changed “ventral view” by “caudo-rostral view”, which fits for all the right panels in **Figure 1**.

- 6) Page 5, lines 113-114, “A larger proportion of CSF-cNs is located in the ventral CC (Fig. 2A, Right; 2C) and neurons are closer to the CC in cervical and lumbar segments but further away in the thoracic one (Fig. 2D)”. **Isn't that in the lumbar segment the CSF-cNs further away from CC according to Fig. 2D (the blue dots)?**

We thank Reviewer #3 for this comment.

In Figure 2D, the blue dots represent data for the thoracic segment. What can be seen is that thoracic CSF-cNs (blue dots) are slightly, although significantly, further away from the CC than cervical or lumbar CSF-cNs. Our statement is therefore correct.

- 7) Page 6, line 128, “...compared to T- and C-CSF-cNs (Table 1)”, should be “...compared to L- and C-CSF-cNs (Table 1)”.

We agree and have corrected this mistake.

- 8) Page 6, line 132, “...depolarized CSF-cNs in anterior regions”, better to use “rostral regions”.

Modified accordingly.

- 9) Page 7, lines 141-142: “the findings suggest that CSF-cNs share similar intrinsic properties but exhibit differential excitability along the spinal cord”. Does this mean that “excitability” is not an “intrinsic property”.

We agree with Reviewer #3 that excitability as intended here corresponds to the capability for a neuron to generate action potential and with which pattern. Since this response is triggered by an external stimulus (*i.e.* DC current injection) and not by synaptic inputs it corresponds to an intrinsic property mediated by active conductances.

We therefore modified the revised manuscript to distinguish intrinsic properties between passive (Rm, Cm ...) and active ones (spiking capabilities).

In the revised manuscript this reads as follow:

P10, L221-225:

“Overall, our findings suggest that CSF-cNs are highly resistive neurons showing passive properties as well as firing patterns that differ along the cc. They further exhibit shared chemosensory functions by responding to variations in extracellular pH. These differences were observed between CSF-cNs recorded in different segments but not associated to a specific localization around the cc.”

- 10) Page 8, line 182, can the authors specify which subtypes of ASICs are expressed in mouse CSF-cNs? In lamprey it is primarily ASIC3.

In our previous reports (Orts Del'Immagine *et al.*, 2012 and 2016), we indicated that medullar CSF-cNs expressed the ASIC1a homomers and ASIC1a/2b heteromers (sensitivity to the psalmotoxin). In the present study we did not use pharmacological tools to determine the ASIC subtype expressed in mouse spinal CSF-cNs. However, based on the similar current kinetics observed in our study, we can assume that spinal CSF-cNs would also express ASIC1a homomers and ASIC1a/2b heteromers. This assumption is supported by the transcriptomic data recently published by Yu *et al.* (2024). We present below a raster plot based on the mentioned study that confirms the presence of ASIC1 and 4 genes (score of 9 and 8.1, respectively for samples 3 and 4 and see the **Supplementary Figure 3** about gene expression in spinal CSF-cNs adapted from the mentioned publication).

In the revised version of the manuscript, we now state:

P9-10, L216-220:

“The current properties observed in our study are in line with those reported in previous reports(25–27) and suggest that they are mediated by PKD2L1 channels and ASIC. An observation that is further supported by the transcriptomic analysis recently carried out by Yue and Collaborators(11) where gene for PKD2L1 and ASICs 1 and 2 can be found (see Supplementary Figure 3A and Discussion).”

11) Page 11, line 255, there is an extra “(“ *The extra ’)*’ was deleted.

12) Page 15, line 354, please specify “mACh-Rs.s”, or it is one “s” extra. *The extra ‘s’ was deleted.*

13) *In the Discussion, the authors need to compare the morphology and the distribution of CSF-cNs between different animal species, especially with those at a higher level, such as primates.*

There are such articles e.g.:

Liu et al. PKD2L1-expressing cerebrospinal fluid contacting neurons in spinal cords of rodents, carnivores, and primates. *Int. J. Mol. Sci.* 2023, 24, 13582. <https://doi.org/10.3390/ijms241713582>

Tonelli Gombalová Z et al. Majority of cerebrospinal fluid-contacting neurons in the spinal cord of c57bl/6n mice is present in ectopic position unlike in other studied experimental mice strains and mammalian species. *j. comp. neurol.* 2020, 528, 2523–2550. doi: 10.1002/cne.24909.

We have addressed and discussed this point in the revised manuscript. It now reads

P20-21, L480-501:

“CSF-cNs have been observed in several vertebrate species, where they show similar morphology as well as distribution around and along the cc and. In lampre[6], zebra fish [5,20] and turtle [42], they mainly exhibit a triangular, small soma inserted in the ependymal cell layer with a short dendrite projecting to the cc (but see below for lamprey). Similar features are found for CSF-cNs in postnatal rodents [8,9,14]. With aging, their morphology and localization changes, and one can find either intra-ependymal neurons, embedded within the ependymal cell layer, or subependymal, located below the cc, with a longer dendrite extending into the cc lumen [7,12,14,33]. In mice, spinal CSF-cNs are predominantly found in the ventral region, comprising about 60% of the total. This ventral localization contrasts with the lateral distribution observed in the medulla [7] (but see also Kútna and colleagues [34]) and suggest an anatomical reorganization along the medullo-spinal axis. We also confirm the presence of PKD2L1-expressing neurons (tdTomato+) in more distal ventral locations along the SC [14]. Note that Tonelli Gombalová and colleagues indicated that in C57 Black6/N mice, in contrast to the J substrain, an important proportion of CSF-cNs is observed in ectopic positions away from the cc due to potential dysfunction of Crb1 and Cyfip2 products in this substrain [48]; our present and past studies were conducted on the C57 Black6/J substrain.

CSF-cNs are also observed in NHP [4,5,10,43] and a recent study indicates that CSF-cNs morphology, density and distribution [10] are similar to that observed in rodents with PKD2L1+ neurons that are also localized in the ventral region of the SC away from the cc [10]. Interestingly in Macaques, CSF-cNs are largely present in subependymal position, exhibit long dendrites and are the only neurons localized around the cc in a hypo-neuronal region enriched with astrocytes and microglia [43].”

14) *In connection with the last point, the authors also need to do a discussion from a translation point of view. Considering that in the majority of humans the central canal is closed and there are no reports as to whether there are CSF-cNs in the human spinal cord, how can the results from the animal studies be translated to human?*

We have addressed and discussed this point in the revised manuscript. It now reads

P21, L501-511:

“One crucial question, that is still controversial due to experimental limitations and contradictory reports, concerns the presence of CSF-cNs and whether they would play a similar sensory function in human SC. First, the cc is thought to have collapsed in adult Humans below the cervical region [49] in older subjects but more recent studies have shown that the cc is preserved along the SC [50–52]. Second, there is no evidence for the presence of CSF-cNs in Human and one of the first genetic analysis of human spinal cord failed to reveal the presence of the gene for PKD2L1 presumably due to experimental limitation [53]. Nevertheless, this remains an open question that needs to be addressed in the future to demonstrate whether this unique neuronal population is conserved in Human and if not whether it has been replaced by another system capable of integrating CSF circulating signal in the SC.”

15) Page 16, line 384, two refs. with a # 33 (one should be 32).

Corrected

16) Page 26, line 610, “CS-cNs” should be “CSF-cNs”.

Corrected

17) Page 27, in the section “Histology and imaging”, please specify what kind(s) of mice were used for imaging study.

According to Reviewer #3's suggestion, we have now replaced the sentence [...] “Animals were injected” [...] by

P34, L818-819

[...]“PKD-tdTomato (**Fig. 1 and 2 and Supplementary Fig. 1 and 2**) or ChAT-dTomato (**Supplementary Fig. 4**) animals were injected” [...]

18) Page 9, line 692, specify what “o/n” means. We specified overnight for o/n

19) Page 37, line 876-881, “For the agonists:...”. This seems not to be a complete sentence.

We changed the sentence to

P43, L1040:

“For the agonists, the following compounds were used:” .

20) Page 39, line 928-929, “Results were considered significant when $*p < 0.05$, $p < 0.01$ ”. Only $*P < 0.05$ ” is enough. The “*” is not needed here. Same principle applies to page 42, line 1003.**

Corrected

P48, L1171-1174:

*“Statistical differences were considered as significant for $p < 0.05$ and we give in the text and figure legends the p values for each test. In the corresponding figures, asterisks are shown as follow: ns, *, **, *** and **** for p non significative, < 0.05 , < 0.01 , < 0.001 and < 0.0001 , respectively.”*

21) Page 48, line 1170, “thoracic segment (B)” should be “thoracic segment (C)”.

Corrected

22) In the figure legends, please specify what “N” respective “n” stands for. They are specified in Table 1 and 2, but not in other method parts or figure legends.

This information has been given in the Method section “Processing, analysis and statistics, Electrophysiology”

Our feeling is that adding this information in the figure legends will make them even longer than they are now. We have detailed that when we describe statistical analysis in the Methods section, in the Results section and the legends when it appears for the first time, In the revised manuscript, we have added in the Result section at their first mention what these parameters refers to. Nevertheless, on request of the Editor, we have removed from the legends the mention to average and data range to reduce their length.

Results

P7, L145-146:

“Consistent with previous studies in the brainstem(12,22), spinal CSF-cNs have a high input resistance (R_m , $3.6 \pm 2.3 \text{ G}\Omega$; $N=191$, $n=659$; N and n for the number of animals used and the sample size, respectively)”.

Methods

P47, L1140-1141:

“All data are expressed as mean \pm SD or SEM, as specified, and N and n correspond to the number of animal and data samples, respectively.”

We also explain these parameters explicitly at their first mention in the legends for **Figure 3**.

P57, L1439-1440:

“(N=10 for the number of animals, n=21 for the sample size) “

23) Page 54, line 1304, “time” should be “time”.

Corrected

24) In general, the figure legends are too long. Is it possible to place the statistical details in the text, but not in the legends?

In the journal Guidelines to prepare Figures and legends, it is recommended to detail the statistics in the legends. However we agree that it ends with very long texts for the legends. On the other hand, we decided to avoid too long results and statistical descriptions in the text to ensure easier reading. Following request from the Editors confirming the need to present statistical results in the legends, we have reduce them by removing average and data range values.

25) Figure 4, please place the voltage step gradients (each V_{step}) used in Figure 4A, similar as Figure 5A.

Now it is **Fig. 6**. The figure has been accordingly modified and we took this opportunity to modify also **Fig. 5**.

26) Figure 7B, is “Control” the same as “Glu”?

Yes it is and we modified the figure to replace “Control” by “glutamate”.

27) Figure 9, “+ CGP” in red color is a bit confusing. It is the cervical recording that shows red color. It is better to place “Baclofen” and “+ CGP” above the traces in A2 and A3 as well.

28) Figure 10, same as Figure 9, please do the same for “Oxo” and “+ Atr”.

In these two figures as well as in all figures where traces with an antagonist or a channel blocker are presented, we have modified the color used. Now all are violet.

Point-to point response to Reviewers' comments

We would like to thank the reviewers for their valuable comments. Please find below a point-to-point answer to the comments and requests of our revised manuscript. We hope to have adequately address the point raised and that in his revised version our manuscript will be suitable for publication.

Best regards,
Prof. N. Wanaverbecq

Reviewer #1 (Remarks to the Author):

The authors have addressed most of my concerns regarding overstating observations/ conclusions and the data is technically sound.

However, I remain unconvinced that the results:

- consistent CSF-cN properties along the spinal cord, with only some variations in ionotropic current densities.
- the functional implications of which have not been addressed

constitute a significant advance bringing new biological insight or a conceptual advance.

See answer to Reviewer #2's comments that raised similar issues.

Regarding the **functional implications for the electrophysiological differences** reported in our study. We agree that we do not provide experimental evidence supporting the functional relevance of the observed differences but raised these points in the discussion see below).

It was necessary to first assess that differences exist or not between CSF-cNs from different SC segments (the scope of our study) to then address their functional relevance. This latter point requires dedicated experiments, beyond the scope of the present study, to test and compare between segments the consequence on CSF-cNs physiology and on the activity of poorly characterized postsynaptic partners. At that stage, we can only speculate on the functional implications of the reported properties.

One has to take into account, that, in mice, the analysis of CSF-cN connectivity, and as a consequence on their activity and on the network, has only recently started to be investigated and is technically challenging due to SC anatomy and the need to develop transgenic approaches. We have initiated (see Gerstmann *et al.*, 2024) and are conducting this characterization by combining slice electrophysiological recordings with the so-called ChannelRhodopsine-2 Assisted Circuit Mapping (CRACM, Petreanu, *et al.*, 2007) approach. We are finalizing a study on lumbar CSF-cNs connectivity and pursue such studies to resolve this crucial question in the future. Further, when considering the high membrane resistance ($> 1 \text{ G}\Omega$ i.e. 20 pA current leads to 20 mV depolarization) of CSF-cNs, even small difference in ionic conductances will have a large influence on their activity and excitability.

For Firing Patterns

P25-27, L618-644:

"We also observed that C- and L-CSF-cNs exhibit mainly tonic firing while thoracic neurons mostly show single-spiking pattern. [...] This result suggests different properties and ionic conductances expression for CSF-cNs along the rostro-caudal axis. Due to the recording conditions and dedicated protocols implemented to analyze on one hand AP firing (potassium-based intracellular solution for current-clamp recordings) and on the other isolated ionic conductances (cesium- and TEA-based intracellular solution in voltage-clamp recordings to isolate Nav), we cannot correlate spiking profile and Nav densities. This might also depend on the specific network CSF-cNs are inserted in as well as to the exposure to given bioactive signals. Further, CSF-cNs are present in the so-called spinal niche in contact with the CSF and appear to interact with ependymal cells [11]. One could suggest that they might change

functional properties as a function of the physio-pathological state as recently demonstrated following SC injury [11]. The observed difference could also be explained by the properties of the networks CSF-cNs are inserted in. However, one has to remain cautious in concluding about AP discharge activity when using whole-cell patch-clamp technique especially in neurons with small somatas. An answer to this point could be addressed in vivo using imaging approaches (calcium and voltage sensitive probes) or electrophysiological recordings.”

For the calcium signaling

P28, L673-677:

“We also found that thoracic CSF-cNs have the highest expression level of CaV channels, as reflected in the larger amplitude of I_{Ca} densities recorded in this region. This suggests that Ca²⁺ signaling is differentially regulated along the SC axis, with higher CaV density and calcium signaling in thoracic segments.”

For the synaptic receptors

P30-32, L728-774:

Our results further indicate differences “Our results further indicate differences in the current densities of glycine and GABA_A receptors-mediated responses that is highest in thoracic CSF-cNs. We also report in lumbar CSF-cNs lower current densities for glutamate (AMPA/kainate)-mediated responses underlying the observed lower excitatory action of AMPA/kainate receptor activation. Due to the recording configuration (whole-cell patch-clamp and artificial intracellular chloride concentration), it is difficult to discuss the functional consequence of the inhibitory signaling onto spinal CSF-cNs (see also Riondel and Colleagues [36]). Nevertheless, altogether, these results might imply that the excitatory and inhibitory drives onto CSF-cNs differ along the cc axis and that CSF-cNs would be differentially modulated through synaptic inputs within local spinal networks. The functional consequence of this differential excitatory/inhibitory drives is unknown but opens future lines of research where the underlying connectivity leading to a differential integration within segment specific local networks needs to be characterized. We are currently addressing these aspects in dedicated morphological and functional studies. Additionally, CSF-cNs along the spinal cc axis express muscarinic and nicotinic cholinergic receptors. Despite the small amplitude of these glutamatergic and cholinergic currents, due to the high CSF-cN input resistance, receptor activation can induce significant depolarization and even trigger APs. Thus, activation of glutamatergic and cholinergic ionotropic receptors in CSF-cNs likely modulates their excitability. However, the effects induced by the activation of these two receptors are different suggesting that CSF-cNs might integrate or code differentially the synaptic inputs they receive depending on the network they are inserted in. Within the spinal tissue numerous GABAergic, glutamatergic and cholinergic interneurons are present and shape spinal activity and outputs (somatic and autonomous motor systems). Recent reports [11,15,16] and our data indicate that CSF-cNs express the major synaptic receptors and would receive inputs from or project to spinal interneurons. Therefore, CSF-cNs might be key player in the SC circuitry in the mammals by bidirectionally interacting with different neuronal populations to regulate CNS activity.

Spontaneous synaptic activity has been observed in medullar CSF-cNs, but sparsely in spinal CSF-cNs, raising questions about the source of neurotransmitters and their release conditions in activating spinal CSF-cNs. Neurotransmitters might be released via synaptic contacts with glutamatergic, GABAergic, and cholinergic neurons within the SC, nevertheless, except for recurrent connectivity among CSF-cNs as source for GABA [15,16], functional contacts with CSF-cNs remain to be definitively demonstrated. Another potential origin could be paracrine release from neurons in the surrounding parenchyma, but this seems unlikely, as no changes in baseline current were observed upon exposure to selective antagonists. However, one cannot rule out a loss or ‘dilution’ of such paracrine transmission in in vitro models where slice preparation is perfused with aCSF. Additionally, neurotransmitters have been shown to be present in the CSF and might activate CSF-cNs via their protrusions as suggested by a recent study indicating that CSF-cNs can be activated by κ -opioids released by neighboring cells -

[11]. *Such a route of activation remains to be demonstrated. Nevertheless, there are growing anatomical evidence indicating that CSF-cNs appear integrated within specific spinal network and one major challenge in the coming years will be to further characterize CSF-cN presynaptic partners and understand how they are activated. We recently used monosynaptic retrograde neuronal tracing [16] to show that L-CSF-cNs are primarily contacted by GABAergic neurons, with some glutamatergic input as well. Extending this type of characterization to the entire SC and testing functional connectivity in vivo will be crucial.*

Reviewer #2 (Remarks to the Author):

The manuscript has been improved well, but I still have some points that should be improved more as follows:

1. Abstract:

The revised abstract is not clear.

We have edited the abstract in the revised manuscript to follow Reviewers #1 and #2 comments. The abstract now reads.

which part is the background

[Background] *“Cerebrospinal Fluid-contacting neurons (CSF-cNs) are GABAergic bipolar neurons found, in contact with the cerebrospinal fluid along the vertebrate medullo-spinal central canal. They express Polycystin Kidney Disease 2-Like 1 channels (PKD2L1), members of the Transient Receptor Potential superfamily, and were shown to modulate motor activity and therefore suggested to act as a novel sensory system.*

what is the research question

[Research Question] *However, in mice, they remain largely uncharacterized and it is crucial to comprehensively characterize their morphological and electrophysiological properties to determine whether they form a homogenous neuronal population and understand their role in the CNS.*

which part is the result of this study.

[Results] *We show that CSF-cNs are distributed throughout the spinal cord with a uniform morphology and a primarily ventral localization. They exhibit region-specific properties, expression of voltage-dependent and ligand-gated conductances and detect variation in extracellular pH through activation of PKD2L1 and Acid-sensing Ion Channels. They possess GABAB and muscarinic receptors, not glutamatergic metabotropic ones, to modulate Ca²⁺ channels.*

[Conclusion] *CSF-cNs represent unique sensory neurons with a uniform morphology and electrophysiological properties that appear specific to the SC segment inserted in.*

[Perspectives] *The future challenges in the field, will be to elucidate the physiological stimuli activating CSF-cNs and the neuronal network they are integrated in to modulate body function through specific local spinal network.”*

In addition, **the summary of the findings, particularly their importance or novelty, are not well described.** This issue relates to the comment of Reviewer #1-1

We have edited the revised manuscript to follow Reviewers #1 and #2 comments. The summary of findings now reads:

P19-20, L454-477:

“CSF-cNs have been extensively studied in zebrafish larvae [5,20,21,30,31,37,38], in lamprey [6,26,27], and, to some extent, in mice [7,8,13,14,22,23,25,34], rat [9,12] and non-human primates (NHP) [4,5,10]. However, to date CSF-cN properties were largely under-characterized with data reported for different animal models, ages or different SC segments, but not systematically.

Here, we provide an in-depth characterization of CSF-cN properties along the mouse SC from cervical to lumbar segments and find that CSF-cNs possess a conserved and uniform

morphology across species and mouse SC segments, respectively. We indicate that they express, along classical ionotropic synaptic receptors, Na_v , K_v and Ca_v (LVA and HVA types), the latter being modulated by metabotropic GABAergic and muscarinic receptors but not by glutamatergic ones. We further report electrophysiological differences between CSF-cNs from different segments. Finally, although, they share similar chemosensory properties, our study indicates that they exhibit region-specific firing patterns (tonic vs. single-spike) as well as differential calcium signaling and synaptic receptor-mediated excitatory vs. inhibitory drive (Fig. 13).

Our results provide a comprehensive and systematic characterization of CSF-cN morphological and electrophysiological properties along the mouse SC axis. They suggest that because of their shared sensory properties, CSF-cNs would detect and integrate information along the cc in a synchronized manner. While, due to the regional differences along the spinal levels of their electrophysiological properties, they would be differentially regulated by local partners to serve region-specific modulation of the physiological functions controlled by the spinal network they are inserted in. Overall, our data set ground for future studies to address this crucial question and demonstrate CSF-cN function as a novel sensory system in mammalian CNS.”

2. Introduction:

I feel that **the research question and novel focusing points of this study in comparison to previous studies are not clearly described.**

Perhaps this study aims to “**comprehensively**” examine (and found) anatomical and electrophysiological properties of CSF-cNs, particularly focusing on “**regional differences along the spinal levels**” in mice.

This also relates to the comment of Reviewer #1-1, and **they should be clearly stated in the Introduction.**

We have edited the introduction to response to the comments stressing the research question and the novelty of our study in comparison to previous studies. In particular, we added:

P4-5, L83-100:

“Along the SC axis, specific local networks control specific physiological functions, and one can wonder whether CSF-cNs inserted within a given SC segment would exhibit specific morpho-functional properties. Although, this information is crucial to better characterize CSF-cN physiology and demonstrate their role along the SC, such analysis has not been carried out. We therefore conducted a comprehensive and systematic study to examine spinal CSF-cN anatomical and electrophysiological properties from the cervical to the lumbar segments and assess whether regional differences along the mouse SC levels can be observed.

We found that spinal CSF-cNs, primarily located in the cc ventral region, form a dense morphologically uniform neuronal population that shares similar sensory properties to integrate signals circulating in the CSF along the SC. However, they exhibit region-specific electrophysiological features that might serve the specific role they play in a given spinal network. Anatomical and functional evidence suggest that CSF-cNs act as a novel sensory intrinsic to the CNS. Our study provides novel cues on the physiology of spinal CSF-cNs and is crucial for the deeper understanding of mammalian CSF-cNs within the CNS. It set ground for the future challenges in the field to determine their integration and modulation within local spinal and supraspinal networks and to ultimately demonstrate their role in the modulation of body functions both in physiological and pathological conditions.”

3. Data consistency:

Page 7 line 149-: When comparing across spinal segments, T-CSF-cNs displayed a higher Rm and Cm, compared to C- and L-CSF-cNs while the membrane capacitance was the lowest at the thoracic level (Table 1): I could not see a higher Rm and Cm of T-CSF-cNs in Table 1.

We agree, there was a mistake. R_m and τ_m are similar in CSF-cNs from all segments, while C_m is the highest in L-CSF-cNs (*i.e.* lowest in T-CSF-cNs). We have corrected this statement, it now reads:

P33, L798-801:

“When comparing across spinal segments, our data indicate the CSF-cNs exhibit similar R_m and membrane time constant in all segments, while C_m was the highest in L-CSF-cNs (lowest in T-CSF-cNs) compared to C- and T-CSF-cNs (Table 1).”

4. Supplemental Fig. 3:

please indicate each gene name on the left side of the heatmap. For example, which rows of the heatmap represents ASIC1 and 2 (page 10, line 220)? Other genes are not clear as well. We apologize to Editors and Reviewers, this has been an error in the file conversion process when submitting the revision. We have overseen this mistake and the figure should look as follow. We have uploaded a corrected version of Supplementary Figure 3.

A) Polycystine TRPs (TRPPs) and ASIC

5. Line 635, 638:

tdTomato mice are not “flex” mice.

We agree with Reviewer #2, the tdTomato transgenic mouse model is not flex but floxed (flanked by LoxP sequences). The manuscript has been appropriately corrected.

tdTomato gene is not “floxed” in this mouse line (STOP cassette is floxed). The authors use CAG-loxp-STOP-loxp-tdTomato mice.

We removed the word ‘floxed’ and now use the full nomenclature for the $Gt(ROSA)26^{Sortm14}$ (CAG-tdTomato)^{Hze} model:

P33, L798-801:

“Pkd2l1- and ChAT-Cre animals were cross-breed with $Gt(ROSA)26^{Sortm14}$ (CAG-tdTomato)^{Hze} (The Jackson Laboratory, MGI ID: 3809524; RRID:IMSR_JAX:007914) to generate Pkd2l1-Cre and ChAT-Cre::tdTomato mice and selectively express the tdTomato fluorescent protein in the neuronal population of interest.”

6. Line 1460, 1568:

PKD-tdTomato > PKD2L1-Cre::tdTomato.

Detailed denomination of transgenic mouse models can be quite long. We therefore provided the extended model denomination in the Methods section “Animal models” (P33,

L795-804) associated with shortened ones that were subsequently used in the rest of the manuscript for simplicity.

This is very confusing since “PKD-“ also includes genes of the polycystin family (e.g. Pkd1, Pkd2, etc.). In addition, “PKD-tdTomato” means that the authors use transgenic mice expressing tdTomato directly under the PKD promoter. I recommend using the term “Pkd2l1-Cre” (only the first letter of the gene name should be uppercase) throughout the manuscript. The authors may abbreviate it as ***Pkd2l1-Cre::tdTomato*** (or Pkd2l1-tdTomato, though I do not prefer the latter due to the above reason).

We now refer to this transgenic models as Pkd2l1-Cre and Pkd2l1-Cre::tdTomato as well as ChAT-Cre::tdTomato.

Reviewer #3 (Remarks to the Author):

The revised version addressed all my questions. Thus, I don't have any further comments for the revised manuscript.

We thank Reviewer #3 for his positive answer on our revised manuscript and for his agreement to publish our study.

Point-to point response to Reviewers' comments

We would like to thank the reviewevers for their careful reading of our initial manuscript and for their valluable comments. Please find below a point-to-point answer to the comments and requests. We hope to have adequatly address the point raised and that in his revised version our manuscript will be suitable for publicationn.
We apologize for the delay in our response due to unforeseen reasons.

Best regards,
Prof. N. Wanaverbecq

All comments from reviewer é have been addressed.

Reviewer #2 (Remarks to the Author):

The manuscript was well improved though there are typos and grammatical errors as followings.
I have no further comments.

Line 41: SC segment > *spinal cord segment*

Line 95: CSF-cNs act as a novel sensory intrinsic to the CNS > sensory ??

sensory system intrinsic to the CNS

Line 130: However, ; this can be deleted, *Deleted*

Line 463: from different??. *Corrected*

However, to date CSF-cN properties were largely under-characterized with data reported for different animal models, ages or different SC segments, but not systematically

Line 583: Pkd2l&-Cre > Pkd2l1-Cre, *Corrected*

Line 633: corelate > correlate, *Corrected*

We cannot correlate spiking profile

Line 830, 931 etc.: PKD-tdTomato > Pkd2l1-Cre::tdTomato

Corrected thourghout the manuscript.

Further, the pie in Figure 2A have been replaced by boxplot with whiskers and the legend accordingly modified.

Best regards
N Wanaverbecq